# Reinforcement Learning with Lookahead Information

**Nadav Merlis**
FairPlay Joint Team, CREST, ENSAE Paris
`nadav.merlis@ensae.fr`

## Abstract

We study reinforcement learning (RL) problems in which agents observe the reward or transition realizations at their current state *before deciding which action to take*. Such observations are available in many applications, including transactions, navigation and more. When the environment is known, previous work shows that this lookahead information can drastically increase the collected reward. However, outside of specific applications, existing approaches for interacting with unknown environments are not well-adapted to these observations. In this work, we close this gap and design provably-efficient learning algorithms able to incorporate lookahead information. To achieve this, we perform planning using the empirical distribution of the reward and transition observations, in contrast to vanilla approaches that only rely on estimated expectations. We prove that our algorithms achieve tight regret versus a baseline that also has access to lookahead information – linearly increasing the amount of collected reward compared to agents that cannot handle lookahead information.

## 1 Introduction

In reinforcement learning (RL), agents sequentially interact with a changing environment, aiming to collect as much reward as possible. While performing actions that yield immediate rewards is enticing, agents must also bear in mind that actions influence the state of the environment, affecting the potential reward that could be collected in future steps. When the environment is unknown, agents also need to balance reward maximization based on previous data and exploration – gathering of data that might improve future reward collection.

In the standard interaction model, at each timestep, agents first choose an action and only then observe its outcome on the rewards and state dynamics. As such, agents can only maximize the expected rewards, collected through the expected dynamics. Yet, in many applications, some information on the immediate outcome of actions is known *before* actions are performed. For example, when agents interact through transactions, prices and traded goods are usually agreed upon before performing any exchange ('reward information'). Alternatively, in navigation problems, nearby traffic information is known to the agent before choosing which path to go through ('transition information').

In a recent work, Merlis et al. [2024] shows that even for agents with full statistical knowledge of the environment, such 'lookahead' information can drastically increase the reward collected by agents – by a multiplicative factor of up to $AH$ when immediate rewards are revealed in advance and $A^{H/2}$ when observing the immediate future transitions.[1] Intuitively, agents do not only gain from instantaneously using this information – they can also adapt their planning to account for lookahead information being revealed in subsequent states, significantly increasing their future values. However, the work of Merlis et al. [2024] only tackles planning settings in which the model is known and does not provide algorithms or guarantees when interacting with unknown environments.

---

[1] $A$ is the size of the action space, $S$ is the size of the state space and $H$ is the interaction length.

38th Conference on Neural Information Processing Systems (NeurIPS 2024).

In this work, we aim to design provably-efficient agents that learn how to interact when given immediate ('one-step lookahead') reward or transition information before choosing an action, under the episodic tabular Markov Decision Process model. While such information can always be embedded into the state of the environment, the state space becomes exponential at best, and continuous at worst, rendering most theoretically-guaranteed approaches both computationally and statistically intractable. To alleviate this, we start by deriving dynamic programming ('Bellman') equations *in the original state space* that characterize the optimal lookahead policies. Inspired by these update rules, we present two variants to the MVP algorithm [Zhang et al., 2021b] that allow incorporating either reward or transition lookahead. In particular, we suggest a planning procedure that uses the empirical distribution of the reward/transition observations (instead of the estimated expectations), which might also be applied to other complex settings. We prove that these algorithms achieve tight regret bounds of $\tilde{\mathcal{O}}\left(\sqrt{H^3 SAK}\right)$ and $\tilde{\mathcal{O}}\left(\sqrt{H^2 SK}(\sqrt{H} + \sqrt{A})\right)$ after $K$ episodes (for reward and transition lookahead, respectively), compared to a stronger baseline that also has access to lookahead information. As such, they can collect significantly more rewards than vanilla RL algorithms.

**Outline.** We formally define RL problems with reward/transition lookahead in Section 2 and further discuss the differences between our setting and standard RL problems in Section 3. Then, we present our results in two complementary sections: Section 4 analyzes reward lookahead while Section 5 analyzes transition lookahead. We end with conclusions and future directions in Section 6.

**Related Work.**    Problems with varying lookahead information have been extensively studied in control, with model predictive control [MPC, Camacho et al., 2007] as the most notable example. Conceptually, when interacting with an environment that might be too complex or hard to model, it is oftentimes convenient to use a simpler model that allows accurately predicting its behavior just in the near future. MPC uses such models to repeatedly update its policy using short-term planning. In some cases, the utilized future predictions consist of additive perturbations to the dynamics [Yu et al., 2020], while other cases involve more general future predictions on the model behavior [Li et al., 2019, Zhang et al., 2021a, Lin et al., 2021, 2022]. To the best of our knowledge, these studies focus on comparing the performance of the controller to one with full future information (and thus, linear regret is inevitable), sometimes also considering prediction errors. They do not, however, attempt to learn the predictions. In contrast, we estimate the reward/transition distributions and leverage them to better plan, thus increasing the value gained by the agent. In addition, these works focus on continuous (mostly linear) control problems, whereas we study tabular settings; results from any one of these settings cannot be directly applied to the other.

In RL, lookahead is mostly used as a planning tool; namely, agents test the possible outcomes after performing multiple steps to decide which actions to take or to better estimate the value [Tamar et al., 2017, Efroni et al., 2019a, 2020, Moerland et al., 2020, Rosenberg et al., 2023, El Shar and Jiang, 2020, Biedenkapp et al., 2021, Huang et al., 2019]. Specifically, the future value at the end of the lookahead is often estimated using rollouts, and a longer lookahead is more robust to suboptimality of the rollout policy [Bertsekas, 2023]. However, when agents actually interact with the environment, no additional lookahead information is observed. One notable exception is [Merlis et al., 2024], which analyzes the potential value increase due to multi-step reward lookahead information (and briefly mentions transition lookahead). However, they only tackle planning settings, where the model is known, and do not study learning. In this work, we continue a long line of literature on regret analysis for tabular RL [Jaksch et al., 2010, Jin et al., 2018, Dann et al., 2019, Zanette and Brunskill, 2019, Efroni et al., 2019b, 2021, Simchowitz and Jamieson, 2019, Zhang et al., 2021b, 2023]. Yet, we are not aware of any existing results on regret minimization with reward or transition lookahead information.

Finally, various applications that involve one-step lookahead information have been previously studied. The most notable ones are prophet problems [Correa et al., 2019], where one-step reward lookahead is obtained, and the Canadian traveler problem with resampling [Nikolova and Karger, 2008], which can be formulated through one-step transition lookahead. We discuss the relation to these problems and the relevant existing results when analyzing each type of feedback, and also discuss the relation between transition lookahead and stochastic action sets [Boutilier et al., 2018].

## 2   Setting and Notations

We study episodic tabular Markov Decision Processes (MDPs), defined by the tuple $\mathcal{M} = (\mathcal{S}, \mathcal{A}, H, P, \mathcal{R})$, where $\mathcal{S}$ is the state space (of size $S$), $\mathcal{A}$ is the action space (of size $A$) and $H$ is the

interaction horizon. At each timestep $h \in \{1, \ldots, H\} \triangleq [H]$ of an episode $k \in [K]$, an agent, located in state $s_h^k \in \mathcal{S}$, chooses an action $a_h^k \in \mathcal{A}$ and obtains a reward $R_h^k = R_h(s_h^k, a_h^k) \sim \mathcal{R}_h(s_h^k, a_h^k)$. We assume that the rewards are supported by $[0, 1]$ and of expectations $r_h(s, a)$. Afterward, the environment transitions to a state $s_{h+1}^k \sim P_h(\cdot | s_h^k, a_h^k)$ and the interaction continues until the end of the episode. We use the notation $\boldsymbol{R} \sim \mathcal{R}_h(s)$ (or $\boldsymbol{s}' \sim P_h(s)$) to denote reward (next-state) samples for all actions simultaneously at step $h$ and state $s$ and assume independence between different timesteps.[2] On the other hand, samples from different actions at a specific state/timestep are not necessarily independent.

**Reward Lookahead.** With one-step reward lookahead at timestep $h$ and state $s$, agents first observe the rewards for all actions $\boldsymbol{R}_h(s) \triangleq \{R_h(s, a)\}_{a \in \mathcal{A}}$ and only then choose an action to perform. Formally, we define the set of reward lookahead policies as $\Pi^R = \{\pi : [H] \times \mathcal{S} \times [0, 1]^A \mapsto \Delta_{\mathcal{A}}\}$, where $\Delta_{\mathcal{A}}$ is the probability simplex, and denote $a_h = \pi_h(s_h, \boldsymbol{R}_h)$. The value of a reward lookahead agent is the cumulative rewards gathered by it starting at timestep $h$ and state $s$, denoted by

$$V_h^{R,\pi}(s) = \mathbb{E}\left[\sum_{t=h}^{H} R_t(s_t, \pi_t(s_t, \boldsymbol{R}_t(s_t))) | s_h = s\right].$$

We also define the optimal reward lookahead value to be $V_h^{R,*}(s) = \max_{\pi \in \Pi^R} V_h^{R,\pi}(s)$. When interacting with an unknown environment for $K$ episodes, agents sequentially choose reward lookahead policies $\pi^k \in \Pi^R$ based on all historical information and are measured by their regret,

$$\text{Reg}^R(K) = \sum_{k=1}^{K} \left(V_1^{R,*}(s_1^k) - V_1^{R,\pi^k}(s_1^k)\right).$$

We allow the initial state of each episode $s_1^k$ to be arbitrarily chosen.

**Transition Lookahead.** Denoting $s'_{h+1}(s, a)$, the future state when playing action $a$ at step $h$ and state $s$, one-step transition lookahead agents observe $\boldsymbol{s}'_{h+1}(s) \triangleq \{s'_{h+1}(s, a)\}_{a \in \mathcal{A}}$ before acting. The set of transition lookahead agents is denoted by $\Pi^T = \{\pi : [H] \times \mathcal{S} \times \mathcal{S}^A \mapsto \Delta_{\mathcal{A}}\}$ with values

$$V_h^{T,\pi}(s) = \mathbb{E}\left[\sum_{t=h}^{H} R_t(s_t, \pi_t(s_t, \boldsymbol{s}'_{t+1}(s_t))) | s_h = s\right].$$

The optimal value is $V_h^{T,*}(s) = \max_{\pi \in \Pi^T} V_h^{T,\pi}(s)$, and we similarly define the regret versus optimal transition lookahead agents as $\text{Reg}^T(K) = \sum_{k=1}^{K} \left(V_1^{T,*}(s_1^k) - V_1^{T,\pi^k}(s_1^k)\right)$.

When the type of lookahead is clear from the context, we sometimes denote values by $V_h^\pi$ and $V_h^*$.

**Other Notations.** For any $p \in \Delta_n$ and $V \in \mathbb{R}^n$, we define $\text{Var}_p(V) = \sum_{i=1}^{n} p_i V_i^2 - \left(\sum_{i=1}^{n} p_i V_i\right)^2$. Also, given a transition kernel $P$ and a vector $V \in \mathbb{R}^S$, we let $PV(s, a) = \sum_{s' \in \mathcal{S}} P(s' | s, a) V(s')$ and similarly define it for value or transition kernel differences. We denote by $n_h^k(s, a)$, the number of times the pair $(s, a)$ was visited at timestep $h$ up to episode $k$ (inclusive) and similarly denote $n_h^k(s) = \sum_{a \in \mathcal{A}} n_h^k(s, a)$. We also let $\hat{r}_h^k(s, a) = \frac{1}{n_h^k(s, a)} \sum_{k'=1}^{k} \mathbb{1}\left\{s_h^{k'} = s, a_h^{k'} = a\right\} R_h^{k'}$ and $\hat{P}_h(s' | s, a) = \frac{1}{n_h^k(s, a)} \sum_{k'=1}^{k} \mathbb{1}\left\{s_h^{k'} = s, a_h^{k'} = a, s_{h+1}^{k'} = s'\right\}$ be the empirical expected rewards and transition kernel at $(s_h, a_h) = (s, a)$ using data up to episode $k$ and assume they are initialized to be zero. Finally, we denote by $\hat{\mathcal{R}}_h^k(s)$, the empirical reward distribution across all actions, and use $\hat{P}_h^k(s)$ to denote the empirical joint next-state distribution for all actions. In particular, if $k_i$ is the $i^{th}$ episode where $s$ was visited at step $h$, to sample $\boldsymbol{R} \sim \hat{\mathcal{R}}_h^k(s)$, we uniformly sample $i \sim U\left([n_h^k(s)]\right)$ and return $\boldsymbol{R} = \left\{R_h^{k_i}(s, a)\right\}_{a \in \mathcal{A}}$. A sample $\boldsymbol{s}' \sim \hat{P}_h^k(s)$ similarly returns $\boldsymbol{s}' = \left\{s_{h+1}'^{k_i}(s, a)\right\}_{a \in \mathcal{A}}$.

When we want to indicate the distribution used to calculate an expectation, we sometimes state it in a subscript, e.g., write $E_{\mathcal{R}_h(s)}[R(a)]$ to indicate that $R(a) \sim \mathcal{R}_h(s, a)$ or use $\mathbb{E}_{\mathcal{M}}$ to emphasize

---

[2]This assumption is not used by our algorithms: it is only to ensure that the optimal policy is Markovian.

that all distributions are according to an environment $\mathcal{M}$. In this paper, $\mathcal{O}$-notation only hides absolute constants while $\tilde{\mathcal{O}}$ hides factors of $\text{polylog}(S, A, H, K, \delta)$. We also use the notation $a \vee b = \max\{a, b\}$.

## 3 Comparing the Values of Lookahead Agents and Vanilla RL agents

In the classic RL formulation [e.g., Azar et al., 2017], agents only observe the reward and transition after performing an action and aim to maximize the 'no-lookahead' value, defined by

$$V_h^\pi(s) = \mathbb{E}\left[\sum_{t=h}^{H} r_t(s_t, \pi_t(s_t))|s_h = s\right],$$

where $\pi \in \Pi^\mathcal{M} = \{\pi : [H] \times \mathcal{S} \mapsto \Delta_\mathcal{A}\}$ is a Markovian policy. The optimal value is $V_h^{no}(s) = \max_{\pi \in \Pi^\mathcal{M}} V_h^\pi(s)$ and the regret is classically defined as $\text{Reg}(K) = \sum_{k=1}^{K}\left(V_1^{no}(s_1^k) - V_1^{\pi^k}(s_1^k)\right)$.

By definition, the set of lookahead policies also includes all Markovian policies (since agents are not obliged to use reward/transition information), so the optimal lookahead values are always larger than their no-lookahead counterpart. In other words, denoting the value gain due to lookahead information by $G^R(s) = V_1^{R,*}(s) - V_1^{no}(s)$ and $G^T(s) = V_1^{T,*}(s) - V_1^{no}(s)$, it holds that $G^R(s), G^T(s) \geq 0$. In terms of regret, for any fixed algorithm, we can also write

$$\text{Reg}(K) = \text{Reg}^R(K) - \sum_{k=1}^{K} G^R(s_1^k) = \text{Reg}^T(K) - \sum_{k=1}^{K} G^T(s_1^k).$$

As the value gains are non-negative, it directly implies that any regret bound w.r.t. the lookahead value also leads to the same bound for the standard regret. Even more so, in most cases, lookahead information leads to a strict improvement in the value, that is, $G^R(s), G^T(s) \geq G_0 > 0$. When this happens, any algorithm with sub-linear lookahead regret enjoys a *negative linear* standard regret:

*If* $\text{Reg}^R(K) = o(K)$ *and* $G^R(s_1^k) \geq G_0$ *for all* $k \in [K]$, *then* $\text{Reg}(K) \leq -G_0 K + o(K)$.

The same also holds for transition lookahead. Conversely, any agent that suffers positive standard regret will suffer linear regret compared to the best lookahead agent, i.e.,

*If* $\text{Reg}(K) \geq 0$ *and* $G^R(s_1^k) \geq G_0$ *for all* $k \in [K]$, *then* $\text{Reg}^R(K) \geq G_0 K$.

Notably, any agent that does not use lookahead information will suffer linear lookahead regret in any such environment. We now present two illustrative examples for environments where the lookahead value gain is significant, one for reward lookahead and another for transition lookahead.

**Reward lookahead.** Consider a simple 2-state environment, depicted in Figure 1. Starting at $s_i$, agents can either stay there by playing $a_1$, earning no reward, or play any other action and move to the absorbing $s_f$, obtaining a Bernoulli reward $Ber(1/(A-1)H)$. Actions in the terminal state $s_f$ yield no reward. Without observing the rewards, agents will arbitrarily move from $s_i$ to $s_f$, obtaining a reward $V^{no} = 1/(A-1)H$ in expectation. On the other hand, when agents observe the rewards before acting, they should move from $s_i$ to $s_f$ only if a reward was realized for some action (and otherwise, stay in $s_i$ by playing $a_1$). Such agents will have $(A-1)H$ opportunities to observe a unit reward across all timesteps and actions, collecting in expectation $V^{R,*} = (1 - 1/(A-1)H)^{(A-1)H} \geq 1 - 1/e$. In other words, just by observing the rewards before acting, the agent's value multiplicatively increases by almost $V^{R,*}/V^{no} \approx AH$.

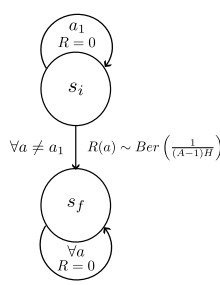

Figure 1: Two-state prophet-like problem

Moreover, the additive value gain is $G^R \approx 1 - 1/e$, so sub-linear lookahead regret with reward information results with a negatively-linear standard regret of $\text{Reg}(K) \lesssim -(1 - 1/e)K$.

**Transition lookahead.** Consider a chain of $H/2$ states (also described in further detail at Appendix C.9 and depicted at Figure 2). In each state, one action deterministically keeps the agent in its

current state, while all other actions move the agent one state forward w.p. $1/A$, but lead to a terminal non-rewarding state otherwise. If the reward is located at the end of the chain, any standard RL agent can collect it only at an exponentially low probability. On the other hand, transition lookahead agents would move forward only if there is an action that allows it while staying at their current state otherwise; such agents will collect the rewards at the end of the chain with constant probability. More specifically, any no-lookahead agent can collect at most $V^{no} = \mathcal{O}(HA^{-H/2})$ rewards, while transition lookahead agents can collect $V^{T,*} = \Omega(H)$; as such, lookahead agents achieve exponential increase in value, and sublinear regret versus the best lookahead agent will yield a standard regret of $\mathrm{Reg}(K) \lesssim -HK$.

In the following sections, we will present agents that are guaranteed to always achieve sublinear regret compared to the best lookahead agent.

## 4    Planning and Learning with One-Step Reward Lookahead

In this section, we analyze RL settings with one-step reward lookahead, in which immediate rewards are observed before choosing an action. One well-known example of this situation is the prophet problem [Correa et al., 2019], where an agent sequentially observes values from known distributions. Upon observing a value, the agent decides whether to take it as a reward and stop the interaction, or discard it and continue to observe more values. This problem has numerous applications and extensions concerning auctions and posted-price mechanisms [Correa et al., 2017]. As shown in [Merlis et al., 2024], it is critical to observe the distribution values before taking a decision; otherwise, the agent's revenue can decrease by a factor of $H$. Notably, the example presented in Figure 1 is a small variant of the prophet problem, where the agent can either take one of $A - 1$ values and finish the interaction or discard them and continue playing by staying at $s_i$; we showed that for this example, the lookahead information increases the value by a factor of $V^{R,*}/V^{no} \approx AH$.

The most natural way to tackle this setting is to extend (augment) the state space to contain the observed rewards; this way, we transition from a state and reward observations to a new state with new reward observations and return to the vanilla MDP formulation. However, this comes at a great cost. Even for Bernoulli rewards, there are $2^A$ possible reward combinations at any given state, and the augmentation increases the state space by this factor – leading to an exponentially-large state space. Even worse, for continuous rewards, the augmented state space becomes continuous, and any performance guarantees that depend on the size of the state space immediately become vacuous. Hence, algorithms that naïvely use this reduction are expected to be both computationally and statistically intractable. We refer to Appendix B.2 for further details on one such augmentation.

We take a different approach and derive Bellman equations for this setting in the *original state space*.

**Proposition 1.** *The optimal value of one-step reward lookahead agents satisfies*

$$V_{H+1}^{R,*}(s) = 0, \qquad\qquad\qquad\qquad \forall s \in \mathcal{S},$$

$$V_h^{R,*}(s) = \mathbb{E}_{\boldsymbol{R} \sim \mathcal{R}_h(s)}\left[\max_{a \in \mathcal{A}}\left\{ R_h(s,a) + \sum_{s' \in \mathcal{S}} P_h(s'|s,a)V_{h+1}^{R,*}(s') \right\}\right], \quad \forall s \in \mathcal{S}, h \in [H].$$

*Also, given reward observations $\boldsymbol{R} = \{R(a)\}_{a \in \mathcal{A}}$ at state $s$ and step $h$, the optimal policy is*

$$\pi_h^*(s, \boldsymbol{R}) \in \arg\max_{a \in \mathcal{A}}\left\{ R(a) + \sum_{s' \in \mathcal{S}} P_h(s'|s,a)V_{h+1}^{R,*}(s') \right\}.$$

We prove Proposition 1 in Appendix B.2, where we present an equivalent environment with extended state space in which one could apply the standard Bellman equations [Puterman, 2014] to calculate the value with reward lookahead. In contrast to the previously discussed augmentation approach, we find it more convenient to divide the augmentation into two steps – at odd steps $2h - 1$, the augmented environment would be in a state $s_h \times \boldsymbol{0}$, while at even steps $2h$, the state is $s_h \times \boldsymbol{R}_h$. Doing so creates an overlap between the values of the original and augmented environments at odd steps, simplifying the proofs. We also use this augmentation to prove a variant of the law of total variance [LTV, e.g. Azar et al., 2017] and a value-difference lemma [e.g. Efroni et al., 2019b].

We remark that calculating the exact value is not always tractable – even for $S = H = 1$ (bandit problems) and Gaussian rewards, Proposition 1 requires calculating the expectation of the maximum

---

**Algorithm 1** Monotonic Value Propagation with Reward Lookahead (MVP-RL)

---

1: **Require:** $\delta \in (0,1)$, bonuses $b_{k,h}^r(s), b_{k,h}^p(s,a)$
2: **for** $k = 1, 2, \ldots$ **do**
3:     Initialize $\bar{V}_{H+1}^k(s) = 0$
4:     **for** $h = H, H-1, \ldots, 1$ **do**
5:        Calculate the truncated values for all $s \in \mathcal{S}$

$$\bar{V}_h^k(s) = \min\left\{ \mathbb{E}_{\boldsymbol{R} \sim \hat{\mathcal{R}}_h^{k-1}(s)}\left[ \max_{a \in \mathcal{A}}\left\{ R(a) + b_{k,h}^p(s,a) + \hat{P}_h^{k-1}\bar{V}_{h+1}^k(s,a) \right\} \right] + b_{k,h}^r(s), H \right\}$$

6:     **end for**
7:     **for** $h = 1, 2, \ldots H$ **do**
8:        Observe $s_h^k$ and $R_h^k(s_h^k, a)$ for all $a \in \mathcal{A}$
9:        Play an action $a_h^k \in \arg\max_{a \in \mathcal{A}}\left\{ R_h^k(s_h^k, a) + b_{k,h}^p(s_h^k, a) + \hat{P}_h^{k-1}\bar{V}_{h+1}^k(s_h^k, a) \right\}$
10:       Collect the reward $R_h^k(s_h^k, a_h^k)$ and transition to the next state $s_{h+1}^k \sim P_h(\cdot | s_h^k, a_h^k)$
11:     **end for**
12: **end for**

---

of Gaussian random variables, which does not admit any simple closed-form solution. On the other hand, these equations allow approximating the value by using reward samples – in the following, we show that it can be used to achieve tight regret bounds when the environment is unknown.

### 4.1 Regret-Minimization with Reward Lookahead

We now present a tractable algorithm that achieves tight regret bounds with one-step reward lookahead. Specifically, we modify the Monotonic Value Propagation (MVP) algorithm [Zhang et al., 2021b] to perform planning using the *empirical reward distributions* – instead of using the empirical reward expectations. To compensate for transition uncertainty, we add a transition bonus that uses the variance of the optimistic next-state values (w.r.t. the empirical transition kernel), designed to be monotone in the future value. Such construction permits using the variance of optimistic values for the bonus calculation while being able to later replace it with the variance of the optimal value (see discussion in Zhang et al. 2021b). A reward bonus is used for the value calculation, but does not affect the action choice in the current state. Intuitively, this is because we get the same amount of information for all the actions of a state, so they have the same level of uncertainty – there is no need for bonuses to encourage reward exploration at the action level.

A high-level description of the algorithm is presented in Algorithm 1, while the full algorithm and its bonuses are stated in Appendix B.3. Notice that the planning requires calculating the expected maximum using the empirical distribution, whose support always contains at most $K$ elements, so both the memory and computations are polynomial. The algorithm ensures the following guarantees:

**Theorem 1.** *When running MVP-RL, with probability at least $1 - \delta$ uniformly for all $K \geq 1$, it holds that* $\text{Reg}^R(K) \leq \mathcal{O}\left( \sqrt{H^3 SAK} \ln \frac{SAHK}{\delta} + H^3 S^2 A \left( \ln \frac{SAHK}{\delta} \right)^2 \right)$.

See proof in Appendix B.7. Remarkably, our upper bound matches the standard lower bound for episodic RL of $\Omega\left( \sqrt{H^3 SAK} \right)$ [Domingues et al., 2021] up to log-factors; this lower bound is proved for known deterministic rewards, so in particular, it also holds for problems with reward lookahead.

To our knowledge, the only comparable bounds in settings with reward lookahead were proven to prophet problems; as agents observe (up to) $n$ distributions at a fixed order, it can be formulated as a deterministic chain-like MDP, with $H = n$, $S = n + 1$ and $A = 2$. Agents start at the head of the chain and can either advance without collecting a reward or collect the observed reward and move to a terminal non-rewarding state (for more details, see Merlis et al. 2024). For this problem, [Gatmiry et al., 2024] proved a regret bound of $\tilde{\mathcal{O}}(n^3 \sqrt{K})$ (albeit requiring a weaker form of feedback), and [Agarwal et al., 2023] proved a bound of $\tilde{\mathcal{O}}(n \sqrt{T})$ – slightly better than ours, but heavily relies on the ability to control which distributions to observe, which is a specific instance of deterministic transitions. We are unaware of any previous results that cover general Markovian dynamics.

## 4.2 Proof Concepts

When analyzing the regret of RL algorithms, a key step usually involves bounding the difference between the value of a policy in two different environments ('value-difference lemma'). In particular, for a given policy $\pi^k$, many algorithms maintain a confidence interval on the value $V_h^{\pi^k}(s) \in \left[\underline{V}_h^k(s), \bar{V}_h^k(s)\right]$, calculated based on optimistic and pessimistic MDPs that use the empirical model with bonuses/penalties [Dann et al., 2019, Zanette and Brunskill, 2019, Efroni et al., 2021]. Then, the instantaneous regret (without lookahead) is bounded using the optimistic values by

$$\bar{V}_h^k(s_h) - V_h^{\pi^k}(s_h) = \left(\hat{r}_h^{k-1}(s_h, a_h) - r_h(s_h, a_h)\right) + \left(\hat{P}_h^{k-1} - P_h\right)\bar{V}_h^k(s_h, a_h)$$
$$+ P_h\left(\bar{V}_{h+1}^k - V_{h+1}^{\pi^k}\right)(s_h, a_h) + \text{bonuses},$$

while the pessimistic values are used either as part of the bonuses or while bounding them. However, when trying to perform a similar decomposition with reward lookahead, we do not have the difference of expected rewards, but rather terms of the form

$$\mathbb{E}_{\boldsymbol{R} \sim \hat{\mathcal{R}}_h^{k-1}(s_h)}\left[R(\pi_h^k(s_h, \boldsymbol{R}))\right] - \mathbb{E}_{\boldsymbol{R} \sim \mathcal{R}_h(s_h)}\left[R(\pi_h^k(s_h, \boldsymbol{R}))\right]$$

(see, e.g., the last term of Lemma 4 in the appendix). As the action can be an arbitrary function of the reward realization, this term is extremely challenging to bound. For example, one could couple both distributions while trying to relate this error term to a Wasserstein distance between the empirical and real reward distribution; however, such distances exhibit much slower error rates than standard mean estimation [Fournier and Guillin, 2015]. Instead, we follow a different approach and show that uniformly for all possible expected next-state values $\hat{P}V \in [0, H]^A$ (as a function of the action at a given state), it holds w.h.p. that

$$\left|\mathbb{E}_{\boldsymbol{R} \sim \hat{\mathcal{R}}_h^{k-1}(s)}\left[\max_a\left\{R(a) + \hat{P}V(s, a)\right\}\right] - \mathbb{E}_{\boldsymbol{R} \sim \mathcal{R}_h(s)}\left[\max_a\left\{R(a) + \hat{P}V(s, a)\right\}\right]\right|$$
$$\lesssim \sqrt{\frac{A \ln \frac{1}{\delta}}{n_h^{k-1}(s) \vee 1}}. \tag{1}$$

Throughout the proof, whenever we face an expectation w.r.t. the empirical rewards, we reformulate the expression to fit the form of Equation (1) and use it as a 'change of measure' tool. We remark that while this confidence interval admits an extra $A$-factor compared to standard bounds, the counts only depend on the visits to the state (and not to the state-action), which compensates for this factor.

The choice of MVP for the bonus is similarly motivated – unlike some other bonuses (e.g., Zanette and Brunskill 2019), MVP does not require pessimistic values – either in the bonus itself or in its analysis. In contrast to the optimistic ones, the pessimistic values are not calculated via value iteration, but rather by following the policy $\pi^k$ in the pessimistic environment. As such, they cannot be easily manipulated to fit the form in Equation (1).

The analysis of the transitions adapts the techniques in [Efroni et al., 2021], while requiring extra care in handling the dependence of actions in the rewards.

## 5 Reinforcement Learning with One-Step Transition Lookahead

We now move to analyzing problems with one-step transition lookahead, where the resulting next state due to playing any of the actions is revealed before deciding which action to play. For example, consider the stochastic Canadian traveler problem with resampling [Nikolova and Karger, 2008, Boutilier et al., 2018]. In this problem, an agent wants to navigate on a graph as fast as possible from a source to a target, but observes which edges at a node are available only upon reaching this node. When edge availability is stochastic and resampled every time a node is visited, this is a clear case of one-step transition lookahead, as the information on the availability of edges is given before trying to traverse them. The example in Section 3 and Appendix C.9 is one possible formulation of this problem on a chain – agents are awarded for arriving at the end of the chain as fast as possible, but trying to use a non-existing edge results with termination. We showed that in this particular instance, the lookahead value is exponentially larger than the standard value, and any lookahead agent with low regret would greatly surpass no-lookahead agents.

As with reward lookahead, the future states for all actions can be embedded into the state, but doing so increases the size of the state space by a factor of $S^A$, again making this approach intractable (see Appendix C.2 for an example for such an extension). We once more show that this is not necessary; the transition-lookahead optimal values can be calculated using the following Bellman equations:

**Proposition 2.** *The optimal value of one-step transition lookahead agents satisfies*

$$V_{H+1}^{T,*}(s) = 0, \qquad\qquad\qquad\qquad \forall s \in \mathcal{S},$$

$$V_h^{T,*}(s) = \mathbb{E}_{\boldsymbol{s}' \sim P_h(s)}\left[\max_{a \in \mathcal{A}}\left\{r_h(s,a) + V_{h+1}^{T,*}(s'(s,a))\right\}\right], \qquad \forall s \in \mathcal{S}, h \in [H].$$

*Also, given next-state observations $\boldsymbol{s}' = \{s'(a)\}_{a \in \mathcal{A}}$ at state $s$ and step $h$, the optimal policy is*

$$\pi_h^*(s, \boldsymbol{s}') \in \arg\max_{a \in \mathcal{A}}\left\{r_h(s,a) + V_{h+1}^{T,*}(s'(a))\right\}.$$

The proof can be found at Appendix C.2 and again relies on augmenting the state space to incorporate the transitions; this time, we divide the episode into odd steps whose extended state is $s_h \times \boldsymbol{s}'_0$ (for an arbitrary fixed $\boldsymbol{s}'_0 \in \mathcal{S}^A$) and even steps with the state $s_h \times \boldsymbol{s}'_{h+1}$. Beyond planning, this again allows proving a variant of the LTV and of a value-difference lemma.

One important insight is that the policy $\pi_h^*(s, \boldsymbol{s}')$ admits the form of a *list*. Namely, consider the values $V_h^*(s, s', a) = r_h(s,a) + V_{h+1}^{T,*}(s')$ and assume some ordering of next-state-action pairs $\{(s'_i, a_i)\}_{i=1}^{SA}$ such that $V_h^*(s, s'_1, a_1) \geq \cdots \geq V_h^*(s, s'_{SA}, a_{SA})$. Then, an optimal policy would look at all realized pairs $(s'(a), a)$ and play the action with the highest location in this list. We refer the readers to Appendix C.4 for an additional discussion on list representations in transition lookahead.

Similar results could be achieved through a reduction to RL problems with stochastic action sets [Boutilier et al., 2018]. There, at every round, a subset of base actions is sampled, and only these actions are available to the agent. In particular, one could sample $A$ actions of the form $(s', a) \in \mathcal{S} \times \mathcal{A}$ and impose a deterministic transition to $s'$ given this extended action. However, since every original action must be sampled exactly once, this sampling procedure creates a dependence between pairs even when next-states at different actions are independent, adding unnecessary complications. We show that when transitions are independent between states, the expectation in Proposition 2 can be efficiently calculated (see Appendix C.4.1 for details), and otherwise, it can be approximated through sampling, as we do in learning settings.

## 5.1 Regret-Minimization with Transition Lookahead

Relying on similar principals as with reward lookahead, we now present MVP-TL, an adaptation of MVP to settings with one-step transition lookahead (summarized in Algorithm 2; the full details can be found at Appendix C.3). This time, we estimate the empirical expected reward and add a standard Hoeffding-like reward bonus, while performing planning using samples from the *empirical joint distribution* of the next-state for all the actions simultaneously. A variance-based transition bonus is added to the values; though this time, the variance also incorporates the rewards, namely

$$b_{k,h}^p(s) \approx \sqrt{\frac{\text{Var}_{\boldsymbol{s}' \sim \hat{P}_h^{k-1}(s)}(\bar{V}_h^k(s, \boldsymbol{s}'))}{n_h^{k-1}(s) \vee 1}}, \quad \bar{V}_h^k(s, \boldsymbol{s}') = \max_{a \in \mathcal{A}}\left\{\hat{r}_h^{k-1}(s,a) + b_{k,h}^r(s,a) + \bar{V}_{h+1}^k(s'(a))\right\}.$$

The motivation for this modification is the technical challenges described in Section 4.2, in the context of reward lookahead. For reward lookahead, we analyzed a value term that included both the rewards and next-state values, and used concentration arguments to move from the empirical reward distribution to the real one. For transition lookahead, similar values are analyzed, but we require variance-based concentration to obtain tighter regret bounds [Azar et al., 2017], so this variance naturally arises. The bonus is again designed to be monotone, as in the original MVP algorithm, and does not affect the immediate action choice – only the optimistic lookahead value. As before, the planning relies on sampling the next-state observations at previous episodes, and so it is polynomial, even if the precise joint distribution is complex. The algorithm enjoys the following regret bounds:

**Theorem 2.** *When running MVP-TL, with probability at least $1 - \delta$ uniformly for all $K \geq 1$, it holds that* $\text{Reg}^T(K) \leq \mathcal{O}\left(\sqrt{H^2 SK}\left(\sqrt{H} + \sqrt{A}\right)\ln\frac{SAHK}{\delta} + H^3 S^4 A^3\left(\ln\frac{SAHK}{\delta}\right)^2\right).$

---

**Algorithm 2** Monotonic Value Propagation with Transition Lookahead (MVP-TL)

---

1: **Require:** $\delta \in (0,1)$, bonuses $b^r_{k,h}(s,a), b^p_{k,h}(s)$
2: **for** $k = 1, 2, \dots$ **do**
3:     Initialize $\bar{V}^k_{H+1}(s) = 0$
4:     **for** $h = H, H-1, \dots, 1$ **do**
5:         Calculate the truncated values for all $s \in \mathcal{S}$

$$\bar{V}^k_h(s) = \min\left\{\mathbb{E}_{s' \sim \hat{P}^{k-1}_h(s)}\left[\max_{a \in \mathcal{A}}\left\{\hat{r}^{k-1}_h(s,a) + b^r_{k,h}(s,a) + \bar{V}^k_{h+1}(s'(a))\right\}\right] + b^p_{k,h}(s), H\right\}$$

6:     **end for**
7:     **for** $h = 1, 2, \dots H$ **do**
8:         Observe $s^k_h$ and $s'^k_{h+1}(s^k_h, a)$ for all $a \in \mathcal{A}$
9:         Play an action $a^k_h \in \arg\max_{a \in \mathcal{A}}\left\{\hat{r}^{k-1}_h(s^k_h, a) + b^r_{k,h}(s^k_h, a) + \bar{V}^k_{h+1}(s'^k_{h+1}(s^k_h, a))\right\}$
10:         Collect the reward $R^k_h \sim \mathcal{R}_h(s^k_h, a^k_h)$ and transition to the next state $s^k_{h+1} = s'^k_{h+1}(s^k_h, a^k_h)$
11:     **end for**
12: **end for**

---

See proof in Appendix C.8. For transition lookahead, the regret bounds we provide exhibit two rates, both corresponding to a natural adaptation of known lower bounds to transition lookahead.

1. *'Bandit rate'* $\mathcal{O}(\sqrt{H^2SAK})$: this is the rate due to reward stochasticity. Consider a problem where at odd timesteps $2h - 1$ and across all states, all actions have rewards of mean $1/2 - \epsilon$, except for one action of mean $1/2$. Assuming that the state-distribution is uniform, each such timestep forms a hard instance of a contextual bandit problem with $S$ contexts, exhibiting a regret of $\Omega(\sqrt{SAK})$ [Auer et al., 2002, Bubeck et al., 2012]. Since there are $H/2$ odd steps and we can design each step independently, the total regret would be $\Omega(H\sqrt{SAK})$. The even steps can be used to 'remove' the lookahead and create a uniform state distribution. To do so, we set that when taking an action at odd steps, we always transition to a fixed state $s_d$. From this state, one action $a_1$ leads uniformly to all states, while the rest of the actions lead to an absorbing non-rewarding state – rendering them strictly suboptimal. Thus, no-regret agents will only play $a_1$, regardless of the lookahead information, and the state distribution at odd timesteps will be uniform.

2. *'Transition learning rate'* $\mathcal{O}(\sqrt{H^3SK})$: recall that the vanilla RL lower bound designs a tree with $\Omega(S)$ leaves, to which agents need to navigate at the right timing (with $\Omega(H)$ options) and take the right action (out of $A$). While all leaves might transition agents to a rewarding state, one combination of state-action-timing has a slightly higher probability of doing so [Domingues et al., 2021]. This roughly creates a bandit problem with $SAH$ arms, constructed such that the maximal reward is $\Omega(H)$, yielding a total regret of $H\sqrt{HSAK}$. Now consider the following simple modification where in each leaf, only one action can lead to a reward (and the rest of the actions are 'useless' – never lead to rewards). Thus, the agent still needs to test all leaves at all timings, and so there are still $SH$ 'arms' with a corresponding regret of $\sqrt{H^3SK}$. Moreover, to test a leaf at a certain timing, we must navigate to it, and since the agent is going to play the single useful action at the leaf, transition lookahead does not provide any additional information.

As discussed before, transition lookahead can be formulated as an RL instance with stochastic action sets. While Boutilier et al. [2018] prove that with stochastic action sets, Q-learning asymptotically converges, they provide no learning algorithm nor regret bounds. Therefore, to our knowledge, our result is the first to achieve sublinear regret with transition lookahead.

### 5.2  Proof Concepts

Transition lookahead causes similar issues as reward lookahead. Hence, it is natural to apply a similar analysis approach – first, formulate the value as the expectation w.r.t. the next-state observations of the maximum of action-observation dependent values; then use uniform concentration as a 'change of measure' tool between the empirical and real next-state distribution. In particular, if $V(s, s', a)$ represents the value starting from state $s$, performing $a$ and transitioning to $s'$, one can show that for

all $V(s,\cdot,\cdot) \in [0,H]^{SA}$ (see Lemma 19),

$$\left| \mathbb{E}_{\boldsymbol{s}' \sim \hat{P}_h^{k-1}(s)} \left[ \max_a V(s, s'(a), a) \right] - \mathbb{E}_{\boldsymbol{s}' \sim P_h(s)} \left[ \max_a V(s, s'(a), a) \right] \right|$$

$$\lesssim \sqrt{\frac{SA \ln \frac{1}{\delta} \text{Var}_{\boldsymbol{s}' \sim \hat{P}_h^{k-1}(s)} \max_a V(s, s'(a), a)}{n_h^{k-1}(s) \vee 1}}, \quad (2)$$

where the variance term stems from using a Bernstein-like concentration bound. However, in contrast to the reward lookahead, the $\sqrt{SA}$-factor propagates to the dominant term of the regret, so pursuing this approach would lead to a worse regret bound of $\tilde{\mathcal{O}}\left( \sqrt{H^3 S^2 AK} \right)$.

To avoid this, we pinpoint the two locations where this change of measure is needed – the proof that $\bar{V}_h^k$ is optimistic and the regret decomposition – and make sure to perform this change of measure only on a single value $V_h^*(s, s', a) = r_h(s, a) + V_{h+1}^*(s')$, mitigating the need to cover all possible values and removing the additional $\sqrt{SA}$-factor. However, doing so leaves us with a residual term. Defining $V_h^*(s, \boldsymbol{s}') = \max_{a \in \mathcal{A}}\{V_h^*(s, s'(a), a)\}$ and assuming a similar optimistic value $\bar{V}_h^k(s, \boldsymbol{s}')$, this term is of the form

$$\mathbb{E}_{\boldsymbol{s}' \sim \hat{P}_h^{k-1}(s)}\left[ \bar{V}_h^k(s, \boldsymbol{s}') - V_h^*(s, \boldsymbol{s}') \right] - \mathbb{E}_{\boldsymbol{s}' \sim P_h(s)}\left[ \bar{V}_h^k(s, \boldsymbol{s}') - V_h^*(s, \boldsymbol{s}') \right].$$

While similar terms have been analyzed before [e.g., Zanette and Brunskill, 2019, Efroni et al., 2021], the analysis leads to a constant regret term that depends on the support of the distribution in question; in our case, it is the distribution over all possible next-states – of cardinality $S^A$. Therefore, following the same derivation would lead to an exponential additive regret term.

We overcome it by utilizing the fact that both the optimistic policy and the optimal one decide which action to take according to a list of next-state-actions $(s', a)$. In other words, instead of looking at the next-state $\boldsymbol{s}'$ (with $S^A$ possible values) to determine a value, we look at the highest-ranked realized pair $(s', a)$ in the list that corresponds to the policy that induces the value (with $SA$ possible rankings). Since we have two values, we need to calculate the probability of being at a certain list location for both $\pi^k$ and $\pi^*$, but the cardinality of this space is $(SA)^2$: polynomial and not exponential.

## 6 Conclusions and Future Work

In this work, we presented an RL setting in which immediate rewards or transitions are observed before actions are chosen. We showed how to design provably and computationally efficient algorithms for this setting that achieve tight regret bounds versus a strong baseline that also uses lookahead information. Our algorithms rely on estimating the distribution of the reward or transition observations, a concept that might be utilized in other settings. In particular, we believe that our techniques for transition lookahead could be extended to RL problems with stochastic action sets [Boutilier et al., 2018], but leave this for future work.

One natural extension to our work would be to consider multi-step lookahead information – observing the transition/rewards $L$ steps in advance. We conjecture that from a statistical point of view, a similar algorithmic approach that samples from the empirical observation distribution would be efficient. However, it is not clear how to perform efficient planning with such feedback.

Another possible direction would be to derive model-free algorithms [Jin et al., 2018], with the aim to improve the computation efficiency of the solutions; our model-based algorithms require at most $\mathcal{O}(KS^2AH)$ computations per episode due to the planning stage, while model-free algorithms might potentially allow just $\mathcal{O}(AH)$ computations per episode.

On the practical side, previous works presented RL algorithms that utilize/estimate a world model with multi-step lookahead to perform planning and learning [Schrittwieser et al., 2020, Chung et al., 2024], aiming to achieve the optimal no-lookahead value. For some of these approaches, it is quite natural to replace the simulated world behavior with lookahead information on the real future realization. We leave this adaptation and evaluation to future studies.

Finally, the notion of lookahead could be studied in various other decision-making settings (e.g., linear MDPs Jin et al. 2020) and can also be generalized to situations where lookahead information can be queried under some budget constraints [Efroni et al., 2021] or when agents only observe noisy lookahead predictions; we leave these problems for future research.

## Acknowledgements

We thank Alon Cohen and Austin Stromme for the helpful discussions. This project has received funding from the European Union's Horizon 2020 research and innovation programme under the Marie Skłodowska-Curie grant agreement No 101034255.

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

# Table of Contents

## A   Structure of the Appendix

Both reward and transition lookahead appendices share the following structure. First, we describe our assumption on the data generation process and analyze general properties of reward and transition lookahead. This is done by looking at an extended MDP that incorporates the lookahead information into the state. Then, we present the full algorithm and describe the relevant probabilistic events that ensure the concentration of all the empirical quantities. For transition lookahead, we require some additional notions for the event definitions (including the list representation of values and policies), which are explained in a separate subsection.

Given the concentration-related good event, we can prove that the planning procedure in the algorithm is optimistic, which we do in the subsequent subsection. Then, we define an additional good event that allows adding and removing conditional expectations in a way that will be needed for the proof.

At this point, we provided all (almost all) the results required for the regret analysis, and the proof of the main theorems is stated. The proofs also require some additional analysis for the bonuses (and especially variance terms), which is located at the end of the regret analysis.

For transition lookahead, the appendix includes one more part that further analyzes the example presented in Section 3.

At the end of the appendix, we state and prove several lemmas that will be used throughout our analysis, while also stating several existing results that will be of use.

# B Proofs for Reward Lookahead

## B.1 Data Generation Process

To simplify the proofs, we assume the following 'tabular' data-generation process: Before the game starts, a set of $K$ samples from the transition probabilities and rewards is generated for all $(s, a, h)$. Once a state $s$ at step $h$ is visited for the $i^{th}$ time, the $i^{th}$ sample from the reward distribution $\mathcal{R}_h(s)$ is the reward realization for all action $a \in \mathcal{A}$. When a state-action pair is visited for the $i^{th}$ time, the $i^{th}$ sample from the transition kernel $P_h(\cdot|s, a)$ determines the next-state realization. In particular, it implies that the reward samples from the first $i$ visits to a state are i.i.d., and the same for the next-states samples and state-action visitations. Throughout this appendix, we use the notation $\boldsymbol{R}_h^k = \left\{ R_h^k(s_h^k, a) \right\}_{a \in \mathcal{A}}$ to denote the reward observation at episode $k$ and timestep $h$ for all the actions.

For the proof, we define the following three filtrations. Let

$$F_{k,h} = \sigma\left( \left\{s_t^1, a_t^1, \boldsymbol{R}_t^1\right\}_{t \in [H]}, \dots, \left\{s_t^{k-1}, a_t^{k-1}, \boldsymbol{R}_t^{k-1}\right\}_{t \in [H]}, \left\{s_t^k, a_t^k, \boldsymbol{R}_t^k\right\}_{t \in [h]}, s_{h+1}^k \right),$$

$$F_{k,h}^R = \sigma\left( \left\{s_t^1, a_t^1, \boldsymbol{R}_t^1\right\}_{t \in [H]}, \dots, \left\{s_t^{k-1}, a_t^{k-1}, \boldsymbol{R}_t^{k-1}\right\}_{t \in [H]}, \left\{s_t^k, a_t^k, \boldsymbol{R}_t^k\right\}_{t \in [h+1]} \right),$$

the filtrations that contains all information until episode $k$ and step $h$, as well as the state at timestep $h + 1$, or all information of time $h + 1$, respectively. We make this distinction so that $F_{k,h-1}$ contains only $s_h^k$, while $F_{k,h-1}^R$ also contains $a_h^k$. We also define

$$F_k = \sigma\left( \left\{s_t^1, a_t^1, \boldsymbol{R}_t^1\right\}_{t \in [H]}, \dots, \left\{s_t^k, a_t^k, \boldsymbol{R}_t^k\right\}_{t \in [H]}, s_1^{k+1} \right),$$

which contains all information up to the end of the $k^{th}$ episode, as well as the initial state at episode $k + 1$.

## B.2 Extended MDP for Reward Lookahead

In this appendix, we present an alternative formulation of the one-step reward lookahead that falls under the vanilla (no-lookahead) model and would be helpful for the analysis.

Throughout the section, we study the relations between MDPs with and without reward lookahead, and between different MDPs with lookahead. Therefore, for clarity, we state the concerning MDP in the value, e.g. $V^{R,\pi}(s|\mathcal{M})$. Specifically in this subsection, we distinguish between values without lookahead (denoted $V^\pi$) and values with lookahead (denoted $V^{R,\pi}$). In the following subsections, unless stated otherwise, we will only consider lookahead values; for brevity, and with some abuse of notations, we will then omit the $R$ in the value notation.

For any MDP $\mathcal{M} = (\mathcal{S}, \mathcal{A}, H, P, \mathcal{R})$, define an equivalent extended MDP $\mathcal{M}^R$ of horizon $2H$ that separates the state transition and reward generation as follows:

1. Assume w.l.o.g. that $\mathcal{M}$ starts at some initial state $s_1$. The extended environment starts at a state $s_1 \times \boldsymbol{0}$, where $\boldsymbol{0} \in \mathbb{R}^A$ is the zeros vector.

2. For any $h \in [H]$, at timestep $2h - 1$, the environment $\mathcal{M}^R$ transitions from state $s_h \times \boldsymbol{0}$ to $s_h \times \boldsymbol{R}$, where $\boldsymbol{R} \sim \mathcal{R}_h(s)$ is a vector containing the rewards for all actions $a \in \mathcal{A}$. This transition occurs regardless of the action that was played. At timestep $2h$, given an action $a_h$ the environment transitions from $s_h \times \boldsymbol{R}$ to $s_{h+1} \times \boldsymbol{0}$, where $s_{h+1} \sim P_h(\cdot|s_h, a_h)$.

3. The reward at a state $s \times \boldsymbol{R}$ when playing an action $a$ is $R(a)$, namely, the reward is deterministic and only obtained on even timesteps.

We emphasize that throughout the section, we assume that $\mathcal{M}$ and $\mathcal{M}^R$ are coupled; that is, assume that under a policy $\pi$ in $\mathcal{M}$, the agent visits a state $s_h$, observes $\boldsymbol{R}_h$, plays an action $a_h$ and transitions to $s_{h+1}$. Then, in $\mathcal{M}^R$, the agent starts from $s_h \times \boldsymbol{0}$, transitions to $s_h \times \boldsymbol{R}$ (regardless of the action it played), takes the action $a_h$ and finally transitions to $s_{h+1} \times \boldsymbol{0}$.

Since the reward is embedded into the state, any state-dependent policy in $\mathcal{M}^R$ is a one-step reward lookahead policy in the original MDP. Moreover, the policy at the odd steps of $\mathcal{M}$ does not affect

the value, and assuming that the policy at the even steps in $\mathcal{M}^R$ is the same as the policy in $\mathcal{M}$, we trivially get the following relation between the values

$$V_{2h}^\pi(s, \boldsymbol{R}|\mathcal{M}^R) = \mathbb{E}\left[\sum_{t=h}^H R_t(s_t, a_t)|s_h = s, R_h(s, \cdot) = \boldsymbol{R}, \pi\right] \triangleq V_h^{R,\pi}(s, \boldsymbol{R}|\mathcal{M}),$$

$$V_{2h-1}^\pi(s, \boldsymbol{0}|\mathcal{M}^R) = \mathbb{E}\left[\sum_{t=h}^H R_t(s_t, a_t)|s_h = s, \pi\right] = V_h^{R,\pi}(s|\mathcal{M}). \tag{3}$$

While $\mathcal{M}^R$ has a continuous state space, which generally makes algorithm design impractical, this representation permits applying classic results on MDPs to environments with one-step lookahead.

As a remark, rewards could be directly embedded into the state without separating the state and reward updates. However, this creates unnecessary complications when analyzing the relations between similar environments. This is because we are mainly interested in the value given the state – in expectation over the realized rewards. In particular, value-difference are analyzed assuming a shared initial state, but in our case, we do not want to assume the same reward realization, but rather also account for the distance between reward distributions, which the step separation enables. For similar reasons, this representation also simplifies the proof of the law of total variance [Azar et al., 2017].

**Proposition 1.** *The optimal value of one-step reward lookahead agents satisfies*

$$V_{H+1}^{R,*}(s) = 0, \qquad\qquad \forall s \in \mathcal{S},$$

$$V_h^{R,*}(s) = \mathbb{E}_{\boldsymbol{R} \sim \mathcal{R}_h(s)}\left[\max_{a \in \mathcal{A}}\left\{R_h(s, a) + \sum_{s' \in \mathcal{S}} P_h(s'|s, a)V_{h+1}^{R,*}(s')\right\}\right], \quad \forall s \in \mathcal{S}, h \in [H].$$

*Also, given reward observations $\boldsymbol{R} = \{R(a)\}_{a \in \mathcal{A}}$ at state $s$ and step $h$, the optimal policy is*

$$\pi_h^*(s, \boldsymbol{R}) \in \arg\max_{a \in \mathcal{A}}\left\{R(a) + \sum_{s' \in \mathcal{S}} P_h(s'|s, a)V_{h+1}^{R,*}(s')\right\}.$$

*Proof.* We prove the result in the extended MDP $\mathcal{M}^R$ and remind the reader that in this formulation, the policy only uses state information, as in the standard RL formulation. In particular, it implies that there exists a Markovian optimal policy that uniformly maximizes the value (in the extended state space), and the optimal value is given through the dynamic-programming equations [Puterman, 2014]

$$V_{2H+1}^*(s, \boldsymbol{R}|\mathcal{M}^R) = 0, \qquad\qquad \forall s \in \mathcal{S}, \boldsymbol{R} \in \mathbb{R}^A,$$

$$V_{2h}^*(s, \boldsymbol{R}|\mathcal{M}^R) = \max_a\left\{R(a) + \sum_{s' \in \mathcal{S}} P_h(s'|s, a)V_{2h+1}^*(s', \boldsymbol{0}|\mathcal{M}^R)\right\}, \quad \forall h \in [H], s \in \mathcal{S}, \boldsymbol{R} \in \mathbb{R}^A,$$

$$V_{2h-1}^*(s, \boldsymbol{0}|\mathcal{M}^R) = \mathbb{E}_{\mathcal{R}_h(s)}\left[V_{2h}^*(s, \boldsymbol{R}|\mathcal{M}^R)\right], \qquad\qquad \forall h \in [H], s \in \mathcal{S}. \tag{4}$$

By the equivalence between $\mathcal{M}$ and $\mathcal{M}^R$ for all policies, this is also the optimal value in $\mathcal{M}$. Specifically, combining both recursion equations and substituting the relation between the original and extended values of Equation (3), we get the desired value recursion for any $h \in [H]$ and $s \in \mathcal{S}$:

$$\begin{aligned}
V_h^{R,*}(s|\mathcal{M}) &= V_{2h-1}^*(s, \boldsymbol{0}|\mathcal{M}^R) \\
&= \mathbb{E}_{\mathcal{R}_h(s)}\left[V_{2h}^*(s, \boldsymbol{R}|\mathcal{M}^R)\right] \\
&= \mathbb{E}_{\mathcal{R}_h(s)}\left[\max_a\left\{R(a) + \sum_{s' \in \mathcal{S}} P_h(s'|s, a)V_{2h+1}^*(s', \boldsymbol{0}|\mathcal{M}^R)\right\}\right] \\
&= \mathbb{E}_{\mathcal{R}_h(s)}\left[\max_a\left\{R(a) + \sum_{s' \in \mathcal{S}} P_h(s'|s, a)V_{h+1}^{R,*}(s|\mathcal{M})\right\}\right].
\end{aligned}$$

Similarly, for any $h \in [H]$, $s \in \mathcal{S}$ and $\boldsymbol{R} \in \mathbb{R}^A$, the optimal policy at the even stages of the extended MDP is

$$\pi_{2h}^*(s, \boldsymbol{R}) \in \arg\max_{a \in \mathcal{A}}\left\{R(a) + \sum_{s' \in \mathcal{S}} P_h(s'|s, a)V_{2h+1}^*(s', \boldsymbol{0}|\mathcal{M}^R)\right\},$$

alongside arbitrary actions at odd steps. Playing this policy in the original MDP will lead to an optimal one-step reward lookahead policy, as it achieves the optimal value of the original MDP. This policy directly translates to the optimal policy in the statement, by the equivalence between the original and extended MDPs and the relation $V_{2h+1}^*(s', \mathbf{0}|\mathcal{M}^R) = V_{h+1}^{R,*}(s'|\mathcal{M})$. $\qquad \square$

**Remark 1.** *As in Equation* (4)*, one could also write the dynamic programming equations for any policy* $\pi \in \Pi^R$*, namely*

$$V_{2h}^\pi(s, \mathbf{R}|\mathcal{M}^R) = R(\pi_h(s, \mathbf{R})) + \sum_{s' \in \mathcal{S}} P_h(s'|s, \pi_h(s, \mathbf{R}))V_{2h+1}^\pi(s', \mathbf{0}|\mathcal{M}^R), \quad \forall h \in [H], s \in \mathcal{S}, \mathbf{R} \in \mathbb{R}^A,$$

$$V_{2h-1}^\pi(s, \mathbf{0}|\mathcal{M}^R) = \mathbb{E}_{\mathcal{R}_h(s)}\left[V_{2h}^\pi(s, \mathbf{R}|\mathcal{M}^R)\right], \qquad\qquad \forall h \in [H], s \in \mathcal{S}.$$

*In particular, following the notation of Equation* (3)*, one can also write*

$$V_h^{R,\pi}(s, \mathbf{R}|\mathcal{M}) = R(\pi_h(s, \mathbf{R})) + \sum_{s' \in \mathcal{S}} P_h(s'|s, \pi_h(s, \mathbf{R}))V_{h+1}^{R,\pi}(s'|\mathcal{M}), \quad \text{and,}$$

$$V_h^{R,\pi}(s|\mathcal{M}) = \mathbb{E}_{\mathcal{R}_h(s)}\left[V_h^{R,\pi}(s, \mathbf{R}|\mathcal{M})\right]$$

$$= \mathbb{E}_{\mathcal{R}_h(s)}\left[R(\pi_h(s, \mathbf{R})) + \sum_{s' \in \mathcal{S}} P_h(s'|s, \pi_h(s, \mathbf{R}))V_{h+1}^{R,\pi}(s'|\mathcal{M}))\right].$$

*We will use this notation in some of the proofs.*

Another useful application of the extended MDP is a variation of the law of total variance (LTV), which will be useful in our analysis

**Lemma 3.** *For any deterministic one-step reward lookahead policy* $\pi \in \Pi^R$*, it holds that*

$$\mathbb{E}\left[\sum_{h=1}^H \mathrm{Var}_{P_h(\cdot|s_h,a_h)}(V_{h+1}^{R,\pi}(s_{h+1}))|\pi, s_1\right] \leq \mathbb{E}\left[\left(\sum_{h=1}^H R_h(s_h, a_h) - V_1^{R,\pi}(s_1)\right)^2 |\pi, s_1\right].$$

*Proof.* We apply the law of total variance (Lemma 27) in the extended MDP; there, the rewards are deterministic and equal to either $0$ (at odd steps) or $R_h(s_h, a_h)$ (at even steps), so the total expected rewards are $\sum_{h=1}^H R_h(s_h, a_h)$.

$$\mathbb{E}\left[\left(\sum_{h=1}^H R_h(s_h, a_h) - V_1^\pi(s_1, \mathbf{0}|\mathcal{M}^R)\right)^2 |\pi, s_1\right]$$

$$= \mathbb{E}\left[\underbrace{\sum_{h=1}^H \mathrm{Var}(V_{2h}^\pi(s_h, \mathbf{R}_h(s_h)|\mathcal{M}^R)|(s_h, \mathbf{0}))}_{\text{Odd steps}} + \underbrace{\sum_{h=1}^H \mathrm{Var}(V_{2h+1}^\pi(s_{h+1}, \mathbf{0}|\mathcal{M}^R)|(s_h, \mathbf{R}_h(s_h)))}_{\text{Even steps}} |\pi, s_1\right]$$

$$\geq \mathbb{E}\left[\sum_{h=1}^H \mathrm{Var}(V_{2h+1}^\pi(s_{h+1}, \mathbf{0}|\mathcal{M}^R)|(s_h, \mathbf{R}_h(s_h)))|\pi, s_1\right]$$

$$= \mathbb{E}\left[\sum_{h=1}^H \mathrm{Var}_{P_h(\cdot|s_h,a_h)}(V_{2h+1}^\pi(s_{h+1}, \mathbf{0}|\mathcal{M}^R))|\pi, s_1\right]$$

$$= \mathbb{E}\left[\sum_{h=1}^H \mathrm{Var}_{P_h(\cdot|s_h,a_h)}(V_{h+1}^{R,\pi}(s_{h+1}|\mathcal{M}))|\pi, s_1\right].$$

Noting that $V_1^\pi(s_1, \mathbf{0}|\mathcal{M}^R) = V_1^{R,\pi}(s_1|\mathcal{M})$ concludes the proof. $\qquad \square$

Finally, though not needed in our analysis, we use the extended MDP to prove the following value-difference lemma, which could be of further use in follow-up works. While we prove decomposition just using the next-step values, one could recursively apply the formula until the end of the episode to immediately get another formula that does not depend on the next value.

**Lemma 4** (Value-Difference Lemma with Reward Lookahead). *Let $\mathcal{M}_1 = (\mathcal{S}, \mathcal{A}, H, P^1, \mathcal{R}^1)$ and $\mathcal{M}_2 = (\mathcal{S}, \mathcal{A}, H, P^2, \mathcal{R}^2)$ be two environments. For any deterministic one-step reward lookahead policy $\pi \in \Pi^R$, any $h \in [H]$ and $s \in \mathcal{S}$, it holds that*

$$V_h^{R,\pi}(s|\mathcal{M}_1) - V_h^{R,\pi}(s|\mathcal{M}_2)$$

$$= \mathbb{E}_{\mathcal{M}_1}\left[V_{h+1}^{R,\pi}(s_{h+1}|\mathcal{M}_1) - V_{h+1}^{R,\pi}(s_{h+1}|\mathcal{M}_2)|s_h = s\right]$$

$$+ \mathbb{E}_{\mathcal{M}_1}\left[\sum_{s' \in \mathcal{S}}\left(P_h^1(s'|s_h, \pi_h(s_h, \boldsymbol{R}_h)) - P_h^2(s'|s_h, \pi_h(s_h, \boldsymbol{R}_h))\right)V_{h+1}^{R,\pi}(s'|\mathcal{M}_2)|s_h = s\right]$$

$$+ \mathbb{E}_{\mathcal{M}_1}\left[\mathbb{E}_{\mathcal{R}_h^1(s)}\left[V_h^{R,\pi}(s_h, \boldsymbol{R}|\mathcal{M}_2)\right] - \mathbb{E}_{\mathcal{R}_h^2(s)}\left[V_h^{R,\pi}(s_h, \boldsymbol{R}|\mathcal{M}_2)\right]|s_h = s\right],$$

*where $V_h^{R,\pi}(s, \boldsymbol{R}|\mathcal{M})$ is the value at a state given the reward realization, defined in Equation (3) and given in Remark 1.*

*Proof.* We again work with the extended MDPs $\mathcal{M}_1^R, \mathcal{M}_2^R$. Since under the extension, both the environments and the policy are Markovian, all values obey the following Bellman equations:

$$V_{2h}^\pi(s, \boldsymbol{R}|\mathcal{M}^R) = R(\pi_h(s, \boldsymbol{R})) + \sum_{s' \in \mathcal{S}} P_h(s'|s, \pi(s, \boldsymbol{R}))V_{2h+1}^\pi(s', \boldsymbol{0}|\mathcal{M}^R), \quad \forall h \in [H], s \in \mathcal{S}, \boldsymbol{R} \in \mathbb{R}^A$$

$$V_{2h-1}^\pi(s, \boldsymbol{0}|\mathcal{M}^R) = \mathbb{E}_{\mathcal{R}_h(s)}\left[V_{2h}^\pi(s, \boldsymbol{R}|\mathcal{M}^R)\right], \qquad\qquad\qquad \forall h \in [H], s \in \mathcal{S}.$$

Using the relation between the value of the original and extended MDP (eq. (3)) and the Bellman equations of the extended MDP, for any $h \in [H]$, we have

$$V_h^{R,\pi}(s|\mathcal{M}_1) - V_h^{R,\pi}(s|\mathcal{M}_2)$$
$$= V_{2h-1}^\pi(s, \boldsymbol{0}|\mathcal{M}_1^R) - V_{2h-1}^\pi(s, \boldsymbol{0}|\mathcal{M}_2^R)$$
$$= \mathbb{E}_{\mathcal{R}_h^1(s)}\left[V_{2h}^\pi(s, \boldsymbol{R}|\mathcal{M}_1^R)\right] - \mathbb{E}_{\mathcal{R}_h^2(s)}\left[V_{2h}^\pi(s, \boldsymbol{R}|\mathcal{M}_2^R)\right]$$
$$= \mathbb{E}_{\mathcal{R}_h^1(s)}\left[V_{2h}^\pi(s, \boldsymbol{R}|\mathcal{M}_1^R) - V_{2h}^\pi(s, \boldsymbol{R}|\mathcal{M}_2^R)\right] + \mathbb{E}_{\mathcal{R}_h^1(s)}\left[V_{2h}^\pi(s, \boldsymbol{R}|\mathcal{M}_2^R)\right] - \mathbb{E}_{\mathcal{R}_h^2(s)}\left[V_{2h}^\pi(s, \boldsymbol{R}|\mathcal{M}_2^R)\right]$$
$$= \mathbb{E}_{\mathcal{R}_h^1(s)}\left[V_{2h}^\pi(s, \boldsymbol{R}|\mathcal{M}_1^R) - V_{2h}^\pi(s, \boldsymbol{R}|\mathcal{M}_2^R)\right] + \mathbb{E}_{\mathcal{R}_h^1(s)}\left[V_h^{R,\pi}(s, \boldsymbol{R}|\mathcal{M}_2)\right] - \mathbb{E}_{\mathcal{R}_h^2(s)}\left[V_h^{R,\pi}(s, \boldsymbol{R}|\mathcal{M}_2)\right]$$
$$= \mathbb{E}_{\mathcal{M}_1}\left[V_{2h}^\pi(s_h, \boldsymbol{R}_h|\mathcal{M}_1^R) - V_{2h}^\pi(s_h, \boldsymbol{R}_h|\mathcal{M}_2^R)|s_h = s\right]$$
$$+ \mathbb{E}_{\mathcal{R}_h^1(s)}\left[V_h^{R,\pi}(s, \boldsymbol{R}|\mathcal{M}_2)\right] - \mathbb{E}_{\mathcal{R}_h^2(s)}\left[V_h^{R,\pi}(s, \boldsymbol{R}|\mathcal{M}_2)\right]. \tag{5}$$

We now focus on the first term. Denoting $a_h = \pi_h(s_h, \boldsymbol{R}_h)$ the action taken by the agent at environment $\mathcal{M}_1$, We have

$$V_{2h}^\pi(s_h, \boldsymbol{R}_h|\mathcal{M}_1^R) - V_{2h}^\pi(s_h, \boldsymbol{R}_h|\mathcal{M}_2^R)$$

$$= \left(R_h(a_h) + \sum_{s' \in \mathcal{S}} P_h^1(s'|s_h, a_h)V_{2h+1}^\pi(s', \boldsymbol{0}|\mathcal{M}_1^R)\right)$$

$$- \left(R_h(a_h) + \sum_{s' \in \mathcal{S}} P_h^2(s'|s_h, a_h)V_{2h+1}^\pi(s', \boldsymbol{0}|\mathcal{M}_2^R)\right)$$

$$= \sum_{s' \in \mathcal{S}} P_h^1(s'|s_h, a_h)V_{h+1}^{R,\pi}(s'|\mathcal{M}_1) - \sum_{s' \in \mathcal{S}} P_h^2(s'|s_h, a_h)V_{h+1}^{R,\pi}(s'|\mathcal{M}_2)$$

$$= \sum_{s' \in \mathcal{S}} P_h^1(s'|s_h, a_h)\left(V_{h+1}^{R,\pi}(s'|\mathcal{M}_1) - V_{h+1}^{R,\pi}(s'|\mathcal{M}_2)\right)$$

$$+ \sum_{s' \in \mathcal{S}}\left(P_h^1(s'|s_h, a_h) - P_h^2(s'|s_h, a_h)\right)V_{h+1}^{R,\pi}(s'|\mathcal{M}_2)$$

$$= E_{\mathcal{M}_1}\left[V_{h+1}^{R,\pi}(s_{h+1}|\mathcal{M}_1) - V_{h+1}^{R,\pi}(s_{h+1}|\mathcal{M}_2)|s_h, a_h\right]$$

$$+ \sum_{s' \in \mathcal{S}}\left(P_h^1(s'|s_h, a_h) - P_h^2(s'|s_h, a_h)\right)V_{h+1}^{R,\pi}(s'|\mathcal{M}_2).$$

Substituting this back into Equation (5), we have

$$
\begin{aligned}
V_h^\pi(s|\mathcal{M}_1) &- V_h^\pi(s|\mathcal{M}_2) \\
&= \mathbb{E}_{\mathcal{M}_1}\left[E_{\mathcal{M}_1}\left[V_{h+1}^{R,\pi}(s_{h+1}|\mathcal{M}_1) - V_{h+1}^{R,\pi}(s_{h+1}|\mathcal{M}_2)|s_h, a_h\right]|s_h = s\right] \\
&\quad + \mathbb{E}_{\mathcal{M}_1}\left[\sum_{s'\in\mathcal{S}}\left(P_h^1(s'|s_h, a_h) - P_h^2(s'|s_h, a_h)\right)V_{h+1}^{R,\pi}(s'|\mathcal{M}_2)|s_h = s\right] \\
&\quad + \mathbb{E}_{\mathcal{R}_h^1(s)}\left[V_h^{R,\pi}(s, \boldsymbol{R}|\mathcal{M}_2)\right] - \mathbb{E}_{\mathcal{R}_h^2(s)}\left[V_h^{R,\pi}(s, \boldsymbol{R}|\mathcal{M}_2)\right] \\
&= \mathbb{E}_{\mathcal{M}_1}\left[V_{h+1}^{R,\pi}(s_{h+1}|\mathcal{M}_1) - V_{h+1}^{R,\pi}(s_{h+1}|\mathcal{M}_2)|s_h = s\right] \\
&\quad + \mathbb{E}_{\mathcal{M}_1}\left[\sum_{s'\in\mathcal{S}}\left(P_h^1(s'|s_h, \pi_h(s_h, \boldsymbol{R}_h)) - P_h^2(s'|s_h, \pi_h(s_h, \boldsymbol{R}_h))\right)V_{h+1}^{R,\pi}(s'|\mathcal{M}_2)|s_h = s\right] \\
&\quad + \mathbb{E}_{\mathcal{M}_1}\left[\mathbb{E}_{\mathcal{R}_h^1(s)}\left[V_h^{R,\pi}(s_h, \boldsymbol{R}|\mathcal{M}_2)\right] - \mathbb{E}_{\mathcal{R}_h^2(s)}\left[V_h^{R,\pi}(s_h, \boldsymbol{R}|\mathcal{M}_2)\right]|s_h = s\right].
\end{aligned}
$$

$\square$

## B.3 Full Algorithm Description for Reward Lookahead

---

**Algorithm 3** Monotonic Value Propagation with Reward Lookahead (MVP-RL)

---

1: **Require:** $\delta \in (0,1)$, bonuses $b_{k,h}^r(s), b_{k,h}^p(s,a)$
2: **for** $k = 1, 2, ...$ **do**
3:     Initialize $\bar{V}_{H+1}^k(s) = 0$
4:     **for** $h = H, H-1, .., 1$ **do**
5:         **for** $s \in \mathcal{S}$ **do**
6:             **if** $n_h^{k-1}(s) = 0$ **then**
7:                 $\bar{V}_h^k(s) = H$
8:             **else**
9:                 Calculate the truncated values

$$\bar{V}_h^k(s) = \min\left\{ \frac{1}{n_h^{k-1}(s)} \sum_{t=1}^{n_h^{k-1}(s)} \max_{a \in \mathcal{A}}\left\{ R_h^{k_h^t(s)}(s,a) + b_{k,h}^p(s,a) + \hat{P}_h^{k-1}\bar{V}_{h+1}^k(s,a) \right\} + b_{k,h}^r(s), H \right\}$$

10:             **end if**
11:             For any vector $\boldsymbol{R} \in \mathbb{R}^A$, define the policy $\pi^k$

$$\pi_h^k(s, \boldsymbol{R}) \in \arg\max_{a \in \mathcal{A}}\left\{ R(a) + b_{k,h}^p(s,a) + \hat{P}_h^{k-1}\bar{V}_{h+1}^k(s,a) \right\}$$

12:         **end for**
13:     **end for**
14:     **for** $h = 1, 2, \ldots H$ **do**
15:         Observe $s_h^k$ and $\boldsymbol{R}_h^k = \left\{ R_h^k(s_h^k, a) \right\}_{a \in \mathcal{A}}$
16:         Play an action $a_h^k = \pi_h^k(s_h^k, \boldsymbol{R}_h^k)$
17:         Collect the reward $R_h^k(s_h^k, a_h^k)$ and transition to the next state $s_{h+1}^k \sim P_h(\cdot|s_h^k, a_h^k)$
18:     **end for**
19:     Update the empirical estimators and counts for all visited state-actions
20: **end for**

---

We use a variant of the MVP algorithm [Zhang et al., 2021b] while adapting their proof and the one from [Efroni et al., 2021]. The algorithm is described in Algorithm 3 and uses the following bonuses:

$$b_{k,h}^r(s) = 3\sqrt{\frac{AL_\delta^k}{2(n_h^{k-1}(s) \vee 1)}},$$

$$b_{k,h}^p(s,a) = \min\left\{ \frac{20}{3}\sqrt{\frac{\mathrm{Var}_{\hat{P}_h^{k-1}(\cdot|s,a)}(\bar{V}_{h+1}^k)L_\delta^k}{n_h^{k-1}(s,a) \vee 1}} + \frac{400}{9}\frac{HL_\delta^k}{n_h^{k-1}(s,a) \vee 1}, H \right\}$$

where $L_\delta^k = \ln\frac{144S^2AH^2k^3(k+1)}{\delta}$, and for brevity, we shorten $\mathrm{Var}_{\hat{P}_h^{k-1}(\cdot|s,a)}(\bar{V}_{h+1}^k(s'))$ to $\mathrm{Var}_{\hat{P}_h^{k-1}(\cdot|s,a)}(\bar{V}_{h+1}^k)$ (omitting the state from the value).

For the optimistic value iteration, we use the notation $k_h^t(s)$ to represent the $t^{th}$ episode where the state $s$ was visited at the $h^{th}$ timestep. Thus, line 9 of Algorithm 3 is the expectation w.r.t. the empirical reward distribution $\hat{\mathcal{R}}_h^{k-1}(s)$ (when defining its realization to be zero when $n_h^{k-1}(s) = 0$). Since the bonuses are larger than $H$ when $n_h^{k-1}(s) = 0$, one could write the update in more concisely as

$$\bar{V}_h^k(s) = \min\left\{ \mathbb{E}_{\boldsymbol{R} \sim \hat{\mathcal{R}}_h^{k-1}(s)}\left[ \max_{a \in \mathcal{A}}\left\{ R(a) + b_{k,h}^p(s,a) + \hat{P}_h^{k-1}\bar{V}_{h+1}^k(s,a) \right\} \right] + b_{k,h}^r(s), H \right\}.$$

We will often use this representation in our analysis.

## B.4 The First Good Event – Concentration

We now define the first good event, which ensures that all empirical quantities are well-concentrated. For the transitions, we require each element to concentrate well, as well as both the inner product and the variance w.r.t. the optimal value function. For the reward, we make sure that the maximum of the rewards to concentrate well (with any possible bias, that will later correspond with the next-state values). Formally, for any fixed vector $u \in \mathbb{R}^A$, denote

$$m_h(s, u) = \mathbb{E}_{\boldsymbol{R} \sim \mathcal{R}_h(s)}\left[\max_a \{R_h(a) + u(a)\}\right],$$

$$\hat{m}_h^k(s, u) = \mathbb{E}_{\boldsymbol{R} \sim \hat{\mathcal{R}}_h^k(s)}\left[\max_a \{R_h(a) + u(a)\}\right]$$

with the convention that $\hat{m}_h^k(s, u) = \max_a u(a)$ if $n_h^k(s) = 0$. We define the following good events:

$$E^p(k) = \left\{ \forall s, s', a, h : \ |P_h(s'|s, a) - \hat{P}_h^{k-1}(s'|s, a)| \leq \sqrt{\frac{2P(s'|s, a)L_\delta^k}{n_h^{k-1}(s, a) \vee 1}} + \frac{L_\delta^k}{n_h^{k-1}(s, a) \vee 1} \right\}$$

$$E^{pv1}(k) = \left\{ \forall s, a, h : \ \left|\left(\hat{P}_h^{k-1} - P_h\right)V_{h+1}^*(s, a)\right| \leq \sqrt{\frac{2\mathrm{Var}_{P_h(\cdot|s, a)}(V_{h+1}^*)L_\delta^k}{n_h^{k-1}(s, a) \vee 1}} + \frac{HL_\delta^k}{n_h^{k-1}(s, a) \vee 1} \right\}$$

$$E^{pv2}(k) = \left\{ \forall s, a, h : \ \left|\sqrt{\mathrm{Var}_{P_h(\cdot|s, a)}(V_{h+1}^*)} - \sqrt{\mathrm{Var}_{\hat{P}_h^{k-1}(\cdot|s, a)}(V_{h+1}^*)}\right| \leq 4H\sqrt{\frac{L_\delta^k}{n_h^{k-1}(s, a) \vee 1}} \right\}$$

$$E^r(k) = \left\{ \forall s, h, \forall u \in [0, 2H]^A : \ \left|m_h(s, u) - \hat{m}_h^{k-1}(s, u)\right| \leq 3\sqrt{\frac{AL_\delta^k}{2(n_h^{k-1}(s) \vee 1)}} \right\}$$

where we again use $L_\delta^k = \ln \frac{144S^2 AH^2 k^3(k+1)}{\delta}$. Then, we define the first good event as

$$\mathbb{G}_1 = \bigcap_{k \geq 1} E^r(k) \bigcap_{k \geq 1} E^p(k) \bigcap_{k \geq 1} E^{pv1}(k) \bigcap_{k \geq 1} E^{pv2}(k),$$

for which, the following holds:

**Lemma 5** (The First Good Event). *The good event $\mathbb{G}_1$ holds w.p. $\Pr(\mathbb{G}_1) \geq 1 - \delta/2$.*

*Proof.* The proof of the first three events uses standard concentration arguments (see, e.g., Efroni et al. 2021) and is stated for completeness. For any fixed $k \geq 1, s, a, h$ and number of visits $n \in [k]$, we utilize Lemma 16 w.r.t. the transition kernel $P_h(\cdot|s, a)$, the value $V_{h+1}^* \in [0, H]$ and probability $\delta' = \frac{\delta}{8SAHk^2(k+1)}$; notice that by the assumption that samples are generated i.i.d. before the game starts, given the number of visits, all samples are i.i.d., so standard concentration could be applied. By taking the union bound over all $n \in [k]$ and slightly increasing the constants to ensure that $n = 0$ trivially holds, we get that the events also hold for any number of visit $n_h^{k-1}(s, a) \in \{0 \dots, k\}$, and taking another union bound over all $k \geq 1, s, a, h$ ensures that each of the events $\cap_{k \geq 1} E^p(k), \cap_{k \geq 1} E^{pv1}(k)$ and $\cap_{k \geq 1} E^{pv2}(k)$ holds w.p. at least $1 - \frac{\delta}{8}$

We now focus on bounding the probability of the event $\cap_k E^r(k)$. For any fixed $k, h$ and $s$, observe that the event trivially holds if $n_h^k = 0$, then the event trivially holds, since for all $u \in [0, 2H]^A$,

$$\left|m_h(s, u) - \hat{m}_h^{k-1}(s, u)\right| = \left|\mathbb{E}_{\boldsymbol{R} \sim \mathcal{R}_h(s)}\left[\max_a \{R_h(s, a) + u(a)\}\right] - \max_a \{u(a)\}\right| \overset{(*)}{\leq} 1 \leq 3\sqrt{\frac{AL_\delta^k}{2}},$$

where $(*)$ uses the boundedness of the rewards in $[0, 1]$. Next, recall that for any fixed $n_h^{k-1} = n \in [k]$, the rewards samples at state $s$ and step $h$ are i.i.d. vectors on $[0, 1]^A$. Therefore, by Lemma 18,

$$\Pr\left\{ n_h^{k-1}(s) = n, \forall u \in [0, 2H]^A : \ \left|m_h(s, u) - \hat{m}_h^{k-1}(s, u)\right| > 3\sqrt{\frac{AL_\delta^k}{2(n_h^{k-1}(s) \vee 1)}} \right\} \leq \frac{\delta}{8SAHk^2(k+1)}.$$

Taking a union bound on all possible values of $n \in [k]$, $s$ and $h$, we get

$$\Pr\{E^r(k)\} \geq 1 - SAk \cdot \frac{\delta}{8SAHk^2(k+1)} \geq 1 - \frac{\delta}{8k(k+1)}.$$

By summing over all $k \geq 1$, the event $\cap_k E^r(k)$ holds with a probability of at least $1 - \delta/8$. Finally, taking the union bound with the other three events leads to the desired result of $\Pr(\mathbb{G}_1) \geq 1 - \delta/2$. □

## B.5 Optimism of the Upper Confidence Value Functions

In this subsection, we prove that under the good event $\mathbb{G}_1$, the values $\bar{V}^k$ that MVP-RL produces are optimistic.

**Lemma 6** (Optimism). *Under the first good event $\mathbb{G}_1$, for all $k \in [K]$, $h \in [H]$ and $s \in \mathcal{S}$, it holds that $V_h^*(s) \leq \bar{V}_h^k(s)$.*

*Proof.* The proof follows by backward induction on $H$; see that the claim trivially holds for $h = H+1$, where both values are defined to be zero.

Now assume by induction that for some $k \in [K]$ and $h \in [H]$, the desired inequalities hold at timestep $h + 1$ for all $s \in \mathcal{S}$; we will show that this implies that they also hold at timestep $h$.

At this point, we also assume w.l.o.g. that $\bar{V}_h^k(s) < H$, and in particular, the value is not truncated; otherwise, by the boundedness of the rewards, $V_h^*(s) \leq H = \bar{V}_h^k(s)$. For similar reasons, we assume w.l.o.g. that $b_{k,h}^p(s,a) < H$, so that it is also not truncated.

By the optimism of the value at step $h + 1$ due to the induction hypothesis and the monotonicity of the bonus (Lemma 23), under the good event, we have for all $s \in \mathcal{S}$ and $a \in \mathcal{A}$ that

$$\hat{P}_h^{k-1} \bar{V}_{h+1}^k(s,a) + b_{k,h}^p(s,a)$$

$$\geq \hat{P}_h^{k-1} \bar{V}_{h+1}^k(s,a) + \max\left\{ \frac{20}{3}\sqrt{\frac{\mathrm{Var}_{\hat{P}_h^{k-1}(\cdot|s,a)}(\bar{V}_{h+1}^k)L_\delta^k}{n_h^{k-1}(s,a) \vee 1}}, \frac{400}{9}\frac{HL_\delta^k}{n_h^{k-1}(s,a) \vee 1} \right\}$$

$$\geq \hat{P}_h^{k-1} V_{h+1}^*(s,a) + \max\left\{ \frac{20}{3}\sqrt{\frac{\mathrm{Var}_{\hat{P}_h^{k-1}(\cdot|s,a)}(V_{h+1}^*)L_\delta^k}{n_h^{k-1}(s,a) \vee 1}}, \frac{400}{9}\frac{HL_\delta^k}{n_h^{k-1}(s,a) \vee 1} \right\} \quad \text{(Lemma 23)}$$

$$\geq \hat{P}_h^{k-1} V_{h+1}^*(s,a) + \frac{10}{3}\sqrt{\frac{\mathrm{Var}_{\hat{P}_h^{k-1}(\cdot|s,a)}(V_{h+1}^*)L_\delta^k}{n_h^{k-1}(s,a) \vee 1}} + \frac{200}{9}\frac{HL_\delta^k}{n_h^{k-1}(s,a) \vee 1}$$

$$\geq \hat{P}_h^{k-1} V_{h+1}^*(s,a) + \frac{10}{3}\sqrt{\frac{\mathrm{Var}_{P_h(\cdot|s,a)}(V_{h+1}^*)L_\delta^k}{n_h^{k-1}(s,a) \vee 1}} + \frac{8HL_\delta^k}{n_h^{k-1}(s,a) \vee 1} \quad \text{(Under } E^{pv2}(k)\text{)}$$

$$\geq P_h V_{h+1}^*(s,a). \quad \text{(Under } E^{pv1}(k)\text{)}$$

Thus, under the good event and the induction hypothesis, we have that

$$\bar{V}_h^k(s) = \mathbb{E}_{\boldsymbol{R} \sim \hat{\mathcal{R}}_h(s)}\left[ \max_{a \in \mathcal{A}}\left\{ R(a) + b_{k,h}^p(s,a) + \hat{P}_h^{k-1}\bar{V}_{h+1}^k(s,a) \right\} \right] + b_{k,h}^r(s)$$

$$\geq \mathbb{E}_{\boldsymbol{R} \sim \hat{\mathcal{R}}_h(s)}\left[ \max_{a \in \mathcal{A}}\left\{ R(a) + P_h V_{h+1}^*(s,a) \right\} \right] + b_{k,h}^r(s).$$

In particular, using Proposition 1, we get

$$\bar{V}_h^k(s) - V_h^*(s) \geq \mathbb{E}_{\boldsymbol{R} \sim \hat{\mathcal{R}}_h(s)}\left[ \max_{a \in \mathcal{A}}\left\{ R(a) + P_h V_{h+1}^*(s,a) \right\} \right] + b_{k,h}^r(s)$$

$$- \mathbb{E}_{\boldsymbol{R} \sim \mathcal{R}_h(s)}\left[ \max_{a \in \mathcal{A}}\left\{ R(a) + P_h V_{h+1}^*(s,a) \right\} \right]$$

$$\geq 0,$$

where the last inequality holds under the event $E^r(k)$ with $u(a) = P_h V_{h+1}^*(s,a) \in [0, H]^A$. $\quad\square$

## B.6 The Second Good Event – Martingale Concentration

In this subsection, we present four good events that will allow us to replace the expectation over the randomizations inside each episode with their realization.

Define the following bonus-like term that will later appear in the proof due to value concentration:

$$b_{k,h}^{pv1}(s,a) = \min\left\{\sqrt{\frac{2\mathrm{Var}_{P_h(\cdot|s,a)}(V_{h+1}^*)L_\delta^k}{n_h^{k-1}(s,a) \vee 1}} + \frac{4H^2 S L_\delta^k}{n_h^{k-1}(s,a) \vee 1}, H\right\},$$

and let

$$Y_{1,h}^k := \bar{V}_{h+1}^k(s_{h+1}^k) - V_{h+1}^{\pi^k}(s_{h+1}^k),$$
$$Y_{2,h}^k = \mathrm{Var}_{P_h(\cdot|s_{t,h},a_{t,h})}(V_{h+1}^{\pi^k}),$$
$$Y_{3,h}^k = b_{k,h}^p(s_h^k,a_h^k) + b_{k,h}^{pv1}(s_h^k,a_h^k).$$

The second good event is the intersection of the events $\mathbb{G}_2 = E^{\mathrm{diff1}} \cap E^{\mathrm{diff2}} \cap E^{\mathrm{Var}} \cap E^{bp}$ defined as follows.

$$E^{\mathrm{diff1}} = \left\{\forall h \in [H], K \geq 1 : \sum_{k=1}^K \mathbb{E}[Y_{1,h}^k|F_{k,h-1}] \leq \left(1 + \frac{1}{2H}\right)\sum_{k=1}^K Y_{1,h}^k + 18H^2 \ln\frac{8HK(K+1)}{\delta}\right\},$$

$$E^{\mathrm{diff2}} = \left\{\forall h \in [H], K \geq 1 : \sum_{k=1}^K \mathbb{E}[Y_{1,h}^k|F_{k,h-1}^R] \leq \left(1 + \frac{1}{2H}\right)\sum_{k=1}^K Y_{1,h}^k + 18H^2 \ln\frac{8HK(K+1)}{\delta}\right\},$$

$$E^{\mathrm{Var}} = \left\{K \geq 1 : \sum_{k=1}^K\sum_{h=1}^H Y_{2,h}^k \leq 2\sum_{k=1}^K\sum_{h=1}^H \mathbb{E}[Y_{2,h}^k|F_{k-1}] + 4H^3 \ln\frac{8HK(K+1)}{\delta}\right\},$$

$$E^{bp} = \left\{\forall h \in [H], K \geq 1 : \sum_{k=1}^K \mathbb{E}[Y_{3,h}^k|F_{k,h-1}] \leq 2\sum_{k=1}^K Y_{3,h}^k + 50H^2 \ln\frac{8HK(K+1)}{\delta}\right\},$$

We define the good event $\mathbb{G} = \mathbb{G}_1 \cap \mathbb{G}_2$.

**Lemma 7.** *The good event $\mathbb{G}$ holds with a probability of at least $1 - \delta$.*

*Proof.* The proof follows similarly to Lemmas 15 and 21 of [Efroni et al., 2021].

First, define the random process $W_k = \mathbb{1}\left\{\bar{V}_h^k(s) - V_h^{\pi^k}(s) \in [0, H], \forall h \in [H], s \in \mathcal{S}\right\}$ and define $\tilde{Y}_{1,h}^k = W_k Y_{1,h}^k$, which is bounded in $[0, H]$. Also observe that $W_k$ is $F_{k-1}$ measurable, since both values and policies are calculated based on data up to the episode $k - 1$, and in particular, it is $F_{k,h-1}$ measurable and $\tilde{Y}_{1,h}^k$ is $F_{k,h}$ measurable. thus, by Lemma 25, for any $k \in [K]$ and $h \in [H]$, we have w.p. at least $1 - \frac{\delta}{8HK(K+1)}$ that

$$\sum_{k=1}^K \mathbb{E}[\tilde{Y}_{1,h}^k|F_{k,h-1}] \leq \left(1 + \frac{1}{2H}\right)\sum_{k=1}^K \tilde{Y}_{1,h}^k + 18H^2 \ln\frac{8HK(K+1)}{\delta}.$$

Since $W_k$ is $F_{k,h-1}$ measurable, we can write the event as

$$\sum_{k=1}^K W_k\mathbb{E}[Y_{1,h}^k|F_{k,h-1}] \leq \left(1 + \frac{1}{2H}\right)\sum_{k=1}^K W_k Y_{1,h}^k + 18H^2 \ln\frac{8HK(K+1)}{\delta},$$

and taking the union bound over all $h \in [H]$ and $K \geq 1$, we get w.p. at least $1 - \frac{\delta}{8}$ that the event

$$\tilde{E}^{\mathrm{diff1}} = \left\{\forall h \in [H], K \geq 1 : \sum_{k=1}^K W_k\mathbb{E}[Y_{1,h}^k|F_{k,h-1}] \leq \left(1 + \frac{1}{2H}\right)\sum_{k=1}^K W_k Y_{1,h}^k + 18H^2 \ln\frac{8HK(K+1)}{\delta}\right\}.$$

Importantly, by optimism (Lemma 6), under $\mathbb{G}_1$, it holds that $W_k = 1$ for all $k \geq 1$, so we immediately get that $\mathbb{G}_1 \cap \tilde{E}^{\mathrm{diff1}} = \mathbb{G}_1 \cap E^{\mathrm{diff1}}$.

Following the exact same proof just with the filtration $F_{k,h}^R$ and defining the equivalent $\tilde{E}^{\mathrm{diff2}}$, we get that this event also holds w.p. $1 - \frac{\delta}{8}$ and is the desired event when $\mathbb{G}_1$ holds.

Next, we prove that the other two events also hold w.p. at least $1 - \frac{\delta}{8}$.

By the assumptions of our setting, we know that $V_h^{\pi^k}(s) \in [0, H]$, and so

$$\sum_{h=1}^{H} Y_{2,h}^k = \sum_{h=1}^{H} \mathrm{Var}_{P_h(\cdot|s_{t,h}, a_{t,h})}(V_{h+1}^{\pi^k}) \in [0, H^3].$$

In particular, applying Lemma 25 (w.r.t. the filtration $F_k$) with $C = H^3$ and any fixed $K$, we get w.p. $1 - \frac{\delta}{8HK(K+1)}$ that

$$\sum_{k=1}^{K} \sum_{h=1}^{H} Y_{2,h}^k \leq 2 \sum_{k=1}^{K} \sum_{h=1}^{H} \mathbb{E}[Y_{2,h}^k | F_{k-1}] + 4H^3 \ln \frac{8HK(K+1)}{\delta}.$$

Taking the union bound on all possible values of $K \geq 1$ proves that $E^{\mathrm{Var}}$ holds w.p. at least $1 - \frac{\delta}{8}$.

Similarly, by definition, we have that $Y_{3,h}^k = b_{k,h}^p(s_h^k, a_h^k) + b_{k,h}^{pv1}(s_h^k, a_h^k) \in [0, 2H]$ and is $F_{k,h}$ measurable. Thus, for any fixed $k \geq 1$ and $h \in [H]$, using Lemma 25, we have w.p. $1 - \frac{\delta}{8HK(K+1)}$ that

$$\sum_{k=1}^{K} \mathbb{E}[Y_{3,h}^k | F_{k,h-1}] \leq \left(1 + \frac{1}{4H}\right) \sum_{k=1}^{K} Y_{3,h}^k + 50H^2 \ln \frac{8HK(K+1)}{\delta}$$

$$\leq 2 \sum_{k=1}^{K} Y_{3,h}^k + 50H^2 \ln \frac{8HK(K+1)}{\delta},$$

applying the union bound on all $K \geq 1$, the event $E^{bp}$ holds w.p. $1 - \frac{\delta}{8}$.

To summarize, we have that the event $\mathbb{G}_1$ holds w.p. $1 - \frac{\delta}{2}$ (Lemma 5), and we proved that the events $\tilde{E}^{\mathrm{diff1}}, \tilde{E}^{\mathrm{diff2}}, E^{\mathrm{Var}}, E^{bp}$ hold each w.p. $1 - \frac{\delta}{8}$, so we also have that the event

$$\mathbb{G} = \mathbb{G}_1 \cap \mathbb{G}_2$$
$$= \mathbb{G}_1 \cap E^{\mathrm{diff1}} \cap E^{\mathrm{diff2}} \cap E^{\mathrm{Var}} \cap E^{bp}$$
$$= \mathbb{G}_1 \cap \tilde{E}^{\mathrm{diff1}} \cap \tilde{E}^{\mathrm{diff2}} \cap E^{\mathrm{Var}} \cap E^{bp}$$

holds w.p. at least $1 - \delta$. $\qquad\square$

## B.7 Regret Analysis

We finally analyze the regret of the algorithm

**Theorem 1.** *When running MVP-RL, with probability at least $1 - \delta$ uniformly for all $K \geq 1$, it holds that* $\text{Reg}^R(K) \leq \mathcal{O}\left(\sqrt{H^3 SAK} \ln \frac{SAHK}{\delta} + H^3 S^2 A \left(\ln \frac{SAHK}{\delta}\right)^2\right).$

*Proof.* Assume that the good events $\mathbb{G}$ holds, which by Lemma 7, happens with probability at least $1 - \delta$. Then, by optimism (Lemma 6), for any $k \in [K]$, $h \in [H]$ and $s \in \mathcal{S}$, it holds that $V_h^*(s) \leq \bar{V}_h^k(s)$. Moreover, we can lower bound the value of the policy $\pi^k$ as follows (see Remark 1):

$$
\begin{aligned}
V_h^{\pi^k}(s) &= \mathbb{E}_{\boldsymbol{R} \sim \mathcal{R}_h(s)}\left[R(\pi_h^k(s, \boldsymbol{R})) + P_h V_{h+1}^{\pi^k}(s, \pi_h^k(s, \boldsymbol{R}))\right] \\
&= \mathbb{E}_{\boldsymbol{R} \sim \mathcal{R}_h(s)}\left[R(\pi_h^k(s, \boldsymbol{R})) + \hat{P}_h^{k-1} \bar{V}_{h+1}^k(s, \pi_h^k(s, \boldsymbol{R})) + b_{k,h}^p(s, \pi_h^k(s, \boldsymbol{R}))\right] \\
&\quad + \mathbb{E}_{\boldsymbol{R} \sim \mathcal{R}_h(s)}\left[P_h V_{h+1}^{\pi^k}(s, \pi_h^k(s, \boldsymbol{R})) - \hat{P}_h^{k-1} \bar{V}_{h+1}^k(s, \pi_h^k(s, \boldsymbol{R})) - b_{k,h}^p(s, \pi_h^k(s, \boldsymbol{R}))\right] \\
&\overset{(1)}{=} \mathbb{E}_{\boldsymbol{R} \sim \mathcal{R}_h(s)}\left[\max_{a \in \mathcal{A}}\left\{R(a) + \hat{P}_h^{k-1} \bar{V}_{h+1}^k(s, a) + b_{k,h}^p(s, a)\right\}\right] \\
&\quad + \mathbb{E}_{\boldsymbol{R} \sim \mathcal{R}_h(s)}\left[P_h V_{h+1}^{\pi^k}(s, \pi_h^k(s, \boldsymbol{R})) - \hat{P}_h^{k-1} \bar{V}_{h+1}^k(s, \pi_h^k(s, \boldsymbol{R})) - b_{k,h}^p(s, \pi_h^k(s, \boldsymbol{R}))\right] \\
&\overset{(2)}{\geq} \mathbb{E}_{\boldsymbol{R} \sim \hat{\mathcal{R}}_h^{k-1}(s)}\left[\max_{a \in \mathcal{A}}\left\{R(a) + \hat{P}_h^{k-1} \bar{V}_{h+1}^k(s, a) + b_{k,h}^p(s, a)\right\}\right] - b_{k,h}^r(s) \\
&\quad + \mathbb{E}_{\boldsymbol{R} \sim \mathcal{R}_h(s)}\left[P_h V_{h+1}^{\pi^k}(s, \pi_h^k(s, \boldsymbol{R})) - \hat{P}_h^{k-1} \bar{V}_{h+1}^k(s, \pi_h^k(s, \boldsymbol{R})) - b_{k,h}^p(s, \pi_h^k(s, \boldsymbol{R}))\right] \\
&\overset{(3)}{\geq} \bar{V}_h^k(s) - 2 b_{k,h}^r(s) \\
&\quad + \mathbb{E}_{\boldsymbol{R} \sim \mathcal{R}_h(s)}\left[P_h V_{h+1}^{\pi^k}(s, \pi_h^k(s, \boldsymbol{R})) - \hat{P}_h^{k-1} \bar{V}_{h+1}^k(s, \pi_h^k(s, \boldsymbol{R})) - b_{k,h}^p(s, \pi_h^k(s, \boldsymbol{R}))\right].
\end{aligned}
\tag{6}
$$

Relation (1) is by the definition of $\pi^k$ (see Algorithm 3), while (2) holds under the good event $E^r(k)$ with $u(a) = \hat{P}_h^{k-1} \bar{V}_{h+1}^k(s, a) + b_{k,h}^p(s, a) \in [0, 2H]$ (due to the value and bonus truncation). Finally, (3) is by the definition of $\bar{V}_h^k(s)$, where the inequality also accounts for its possible truncation.

To further bound this, we need to bound

$$
\begin{aligned}
\hat{P}_h^{k-1} \bar{V}_{h+1}^k(s, a) - P_h V_{h+1}^{\pi^k}(s, a) &= P_h\left(\bar{V}_{h+1}^k - V_{h+1}^{\pi^k}\right)(s, a) + \left(\hat{P}_h^{k-1} - P_h\right)\bar{V}_{h+1}^k(s, a) \\
&= P_h\left(\bar{V}_{h+1}^k - V_{h+1}^{\pi^k}\right)(s, a) \\
&\quad + \left(\hat{P}_h^{k-1} - P_h\right)V_{h+1}^*(s, a) + \left(\hat{P}_h^{k-1} - P_h\right)\left(\bar{V}_{h+1}^k - V_{h+1}^*\right)(s, a).
\end{aligned}
$$

The first error term can be bounded under the good event, while the second using Lemma 24. More formally, under the good event $E^{pv1}(k)$, we have

$$
\left|\left(\hat{P}_h^{k-1} - P_h\right)V_{h+1}^*(s, a)\right| \leq \sqrt{\frac{2\text{Var}_{P_h(\cdot|s,a)}(V_{h+1}^*) L_\delta^k}{n_h^{k-1}(s, a) \vee 1}} + \frac{H L_\delta^k}{n_h^{k-1}(s, a) \vee 1},
$$

and by Lemma 24 with $\alpha = 4H$ (using and $P_1 = P_h$, $P_2 = \hat{P}_h^{k-1}$, under $E^p(k)$),

$$
\begin{aligned}
\left|\left(\hat{P}_h^{k-1} - P_h\right)\left(\bar{V}_{h+1}^k - V_{h+1}^*\right)(s, a)\right| &\leq \frac{1}{4H}\mathbb{E}_{P_h(\cdot|s,a)}\left[\bar{V}_{h+1}^k(s') - V_{h+1}^*(s')\right] + \frac{HSL_\delta^k(1 + 4H \cdot 2/4)}{n_h^{k-1}(s, a) \vee 1} \\
&\leq \frac{1}{4H}\mathbb{E}_{P_h(\cdot|s,a)}\left[\bar{V}_{h+1}^k(s') - V_{h+1}^{\pi^k}(s')\right] + \frac{3H^2 SL_\delta^k}{n_h^{k-1}(s, a) \vee 1} \\
&= \frac{1}{4H}P_h\left(\bar{V}_{h+1}^k - V_{h+1}^{\pi^k}\right)(s, a) + \frac{3H^2 SL_\delta^k}{n_h^{k-1}(s, a) \vee 1},
\end{aligned}
$$

where the second inequality is since the value of $\pi^k$ cannot exceed the optimal value.

Since under the good event by Lemma 6, we have $0 \le V_{h+1}^{\pi^k}(s') \le V_{h+1}^*(s') \le \bar{V}_{h+1}^k(s') \le H$, we can trivially bound the error by $H$ and bound

$$\hat{P}_h^{k-1}\bar{V}_{h+1}^k(s,a) - P_h V_{h+1}^{\pi^k}(s,a)$$

$$\le \min\left\{ \left(1 + \frac{1}{4H}\right) \underbrace{P_h\left(\bar{V}_{h+1}^k - V_{h+1}^{\pi^k}\right)(s,a)}_{\ge 0} + \frac{3H^2 S L_\delta^k}{n_h^{k-1}(s,a) \vee 1} + \sqrt{\frac{2\mathrm{Var}_{P_h(\cdot|s,a)}(V_{h+1}^*)L_\delta^k}{n_h^{k-1}(s,a) \vee 1}} + \frac{HL_\delta^k}{n_h^{k-1}(s,a) \vee 1}, H \right\}$$

$$\le \left(1 + \frac{1}{4H}\right) P_h\left(\bar{V}_{h+1}^k - V_{h+1}^{\pi^k}\right)(s,a) + \min\left\{ \sqrt{\frac{2\mathrm{Var}_{P_h(\cdot|s,a)}(V_{h+1}^*)L_\delta^k}{n_h^{k-1}(s,a) \vee 1}} + \frac{4H^2 S L_\delta^k}{n_h^{k-1}(s,a) \vee 1}, H \right\}$$

$$\triangleq \left(1 + \frac{1}{4H}\right) P_h\left(\bar{V}_{h+1}^k - V_{h+1}^{\pi^k}\right)(s,a) + b_{k,h}^{pv1}(s,a).$$

Substituting back to Equation (6) while writing the linear operation $P_h V(s,a)$ as an expectation and letting the action be $a_h = \pi_h^k(s, \mathbf{R})$, we get under $\mathbb{G}$ for all $k \in [K]$, $h \in [H]$ and $s \in \mathcal{S}$ that

$$\bar{V}_h^k(s) - V_h^{\pi^k}(s)$$

$$\le \mathbb{E}_{\mathbf{R} \sim \mathcal{R}_h(s)}\left[ \hat{P}_h^{k-1}\bar{V}_{h+1}^k(s, \pi_h^k(s, \mathbf{R})) - P_h V_{h+1}^{\pi^k}(s, \pi_h^k(s, \mathbf{R})) + b_{k,h}^p(s, \pi_h^k(s, \mathbf{R})) \right] + 2b_{k,h}^r(s)$$

$$\le \mathbb{E}_{\mathbf{R} \sim \mathcal{R}_h(s)}\left[ \left(1 + \frac{1}{4H}\right)\mathbb{E}\left[\bar{V}_{h+1}^k(s_{h+1}) - V_{h+1}^{\pi^k}(s_{h+1})|s_h = s, a_h\right](s,a) + b_{k,h}^{pv1}(s, a_h) + b_{k,h}^p(s, a_h) \right] + 2b_{k,h}^r(s)$$

$$= \mathbb{E}\left[ \left(1 + \frac{1}{4H}\right)\left(\bar{V}_{h+1}^k(s_{h+1}) - V_{h+1}^{\pi^k}(s_{h+1})\right) + b_{k,h}^p(s_h, a_h) + b_{k,h}^{pv1}(s_h, a_h)|s_h = s, \pi^k \right] + 2b_{k,h}^r(s).$$

Next, taking $s = s_h^k$, the action $a_h = \pi_h^k(s, \mathbf{R})$ becomes $a_h^k$, and summing on all $k$, we can rewrite

$$\sum_{k=1}^K \bar{V}_h^k(s_h^k) - V_h^{\pi^k}(s_h^k)$$

$$\le \sum_{k=1}^K \mathbb{E}\left[ \left(1 + \frac{1}{4H}\right)\left(\bar{V}_{h+1}^k(s_{h+1}^k) - V_{h+1}^{\pi^k}(s_{h+1}^k)\right) + b_{k,h}^p(s_h^k, a_h^k) + b_{k,h}^{pv1}(s_h^k, a_h^k)|F_{k,h-1} \right] + 2\sum_{k=1}^K b_{k,h}^r(s_h^k)$$

$$\overset{(1)}{\le} \left(1 + \frac{1}{2H}\right)\left(1 + \frac{1}{4H}\right)\sum_{k=1}^K \left(\bar{V}_{h+1}^k(s_{h+1}^k) - V_{h+1}^{\pi^k}(s_{h+1}^k)\right)$$

$$+ 2\sum_{k=1}^K \left(b_{k,h}^p(s_h^k, a_h^k) + b_{k,h}^{pv1}(s_h^k, a_h^k)\right) + 2\sum_{k=1}^K b_{k,h}^r(s_h^k) + 68H^2 \ln\frac{8HK(K+1)}{\delta}$$

$$\overset{(2)}{\le} \left(1 + \frac{1}{2H}\right)\left(1 + \frac{1}{4H}\right)\sum_{k=1}^K \left(\bar{V}_{h+1}^k(s_{h+1}^k) - V_{h+1}^{\pi^k}(s_{h+1}^k)\right) + \frac{1}{4H}\left(1 + \frac{1}{2H}\right)\sum_{k=1}^K \left(\bar{V}_{h+1}^k(s_{h+1}^k) - V_{h+1}^{\pi^k}(s_{h+1}^k)\right)$$

$$+ 18\sum_{k=1}^K \sqrt{\frac{\mathrm{Var}_{P_h(\cdot|s_h^k, a_h^k)}(V_{h+1}^{\pi^k})L_\delta^k}{n_h^{k-1}(s_h^k, a_h^k) \vee 1}} + \sum_{k=1}^K \frac{1620H^2 S L_\delta^k}{n_h^{k-1}(s_h^k, a_h^k) \vee 1} + 68H^2 \ln\frac{8HK(K+1)}{\delta} + 2\sum_{k=1}^K b_{k,h}^r(s_h^k)$$

$$\le \left(1 + \frac{1}{2H}\right)^2 \sum_{k=1}^K \left(\bar{V}_{h+1}^k(s_{h+1}^k) - V_{h+1}^{\pi^k}(s_{h+1}^k)\right) + 18\sum_{k=1}^K \frac{\sqrt{L_\delta^k \mathrm{Var}_{P_h(\cdot|s_h^k, a_h^k)}(V_{h+1}^{\pi^k})}}{\sqrt{n_h^{k-1}(s_h^k, a_h^k) \vee 1}}$$

$$+ \sum_{k=1}^K \frac{1700H^2 S L_\delta^k}{n_h^{k-1}(s_h^k, a_h^k) \vee 1} + 6\sum_{k=1}^K \sqrt{\frac{A L_\delta^k}{2n_h^{k-1}(s) \vee 1}}$$

where inequality (1) holds when both $E^{\mathrm{diff}1}$ and $E^{bp}$ occur and inequality (2) is by Lemma 8. In the last inequality, we also substituted the definition of the reward bonus. Recursively applying this

inequality up to $h = H + 1$ (where both values are zero), w.p. at least $1 - \delta$, we get

$$
\begin{aligned}
\mathrm{Reg}^R(K) &\leq \sum_{k=1}^{K} \Big( V_1^*(s_1^k) - V_1^{\pi^k}(s_1^k) \Big) \\
&\leq \sum_{k=1}^{K} \Big( \bar{V}_1^k(s_1^k) - V_1^{\pi^k}(s_1^k) \Big) &\text{(Lemma 6)} \\
&\leq 18 \Big( 1 + \frac{1}{2H} \Big)^{2H} \sum_{k=1}^{K} \frac{\sqrt{L_\delta^k \mathrm{Var}_{P_h(\cdot | s_h^k, a_h^k)}(V_{h+1}^{\pi^k})}}{\sqrt{n_h^{k-1}(s_h^k, a_h^k) \vee 1}} + \Big( 1 + \frac{1}{2H} \Big)^{2H} \sum_{k=1}^{K} \frac{1700 H^2 S L_\delta^k}{n_h^{k-1}(s_h^k, a_h^k) \vee 1} \\
&\quad + 6 \Big( 1 + \frac{1}{2H} \Big)^{2H} \sum_{k=1}^{K} \sqrt{\frac{A L_\delta^k}{2 n_h^{k-1}(s) \vee 1}} \\
&\overset{(*)}{\leq} 100 \sqrt{H^3 S A K} L_\delta^K + 50 \sqrt{2SA} H^2 \big( L_\delta^K \big)^{1.5} \\
&\quad + 5000 H^2 S L_\delta^K \cdot SAH(2 + \ln(K)) + 12 \sqrt{A L_\delta^K} \Big( SH + 2\sqrt{SH^2 K} \Big) \\
&= \mathcal{O}\Big( \sqrt{H^3 S A K} L_\delta^K + H^3 S^2 A (L_\delta^K)^2 \Big).
\end{aligned}
$$

Relation $(*)$ is by Lemma 9 and Lemma 20. $\qquad\square$

### B.7.1 Lemmas for Bounding Bonus Terms

**Lemma 8.** *Conditioned on the good event* $\mathbb{G}$, *for any* $h \in [H]$, *it holds that*

$$\sum_{k=1}^{K}\Big(b_{k,h}^{p}(s_h^k, a_h^k) + b_{k,h}^{pv1}(s_h^k, a_h^k)\Big) \leq \frac{1}{8H}\Big(1 + \frac{1}{2H}\Big)\sum_{k=1}^{K}\Big(\bar{V}_{h+1}^k(s_{h+1}^k) - V_{h+1}^{\pi^k}(s_{h+1}^k)\Big)$$

$$+ 9\sum_{k=1}^{K}\sqrt{\frac{\mathrm{Var}_{P_h(\cdot|s_h^k,a_h^k)}(V_{h+1}^{\pi^k})L_\delta^k}{n_h^{k-1}(s_h^k,a_h^k)\vee 1}} + \sum_{k=1}^{K}\frac{810H^2SL_\delta^k}{n_h^{k-1}(s_h^k,a_h^k)\vee 1}.$$

*Proof.* We start by analyzing each of the terms separately. First, we apply Lemma 22 with $\alpha = \frac{20}{3} \cdot 32HL_\delta^k$, noting that under the good event (by Lemma 6), $0 \leq V_{h+1}^{\pi^k}(s) \leq V_{h+1}^*(s) \leq \bar{V}_{h+1}^k(s) \leq H$ and using the event $E^{pv}$; doing so yields

$$b_{k,h}^p(s,a) \leq \frac{20}{3}\sqrt{\frac{\mathrm{Var}_{\hat{P}_h^{k-1}(\cdot|s,a)}(\bar{V}_{h+1}^k)L_\delta^k}{n_h^{k-1}(s,a)\vee 1}} + \frac{400}{9}\frac{HL_\delta^k}{n_h^{k-1}(s,a)\vee 1}$$

$$\leq \frac{20\sqrt{L_\delta^k\mathrm{Var}_{P_h(\cdot|s,a)}(V_{h+1}^{\pi^k})}}{3\sqrt{n_h^{k-1}(s,a)\vee 1}} + \frac{1}{32H}P_h\Big(\bar{V}_{h+1}^k - V_{h+1}^{\pi^k}\Big)(s,a) + \frac{1}{32H}\hat{P}_h^{k-1}\Big(\bar{V}_{h+1}^k - V_{h+1}^{\pi^k}\Big)(s,a)$$

$$+ \frac{6400H^2L_\delta^k}{9n_h^{k-1}(s,a)\vee 1} + \frac{20}{3}\frac{4HL_\delta^k}{n_h^{k-1}(s,a)\vee 1} + \frac{400}{9}\frac{HL_\delta^k}{n_h^{k-1}(s,a)\vee 1}$$

Using Lemma 24 with $\alpha = 1$, under the good event $E^p(k)$ and for any $s, a$, we can further bound

$$\hat{P}_h^{k-1}\Big(\bar{V}_{h+1}^k - V_{h+1}^{\pi^k}\Big)(s,a)$$

$$= P_h\Big(\bar{V}_{h+1}^k - V_{h+1}^{\pi^k}\Big)(s,a) + \Big(\hat{P}_h^{k-1} - P_h\Big)\Big(\bar{V}_{h+1}^k(s') - V_{h+1}^{\pi^k}\Big)(s,a)$$

$$\leq P_h\Big(\bar{V}_{h+1}^k - V_{h+1}^{\pi^k}\Big)(s,a) + P_h\Big(\bar{V}_{h+1}^k - V_{h+1}^{\pi^k}\Big)(s,a) + \frac{HSL_\delta^k(1 + 2\cdot 1/4)}{n_h^{k-1}(s,a)\vee 1} \quad \text{(Lemma 24)}$$

$$\leq 2P_h\Big(\bar{V}_{h+1}^k - V_{h+1}^{\pi^k}\Big)(s,a) + \frac{1.5HSL_\delta^k}{n_h^{k-1}(s,a)\vee 1}$$

Thus, we get the overall bound

$$b_{k,h}^p(s,a) \leq \frac{20\sqrt{L_\delta^k\mathrm{Var}_{P_h(\cdot|s,a)}(V_{h+1}^{\pi^k})}}{3\sqrt{n_h^{k-1}(s,a)\vee 1}} + \frac{3}{32H}P_h\Big(\bar{V}_{h+1}^k - V_{h+1}^{\pi^k}\Big)(s,a) + \frac{785H^2SL_\delta^k}{n_h^{k-1}(s,a)\vee 1}$$

For the second bonus, we apply Lemma 21 w.r.t. $V_{h+1}^{\pi^k}(s) \leq V_{h+1}^*(s)$ and $\alpha = 32\sqrt{2L_\delta^k}H$ and get

$$b_{k,h}^{pv1}(s,a) \leq \sqrt{\frac{2\mathrm{Var}_{P_h(\cdot|s,a)}(V_{h+1}^*)L_\delta^k}{n_h^{k-1}(s,a)\vee 1}} + \frac{4H^2SL_\delta^k}{n_h^{k-1}(s,a)\vee 1}$$

$$\leq \sqrt{\frac{2\mathrm{Var}_{P_h(\cdot|s,a)}(V_{h+1}^{\pi^k})L_\delta^k}{n_h^{k-1}(s,a)\vee 1}} + \frac{1}{32H}P_h\Big(V_{h+1}^* - V_{h+1}^{\pi^k}\Big)(s,a) + \frac{16HL_\delta^k}{n_h^{k-1}(s,a)} + \frac{4H^2SL_\delta^k}{n_h^{k-1}(s,a)\vee 1}$$

$$\leq \sqrt{\frac{2\mathrm{Var}_{P_h(\cdot|s,a)}(V_{h+1}^{\pi^k})L_\delta^k}{n_h^{k-1}(s,a)\vee 1}} + \frac{1}{32H}P_h\Big(\bar{V}_{h+1}^k - V_{h+1}^{\pi^k}\Big)(s,a) + \frac{20H^2SL_\delta^k}{n_h^{k-1}(s,a)\vee 1}$$

where we again used the optimism. Combining both and summing over all $k$, we get

$$\sum_{k=1}^{K}\Big(b_{k,h}^p(s_h^k,a_h^k) + b_{k,h}^{pv1}(s_h^k,a_h^k)\Big) \leq 9\sum_{k=1}^{K}\sqrt{\frac{\mathrm{Var}_{P_h(\cdot|s_h^k,a_h^k)}(V_{h+1}^{\pi^k})L_\delta^k}{n_h^{k-1}(s_h^k,a_h^k)\vee 1}} + \frac{1}{8H}\sum_{k=1}^{K}P_h\Big(\bar{V}_{h+1}^k - V_{h+1}^{\pi^k}\Big)(s_h^k,a_h^k)$$

$$+ \sum_{k=1}^{K}\frac{805H^2SL_\delta^k}{n_h^{k-1}(s_h^k,a_h^k)\vee 1}$$

Finally, under the good event $E^{\mathrm{diff}2}$, it holds that

$$\sum_{k=1}^{K} P_h\left(\bar{V}_{h+1}^k - V_{h+1}^{\pi^k}\right)(s_h^k, a_h^k) = \sum_{k=1}^{K} \mathbb{E}\left[\bar{V}_{h+1}^k(s_{h+1}^k) - V_{h+1}^{\pi^k}(s_{h+1}^k)|F_{k,h-1}^R\right]$$

$$\leq \left(1 + \frac{1}{2H}\right)\sum_{k=1}^{K}\left(\bar{V}_{h+1}^k(s_{h+1}^k) - V_{h+1}^{\pi^k}(s_{h+1}^k)\right) + 18H^2 \ln\frac{8HK(K+1)}{\delta}.$$

Substituting this relation back concludes the proof. $\qquad\square$

**Lemma 9.** *Under the event $E^{\mathrm{Var}}$ it holds that*

$$\sum_{k=1}^{K}\sum_{h=1}^{H}\frac{\sqrt{\mathrm{Var}_{P_h(\cdot|s_h^k,a_h^k)}(V_{h+1}^{\pi^k})}}{\sqrt{n_h^{k-1}(s_h^k,a_h^k)\vee 1}} \leq 2\sqrt{H^3 SAK L_\delta^K} + \sqrt{8SA H^2 L_\delta^K}.$$

*Proof.* Following Lemma 24 of [Efroni et al., 2021], by Cauchy-Schwartz inequality, it holds that

$$\sum_{k=1}^{K}\sum_{h=1}^{H}\frac{\sqrt{\mathrm{Var}_{P_h(\cdot|s_h^k,a_h^k)}(V_{h+1}^{\pi^k})}}{\sqrt{n_h^{k-1}(s_h^k,a_h^k)\vee 1}} \leq \sqrt{\sum_{k=1}^{K}\sum_{h=1}^{H}\mathrm{Var}_{P_h(\cdot|s_h^k,a_h^k)}(V_{h+1}^{\pi^k})}\sqrt{\sum_{k=1}^{K}\sum_{h=1}^{H}\frac{1}{n_h^{k-1}(s_h^k,a_h^k)\vee 1}}.$$

The second term can be bounded by Lemma 20, namely,

$$\sum_{k=1}^{K}\sum_{h=1}^{H}\frac{1}{n_h^{k-1}(s_h^k,a_h^k)\vee 1} \leq SAH(2 + \ln(K)).$$

We further focus on bounding the first term. Under $E^{\mathrm{Var}}$, we have

$$\sum_{k=1}^{K}\sum_{h=1}^{H}\mathrm{Var}_{P_h(\cdot|s_h^k,a_h^k)}(V_{h+1}^{\pi^k})$$

$$\leq 2\sum_{k=1}^{K}\mathbb{E}\left[\sum_{h=1}^{H}\mathrm{Var}_{P_h(\cdot|s_h^k,a_h^k)}(V_{h+1}^{\pi^k})|F_{k-1}\right] + 4H^3 \ln\frac{8HK(K+1)}{\delta} \qquad \text{(Under } E^{\mathrm{Var}}\text{)}$$

$$\leq 2\sum_{k=1}^{K}\mathbb{E}\left[\left(\sum_{h=1}^{H}R_h(s_h^k,a_h^k) - V_1^{\pi^k}(s_1^k)\right)^2|F_{k-1}\right] + 4H^3 \ln\frac{8HK(K+1)}{\delta} \qquad \text{(By Lemma 3)}$$

$$\leq 2H^2 K + 4H^3 \ln\frac{8HK(K+1)}{\delta},$$

where the last inequality is since both the values and cumulative rewards are bounded in $[0, H]$. Combining both, we get

$$\sum_{k=1}^{K}\sum_{h=1}^{H}\frac{\sqrt{\mathrm{Var}_{P_h(\cdot|s_h^k,a_h^k)}(V_{h+1}^{\pi^k})}}{\sqrt{n_h^{k-1}(s_h^k,a_h^k)\vee 1}} \leq \sqrt{2H^2 K + 4H^3 \ln\frac{8HK(K+1)}{\delta}}\sqrt{SAH(2 + \ln(K))}$$

$$\leq \sqrt{2H^2 K + 4H^3 \ln\frac{8HK(K+1)}{\delta}}\sqrt{2SAH \ln\frac{8HK(K+1)}{\delta}}$$

$$\leq 2\sqrt{H^3 SAK L_\delta^K} + \sqrt{8SA H^2 L_\delta^K}.$$

$\qquad\square$

# C  Proofs for Transition Lookahead

## C.1  Data Generation Process

As for the reward transition, we also assume that all data was generated before the game starts for all state-action-timesteps, and it is given to the agent when the relevant $(s, a, h)$ is visited. Thus, the rewards and next-state from the first $i^{th}$ visits at a state (or a state-action pair) at a certain timestep are i.i.d.

Throughout this appendix, we use the notation $\boldsymbol{s}_{h+1}'^k = \left\{ s_{h+1}'^k(s_h^k, a) \right\}_{a \in \mathcal{A}}$ to denote the next-state observations at episode $k$ and timestep $h$ for all the actions, and use the equivalent filtrations to the ones defined at Appendix B.1, namely

$$F_{k,h} = \sigma\left( \left\{ s_t^1, a_t^1, \boldsymbol{s}_{t+1}'^1, R_t^1 \right\}_{t \in [H]}, \ldots, \left\{ s_t^{k-1}, a_t^{k-1}, \boldsymbol{s}_{t+1}'^{k-1}, R_t^{k-1} \right\}_{t \in [H]}, \left\{ s_t^k, a_t^k, \boldsymbol{s}_{t+1}'^k, R_t^k \right\}_{t \in [h]} \right),$$

$$F_k = \sigma\left( \left\{ s_t^1, a_t^1, \boldsymbol{s}_{t+1}'^1 \right\}_{t \in [H]}, \ldots, \left\{ s_t^k, a_t^k, \boldsymbol{s}_{t+1}'^k, R_t^k \right\}_{t \in [H]}, s_1^{k+1} \right).$$

In particular, notice that since both $\boldsymbol{s}_{h+1}'^k$ and $a_h^k$ are $F_{k,h}$ measurable, then so does $s_{h+1}^k$.

## C.2  Extended MDP for Transition Lookahead

In this appendix, we present an equivalent extended MDP that embeds the lookahead into the state to fall under the vanilla MDP model, similarly to Appendix B.2. We use this equivalence to apply various existing results on MDPs without the need to reprove them. We follow the same conventions as Appendix B.2 while denoting transition lookahead values by $V^{T,\pi}(s|\mathcal{M})$ (and again, the superscript $T$ will be omitted in subsequent subsections).

For any MDP $\mathcal{M} = (\mathcal{S}, \mathcal{A}, H, P, \mathcal{R})$, let $\mathcal{M}^T$ be an MDP of horizon $2H$ and state space $\mathcal{S}^{A+1}$ that separates the state transition and next-state generation as follows:

1. Assume w.l.o.g. that $\mathcal{M}$ starts at some initial state $s_1$. The extended environment starts at a state $s_1 \times \boldsymbol{s}_0'$, where $\boldsymbol{s}_0' \in \mathcal{S}^A$ is a vector of $A$ copies of some arbitrary state $s_0 \in \mathcal{S}$.

2. For any $h \in [H]$, at timestep $2h - 1$, the environment $\mathcal{M}^T$ transitions from state $s_h \times \boldsymbol{s}_0'$ to $s_h \times \boldsymbol{s}_{h+1}'$, where $\boldsymbol{s}_{h+1}' \sim P_h(s)$ is a vector containing the next state for all actions $a \in \mathcal{A}$; this transition happens regardless of the action that the agent played. At timestep $2h$, given an action $a_h$, the environment transitions from $s_h \times \boldsymbol{s}_{h+1}'$ to $\boldsymbol{s}_{h+1}'(a) \times \boldsymbol{s}_0'$.

3. The rewards at odd steps $2h - 1$ are zero, while the rewards at even steps $2h$ are $R_h(s_h, a_h) \sim \mathcal{R}_h(s_h, a_h)$ of expectation $r_h(s_h, a_h)$.

As before, since the next state is embedded into the extended state space, any state-dependent policy in $\mathcal{M}^T$ is a one-step transition lookahead policy in the original MDP. Also, the policy at even timesteps does not affect either the rewards or transitions, so it does not affect the value in any way. We again couple the two environments to have the exact same randomness, so assuming that the policy at the even steps in $\mathcal{M}^T$ is the same as the policy in $\mathcal{M}$, we trivially get the following relation between the values

$$V_{2h}^\pi(s, \boldsymbol{s}'|\mathcal{M}^T) = \mathbb{E}\left[ \sum_{t=h}^H R_t(s_t, a_t) | s_h = s, s_{h+1}'(s, \cdot) = \boldsymbol{s}', \pi \right] \triangleq V_h^{T,\pi}(s, \boldsymbol{s}'|\mathcal{M}),$$

$$V_{2h-1}^\pi(s, \boldsymbol{s}_0'|\mathcal{M}^T) = \mathbb{E}\left[ \sum_{t=h}^H R_t(s_t, a_t) | s_h = s, \pi \right] = V_h^{T,\pi}(s|\mathcal{M}). \tag{7}$$

While $\mathcal{M}^T$ is finite, it is exponential in size, so applying any standard algorithm in this environment would lead to exponentially-bad performance bounds. Nonetheless, as with the extended-reward environment, we use this representation to prove useful results on one-step transition lookahead.

**Proposition 2.** *The optimal value of one-step transition lookahead agents satisfies*

$$V_{H+1}^{T,*}(s) = 0, \qquad\qquad\qquad \forall s \in \mathcal{S},$$

$$V_h^{T,*}(s) = \mathbb{E}_{\boldsymbol{s}' \sim P_h(s)}\left[\max_{a \in \mathcal{A}}\left\{r_h(s,a) + V_{h+1}^{T,*}(s'(s,a))\right\}\right], \qquad \forall s \in \mathcal{S}, h \in [H].$$

*Also, given next-state observations $\boldsymbol{s}' = \{s'(a)\}_{a \in \mathcal{A}}$ at state $s$ and step $h$, the optimal policy is*

$$\pi_h^*(s, \boldsymbol{s}') \in \arg\max_{a \in \mathcal{A}}\left\{r_h(s,a) + V_{h+1}^{T,*}(s'(a))\right\}.$$

*Proof.* We prove the result in the extended MDP $\mathcal{M}^T$, in which (as with reward lookahead) the optimal value can be calculated using the Bellman equations as follows [Puterman, 2014]

$$V_{2H+1}^T(s, \boldsymbol{s}'|\mathcal{M}^T) = 0, \qquad\qquad\qquad \forall s \in \mathcal{S}, \boldsymbol{s}' \in \mathcal{S}^A,$$

$$V_{2h}^*(s, \boldsymbol{s}'|\mathcal{M}^T) = \max_a\{r_h(s,a) + V_{2h+1}^*(s'(a), \boldsymbol{s}_0'|\mathcal{M}^T)\}, \quad \forall h \in [H], s \in \mathcal{S}, \boldsymbol{s}' \in \mathcal{S}^A,$$

$$V_{2h-1}^*(s, \boldsymbol{s}_0'|\mathcal{M}^T) = \mathbb{E}_{\boldsymbol{s}' \sim P_h(s)}\left[V_{2h}^*(s, \boldsymbol{s}'|\mathcal{M}^T)\right], \qquad\qquad \forall h \in [H], s \in \mathcal{S}. \quad (8)$$

By the equivalence between $\mathcal{M}$ and $\mathcal{M}^T$ for all policies, this is also the optimal value in $\mathcal{M}$. Combining both recursion equations and substituting Equation (7) leads to the stated value calculation for all $h \in [H]$ and $s \in \mathcal{S}$:

$$\begin{aligned}
V_h^{T,*}(s|\mathcal{M}) &= V_{2h-1}^*(s, \boldsymbol{s}_0'|\mathcal{M}^T) \\
&= \mathbb{E}_{\boldsymbol{s}' \sim P_h(s)}\left[V_{2h}^*(s, \boldsymbol{s}_{h+1}'|\mathcal{M}^T)\right] \\
&= \mathbb{E}_{\boldsymbol{s}' \sim P_h(s)}\left[\max_a\left\{r_h(s,a) + V_{2h+1}^*(s_{h+1}'(a), \boldsymbol{s}_0'|\mathcal{M}^T)\right\}\right] \\
&= \mathbb{E}_{\boldsymbol{s}' \sim P_h(s)}\left[\max_a\left\{r_h(s,a) + V_{h+1}^{T,*}(s_{h+1}'(a)|\mathcal{M})\right\}\right].
\end{aligned}$$

In addition, a given state $s$ and next-state observations $\boldsymbol{s}'$, the optimal policy at the even stages of the extended MDP is

$$\pi_{2h}^*(s, \boldsymbol{s}') \in \arg\max_{a \in \mathcal{A}}\{r_h(s,a) + V_{2h+1}^*(s'(a))\},$$

alongside arbitrary actions at odd steps. Playing this policy in the original MDP will lead to the optimal one-step transition lookahead policy, as it achieves the optimal value of the original MDP. By the value relations between the two environments ($V_{2h+1}^*(s, \boldsymbol{s}_0'|\mathcal{M}^T) = V_{h+1}^{T,*}(s|\mathcal{M})$), this is equivalent to the stated policy. $\qquad\square$

**Remark 2.** *As in Remark 1, one could write the dynamic programming equations for any policy $\pi \in \Pi^T$, and not just to the optimal one, namely*

$$V_{2h}^\pi(s, \boldsymbol{s}'|\mathcal{M}^T) = r_h(s, \pi(s, \boldsymbol{s}')) + V_{2h+1}^*(s'(\pi_h(s, \boldsymbol{s}')), \boldsymbol{s}_0'|\mathcal{M}^T), \quad \forall h \in [H], s \in \mathcal{S}, \boldsymbol{s}' \in \mathcal{S}^A,$$

$$V_{2h-1}^\pi(s, \boldsymbol{s}_0'|\mathcal{M}^T) = \mathbb{E}_{\boldsymbol{s}' \sim P_h(s)}\left[V_{2h}^\pi(s, \boldsymbol{s}'|\mathcal{M}^T)\right], \qquad\qquad \forall h \in [H], s \in \mathcal{S}.$$

*In particular, following the notation of Equation (7), we can write*

$$\begin{aligned}
V_h^{T,\pi}(s, \boldsymbol{s}'|\mathcal{M}) &= r_h(s, \pi_h(s, \boldsymbol{s}')) + V_{h+1}^{T,\pi}(s'(\pi_h(s, \boldsymbol{s}'))|\mathcal{M}), \qquad \text{and,} \\
V_h^{T,\pi}(s|\mathcal{M}) &= \mathbb{E}_{\boldsymbol{s}' \sim P_h(s)}\left[V_h^{T,\pi}(s, \boldsymbol{s}'|\mathcal{M})\right] \\
&= \mathbb{E}_{\boldsymbol{s}' \sim P_h(s)}\left[r_h(s, \pi_h(s, \boldsymbol{s}')) + V_{h+1}^{T,\pi}(s'(\pi_h(s, \boldsymbol{s}'))|\mathcal{M})\right],
\end{aligned}$$

*a notation that will be extensively used for transition lookahead.*

We also prove a variation of the law of total variance (LTV) for transition lookahead:

**Lemma 10.** *For any one-step transition lookahead policy $\pi \in \Pi^T$, it holds that*

$$\mathbb{E}\left[\sum_{h=1}^{H} \mathrm{Var}_{\boldsymbol{s'} \sim P_h(s_h)}(V_h^{T,\pi}(s_h, \boldsymbol{s'}))|\pi, s_1\right] \leq \mathbb{E}\left[\left(\sum_{h=1}^{H} r_h(s_h, a_h) - V_1^{T,\pi}(s_1)\right)^2 |\pi, s_1\right].$$

*Proof.* We apply the law of total variance in the extended MDP; there, the expected rewards are either $0$ (at odd steps) or $r_h(s_h, a_h)$ (at even steps), so the total expected rewards are $\sum_{h=1}^{H} r_h(s_h, a_h)$. Hence, by Lemma 27,

$$\mathbb{E}\left[\left(\sum_{h=1}^{H} r_h(s_h, a_h) - V_1^{\pi}(s_1, \boldsymbol{s'_0}|\mathcal{M}^T)\right)^2 |\pi, s_1\right]$$

$$= \mathbb{E}\left[\underbrace{\sum_{h=1}^{H} \mathrm{Var}(V_{2h}^{\pi}(s_h, \boldsymbol{s'_{h+1}}|\mathcal{M}^T)|(s_h, \boldsymbol{s'_0}))}_{\text{Odd steps}} + \underbrace{\sum_{h=1}^{H} \mathrm{Var}(V_{2h+1}^{\pi}(s_{h+1}, \boldsymbol{s'_0}|\mathcal{M}^T)|(s_h, \boldsymbol{s'_{h+1}}))}_{\text{Even steps}} |\pi, s_1\right]$$

$$\geq \mathbb{E}\left[\sum_{h=1}^{H} \mathrm{Var}(V_{2h}^{\pi}(s_h, \boldsymbol{s'_{h+1}}|\mathcal{M}^T)|(s_h, \boldsymbol{s'_0}))|\pi, s_1\right]$$

$$= \mathbb{E}\left[\sum_{h=1}^{H} \mathrm{Var}_{\boldsymbol{s'} \sim P_h(s_h)}(V_{2h}^{\pi}(s_h, \boldsymbol{s'}|\mathcal{M}^T))|\pi, s_1\right]$$

$$= \mathbb{E}\left[\sum_{h=1}^{H} \mathrm{Var}_{\boldsymbol{s'} \sim P_h(s_h)}(V_h^{T,\pi}(s_h, \boldsymbol{s'}|\mathcal{M}))|\pi, s_1\right].$$

Using again the identity $V_1^{\pi}(s_1, \boldsymbol{s'_0}|\mathcal{M}^T) = V_1^{T,\pi}(s_1|\mathcal{M})$ leads to the desired result. $\square$

Finally, prove a value-difference lemma also for transition lookahead

**Lemma 11** (Value-Difference Lemma with Transition Lookahead). *Let $\mathcal{M}_1 = (\mathcal{S}, \mathcal{A}, H, P^1, \mathcal{R}^1)$ and $\mathcal{M}_2 = (\mathcal{S}, \mathcal{A}, H, P^2, \mathcal{R}^2)$ be two environments. For any deterministic one-step transition lookahead policy $\pi \in \Pi^T$, any $h \in [H]$ and $s \in \mathcal{S}$, it holds that*

$$V_h^{T,\pi}(s|\mathcal{M}_1) - V_h^{T,\pi}(s|\mathcal{M}_2)$$
$$= \mathbb{E}_{\mathcal{M}_1}\left[r_h^1(s_h, \pi_h(s_h, \boldsymbol{s'_{h+1}})) - r_h^2(s_h, \pi_h(s_h, \boldsymbol{s'_{h+1}}))|s_h = s\right]$$
$$+ \mathbb{E}_{\mathcal{M}_1}\left[V_{h+1}^{T,\pi}(s_{h+1}|\mathcal{M}_1) - V_{h+1}^{T,\pi}(s_{h+1}|\mathcal{M}_2)|s_h = s\right]$$
$$+ \mathbb{E}_{\mathcal{M}_1}\left[\mathbb{E}_{\boldsymbol{s'} \sim P_h^1(s_h)}\left[V_h^{T,\pi}(s_h, \boldsymbol{s'}|\mathcal{M}_2)\right] - \mathbb{E}_{\boldsymbol{s'} \sim P_h^2(s_h)}\left[V_h^{T,\pi}(s_h, \boldsymbol{s'}|\mathcal{M}_2)\right]|s_h = s\right].$$

*where $V_h^{T,\pi}(s, \boldsymbol{s'}|\mathcal{M})$ is the value at a state given the reward realization, defined in Equation (7) and given in Remark 2.*

*Proof.* We again work with the extended MDPs $\mathcal{M}_1^T, \mathcal{M}_2^T$ and use their Bellman equations, namely,

$$V_{2h}^{\pi}(s, \boldsymbol{s'}|\mathcal{M}^T) = r_h(s, \pi(s, \boldsymbol{s'})) + V_{2h+1}^{*}(s'(\pi_h(s, \boldsymbol{s'})), \boldsymbol{s'_0}|\mathcal{M}^T), \quad \forall h \in [H], s \in \mathcal{S}, \boldsymbol{s'} \in \mathcal{S}^A,$$
$$V_{2h-1}^{\pi}(s, \boldsymbol{s'_0}|\mathcal{M}^T) = \mathbb{E}_{\boldsymbol{s'} \sim P_h(s)}\left[V_{2h}^{\pi}(s, \boldsymbol{s'}|\mathcal{M}^T)\right], \qquad\qquad \forall h \in [H], s \in \mathcal{S}.$$

Using the relation between the value of the original and extended MDP (eq. (7)) and the Bellman equations of the extended MDP, for any $h \in [H]$, we have

$$V_h^{T,\pi}(s|\mathcal{M}_1) - V_h^{T,\pi}(s|\mathcal{M}_2)$$
$$= V_{2h-1}^\pi(s, \boldsymbol{s}_0'|\mathcal{M}_1^T) - V_{2h-1}^\pi(s, \boldsymbol{s}_0'|\mathcal{M}_2^T)$$
$$= \mathbb{E}_{\boldsymbol{s}' \sim P_h^1(s)}\left[V_{2h}^\pi(s, \boldsymbol{s}'|\mathcal{M}_1^T)\right] - \mathbb{E}_{\boldsymbol{s}' \sim P_h^2(s)}\left[V_{2h}^\pi(s, \boldsymbol{s}'|\mathcal{M}_2^T)\right]$$
$$= \mathbb{E}_{\boldsymbol{s}' \sim P_h^1(s)}\left[V_{2h}^\pi(s, \boldsymbol{s}'|\mathcal{M}_1^T) - V_{2h}^\pi(s, \boldsymbol{s}'|\mathcal{M}_2^T)\right] + \mathbb{E}_{\boldsymbol{s}' \sim P_h^1(s)}\left[V_{2h}^\pi(s, \boldsymbol{s}'|\mathcal{M}_2^T)\right] - \mathbb{E}_{\boldsymbol{s}' \sim P_h^2(s)}\left[V_{2h}^\pi(s, \boldsymbol{s}'|\mathcal{M}_2^T)\right]$$
$$= \mathbb{E}_{\boldsymbol{s}' \sim P_h^1(s)}\left[V_{2h}^\pi(s, \boldsymbol{s}'|\mathcal{M}_1^T) - V_{2h}^\pi(s, \boldsymbol{s}'|\mathcal{M}_2^T)\right] + \mathbb{E}_{\boldsymbol{s}' \sim P_h^1(s)}\left[V_h^{T,\pi}(s, \boldsymbol{s}'|\mathcal{M}_2)\right] - \mathbb{E}_{\boldsymbol{s}' \sim P_h^2(s)}\left[V_h^{T,\pi}(s, \boldsymbol{s}'|\mathcal{M}_2)\right]$$
$$= \mathbb{E}_{\mathcal{M}_1}\left[V_{2h}^\pi(s_h, \boldsymbol{s}_{h+1}'|\mathcal{M}_1^T) - V_{2h}^\pi(s_h, \boldsymbol{s}_{h+1}'|\mathcal{M}_2^T)|s_h = s\right]$$
$$+ \mathbb{E}_{\boldsymbol{s}' \sim P_h^1(s)}\left[V_h^{T,\pi}(s, \boldsymbol{s}'|\mathcal{M}_2)\right] - \mathbb{E}_{\boldsymbol{s}' \sim P_h^2(s)}\left[V_h^{T,\pi}(s, \boldsymbol{s}'|\mathcal{M}_2)\right]. \tag{9}$$

Denoting $a_h = \pi_h(s_h, \boldsymbol{s}_{h+1}')$ the action taken by the agent at environment $\mathcal{M}_1$, We have

$$V_{2h}^\pi(s_h, \boldsymbol{s}_{h+1}'|\mathcal{M}_1^T) - V_{2h}^\pi(s_h, \boldsymbol{s}_{h+1}'|\mathcal{M}_2^T)$$
$$= \left(r_h^1(s_h, a_h) + V_{2h+1}^\pi(s_{h+1}'(a_h), \boldsymbol{s}_0'|\mathcal{M}_1^T)\right) - \left(r_h^2(s_h, a_h) + V_{2h+1}^\pi(s_{h+1}'(a_h), \boldsymbol{s}_0'|\mathcal{M}_2^T)\right)$$
$$= r_h^1(s_h, a_h) - r_h^2(s_h, a_h) + V_{h+1}^{T,\pi}(s_{h+1}'(a_h)|\mathcal{M}_1) - V_{h+1}^{T,\pi}(s_{h+1}'(a_h)|\mathcal{M}_2),$$

when taking the expectation w.r.t. $\mathcal{M}_1$, it holds that $s_{h+1}'(a_h) = s_{h+1}$; substituting this back into Equation (9), we get

$$V_h^\pi(s|\mathcal{M}_1) - V_h^\pi(s|\mathcal{M}_2)$$
$$= \mathbb{E}_{\mathcal{M}_1}\left[r_h^1(s_h, a_h) - r_h^2(s_h, a_h) + V_{h+1}^{T,\pi}(s_{h+1}'(a_h)|\mathcal{M}_1) - V_{h+1}^{T,\pi}(s_{h+1}'(a_h)|\mathcal{M}_2)|s_h = s\right]$$
$$+ \mathbb{E}_{\boldsymbol{s}' \sim P_h^1(s)}\left[V_h^{T,\pi}(s, \boldsymbol{s}'|\mathcal{M}_2)\right] - \mathbb{E}_{\boldsymbol{s}' \sim P_h^2(s)}\left[V_h^{T,\pi}(s, \boldsymbol{s}'|\mathcal{M}_2)\right]$$
$$= \mathbb{E}_{\mathcal{M}_1}\left[r_h^1(s_h, \pi_h(s_h, \boldsymbol{s}_{h+1}')) - r_h^2(s_h, \pi_h(s_h, \boldsymbol{s}_{h+1}'))|s_h = s\right]$$
$$+ \mathbb{E}_{\mathcal{M}_1}\left[V_{h+1}^{T,\pi}(s_{h+1}|\mathcal{M}_1) - V_{h+1}^{T,\pi}(s_{h+1}|\mathcal{M}_2)|s_h = s\right]$$
$$+ \mathbb{E}_{\mathcal{M}_1}\left[\mathbb{E}_{\boldsymbol{s}' \sim P_h^1(s_h)}\left[V_h^{T,\pi}(s_h, \boldsymbol{s}'|\mathcal{M}_2)\right] - \mathbb{E}_{\boldsymbol{s}' \sim P_h^2(s_h)}\left[V_h^{T,\pi}(s_h, \boldsymbol{s}'|\mathcal{M}_2)\right]|s_h = s\right].$$

$\square$

## C.3 Full Algorithm Description for Transition Lookahead

---

**Algorithm 4** Monotonic Value Propagation with Transition Lookahead (MVP-TL)

---

1: **Require:** $\delta \in (0,1)$, bonuses $b^r_{k,h}(s,a), b^p_{k,h}(s)$
2: **for** $k = 1, 2, ...$ **do**
3:     Initialize $\bar{V}^k_{H+1}(s) = 0$
4:     **for** $h = H, H-1, .., 1$ **do**
5:         **for** $s \in \mathcal{S}$ **do**
6:             **if** $n^{k-1}_h(s) = 0$ **then**
7:                 $\bar{V}^k_h(s) = H$
8:             **else**
9:                 Calculate the truncated values

$$\bar{V}^k_h(s) = \min\left\{ \frac{1}{n^{k-1}_h(s)} \sum_{t=1}^{n^{k-1}_h(s)} \max_{a \in \mathcal{A}}\left\{ \hat{r}^{k-1}_h(s,a) + b^r_{k,h}(s,a) + \bar{V}^k_{h+1}(s'^{k^t_h(s)}_{h+1}(s,a)) \right\} + b^p_{k,h}(s), H \right\}$$

10:             **end if**
11:         For any set of next-states $\boldsymbol{s}' \in \mathcal{S}^A$, define the policy $\pi^k$

$$\pi^k_h(s, \boldsymbol{s}') \in \arg\max_{a \in \mathcal{A}}\left\{ \hat{r}^{k-1}_h(s,a) + b^r_{k,h}(s,a) + \bar{V}^k_{h+1}(s'(a)) \right\}$$

12:         **end for**
13:     **end for**
14:     **for** $h = 1, 2, \ldots H$ **do**
15:         Observe $s^k_h$ and $\boldsymbol{s}'^k_{h+1} = \left\{ s'^k_{h+1}(s^k_h, a) \right\}_{a \in \mathcal{A}}$
16:         Play an action $a^k_h = \pi^k_h(s^k_h, \boldsymbol{s}'^k_h)$
17:         Collect the reward $R^k_h \sim \mathcal{R}_h(s^k_h, a^k_h)$ and transition to the next state $s^k_{h+1} = s'^k_{h+1}(s^k_h, a^k_h)$
18:     **end for**
19:     Update the empirical estimators and counts for all visited state-actions
20: **end for**

---

As with reward lookahead, we again use a variant of the MVP algorithm [Zhang et al., 2021b], described in Algorithm 4. For the bonuses, we use the notation

$$\bar{V}^k_h(s, \boldsymbol{s}') = \max_{a \in \mathcal{A}}\left\{ \hat{r}^{k-1}_h(s,a) + b^r_{k,h}(s,a) + \bar{V}^k_{h+1}(s'(a)) \right\}$$

and define the following bonuses:

$$b^r_{k,h}(s,a) = \min\left\{ \sqrt{\frac{L^k_\delta}{n^{k-1}_h(s,a) \vee 1}}, 1 \right\},$$

$$b^p_{k,h}(s) = \frac{20}{3}\sqrt{\frac{\mathrm{Var}_{\boldsymbol{s}' \sim \hat{P}^{k-1}_h(s)}(\bar{V}^k_h(s, \boldsymbol{s}'))L^k_\delta}{n^{k-1}_h(s) \vee 1}} + \frac{400}{3}\frac{HL^k_\delta}{n^{k-1}_h(s) \vee 1},$$

where $L^k_\delta = \ln\frac{16S^3 A^2 Hk^2(k+1)}{\delta}$ and

$$\mathrm{Var}_{\boldsymbol{s}' \sim \hat{P}^{k-1}_h(s)}(\bar{V}^k_h(s, \boldsymbol{s}')) = \mathbb{E}_{\boldsymbol{s}' \sim \hat{P}^{k-1}_h(s)}\left[ \bar{V}^k_h(s, \boldsymbol{s}')^2 \right] - \left( \mathbb{E}_{\boldsymbol{s}' \sim \hat{P}^{k-1}_h(s)}\left[ \bar{V}^k_h(s, \boldsymbol{s}') \right] \right)^2.$$

The notation $k^t_h(s)$ again represents the $t^{th}$ episode where the state $s$ was visited at the $h^{th}$ timestep; in particular, line 9 of the algorithm is the expectation w.r.t. the empirical reward distribution $\hat{P}^{k-1}_h(s)$. Since the transition bonus is larger than $H$ when $n^{k-1}_h(s) = 0$, we can arbitrarily define the expectation w.r.t. $\hat{P}^{k-1}_h(s)$ when $n^{k-1}_h(s) = 0$ to be 0, and one could write the update in a more concise way as

$$\bar{V}^k_h(s) = \min\left\{ \mathbb{E}_{\boldsymbol{s}' \sim \hat{P}^{k-1}_h(s)}\left[ \bar{V}^k_h(s, \boldsymbol{s}') \right] + b^p_{k,h}(s), H \right\}.$$

## C.4 Additional Notations and List Representation

In this subsection, we present additional notations for both values and transition distributions that will be helpful in the analysis. In particular, we show that instead of looking at the distribution over all combinations of next state $s' \in \mathcal{S}^A$, we can look at a ranking of all the next-state-actions and represent important quantities using the effective distribution on these ranks – this moves the problem from being $S^A$-dimensional to a dimension of $SA$.

We start by defining the values starting from state $s \in \mathcal{S}$, playing $a \in \mathcal{A}$ and transitioning to $s' \in \mathcal{S}$, denoted by

$$V_h^\pi(s, s', a) = r_h(s, a) + V_{h+1}^\pi(s'),$$
$$V_h^*(s, s', a) = r_h(s, a) + V_{h+1}^*(s'),$$
$$\bar{V}_h^k(s, s', a) = \hat{r}_h^{k-1}(s, a) + b_{k,h}^r(s, a) + \bar{V}_{h+1}^k(s'),$$

We similarly define (consistently with Remark 2)

$$V_h^\pi(s, \boldsymbol{s}') = V_h^\pi(s, s'(\pi_h(s, \boldsymbol{s}')), \pi_h(s, \boldsymbol{s}')),$$
$$V_h^*(s, \boldsymbol{s}') = \max_a V_h^*(s, s'(a), a), \qquad \text{and },$$
$$\bar{V}_h^k(s, \boldsymbol{s}') = \max_a \bar{V}_h^k(s, s'(a), a).$$

**List representation.** We now move to defining lists of next-state-actions and distributions with respect to such lists. Let $\ell$ be a list that orders all next-state-action pairs from $(s'_{\ell(1)}, a_{\ell(1)})$ to $(s'_{\ell(SA)}, a_{\ell(SA)})$ and define the set of all possible lists to be $\mathcal{L}$ (with $|\mathcal{L}| = (SA)!$). Also, define $\ell^u$, the list induced by a function $u : \mathcal{S} \times \mathcal{A} \mapsto \mathbb{R}$ such that $u(s'_{\ell^u(1)}, a_{\ell^u(1)}) \geq \cdots \geq u(s'_{\ell^u(SA)}, a_{\ell^u(SA)})$, where ties are broken in any fixed arbitrary way. From this point forward, for brevity and when clear from the context, we omit the list from the indexing, e.g., write the list $\ell$ by $(s'_1, a_1), \ldots, (s'_{SA}, a_{SA})$.

We now define the probability of list elements. Denote by $E_i^\ell$ the event that the highest-ranked realized element in the list is element $i$, namely

$$E_i^\ell = \left\{ \boldsymbol{s}' \in \mathcal{S}^A : s'(a_i) = s'_i \text{ and } \forall j < i, s'(a_j) \neq s'_j \right\}. \tag{10}$$

Then, for a probability measure $P$ on $\mathcal{S}^A$, define $\mu(i|\ell, P) = P(\boldsymbol{s}' \in E_i^\ell)$. Notably, when the list is induced by $u$ and element $i$ is the realized highest-ranked elements, we can write $\max_a u(s'(a), a) = u(s'_i, a_i)$, so we have that (e.g. by Lemma 17 with $f(\boldsymbol{s}') = \max_a u(s'(a), a)$)

$$\mathbb{E}_{\boldsymbol{s}' \sim P_h(s)} \left[ \max_a \{u(s'(a), a)\} \right] = \mathbb{E}_{i \sim \mu(\cdot|\ell, P_h(s))}[u(s'_i, a_i)]$$

We also denote by $\hat{\mu}_h^k(i|s; \ell) = \frac{1}{n_h^k(s) \vee 1} \sum_{t=1}^K \mathbb{1}\{s_h^t = s, \boldsymbol{s}_{h+1}'^t \in E_i^\ell\}$, the empirical probability for a list location $i$ to be the highest-realized ranking according to a list $\ell$ at state $s$ and step $h$, based on samples up to episode $k$; We have by Lemma 17 that $\hat{\mu}_h^k(i|s; \ell) = \hat{P}_h^k(E_i^\ell|s)$ and

$$\mathbb{E}_{\boldsymbol{s}' \sim \hat{P}_h^{k-1}(s)} \left[ \max_a \{u(s'(a), a)\} \right] = \mathbb{E}_{i \sim \hat{\mu}_h^{k-1}(\cdot|s; \ell^u)}[u(s'_i, a_i)].$$

Similarly, we will require the distribution probability w.r.t. two lists – the probability that the top element w.r.t. list $\ell$ is $i$ and the top element w.r.t. list $\ell'$ is $j$; we denote the real and empirical probability distributions by $\mu(i, j|\ell, \ell', P)$ and $\hat{\mu}_h^k(i, j|s; \ell, \ell')$, respectively. This allows, for example, using Lemma 17 to write for any $u, v : \mathcal{S} \times \mathcal{A} \mapsto \mathbb{R}$,

$$\mathbb{E}_{\boldsymbol{s}' \sim P_h(s)} \left[ \max_a \{u(s'(a), a)\} - \max_a \{v(s'(a), a)\} \right]$$
$$= \mathbb{E}_{i,j \sim \mu(\cdot|\ell^u, \ell^v, P_h(s))} \left[ u(s'_{\ell^u(i)}, a_{\ell^u(i)}) - v(s'_{\ell^v(j)}, a_{\ell^v(j)}) \right],$$
$$\mathbb{E}_{\boldsymbol{s}' \sim \hat{P}_h^{k-1}(s)} \left[ \max_a \{u(s'(a), a)\} - \max_a \{v(s'(a), a)\} \right]$$
$$= \mathbb{E}_{i,j \sim \hat{\mu}_h^k(\cdot|s; \ell^u, \ell^v)} \left[ u(s'_{\ell^u(i)}, a_{\ell^u(i)}) - v(s'_{\ell^v(j)}, a_{\ell^v(j)}) \right]. \tag{11}$$

Finally, we say that a policy $\pi_h(s, \boldsymbol{s}')$ is induced by lists $\ell_h(s)$ if it chooses an action $a$ such that its next-state $s'(a)$ is ranked higher in $\ell$ than all other realized next-state-action pairs. In particular, the policy $\pi^k$ and the optimal policy $\pi^*$ (defined in Proposition 2) are such policies w.r.t. the lists $\bar{\ell}_h^k(s)$ and $\ell_h^*(s)$ – induced by $\bar{V}_h^k(s, s', a)$ and $V_h^*(s, s', a)$, respectively. As such, for any probability measure $P_h(s)$, function $u : \mathcal{S} \times \mathcal{S} \times \mathcal{A} \mapsto \mathbb{R}$ and a policy $\pi$ induced by a list $\ell$, it holds that

$$\mathbb{E}_{\boldsymbol{s}' \sim P_h(s)}[u(s, s'(\pi(a)), \pi(a))] = \mathbb{E}_{i \sim \mu(\cdot | \ell_h(s), P_h(s))}[u(s, s_i', a_i)]. \tag{12}$$

### C.4.1 Planning with Transition Lookahead

We have already seen the optimal policy is induced by a list $\ell_h^*(s)$, and in particular, we can write the dynamic programming equations of Proposition 2 as

$$V_h^*(s) = \mathbb{E}_{\boldsymbol{s}' \sim P_h(s)}\left[\max_{a \in \mathcal{A}}\left\{r_h(s, a) + V_h^{T,*}(s'(a))\right\}\right]$$

$$= \mathbb{E}_{i \sim \mu(\cdot | \ell_h^*(s), P_h(s))}\left[r_h(s, a_i) + V_{h+1}^*(s'(a_i))\right].$$

Therefore, one way to perform the planning is to build a list $\ell_h^*(s)$ of $(s', a)$ s.t. the values

$$V_h^*(s, s', a) = r_h(s, a) + V_{h+1}^*(s')$$

are sorted in a non-increasing order and calculate the probability of any pair in the list to be the highest-realized pair:

$$\mu(i | \ell, P_h(s)) = P_h(E_i^\ell) = \Pr\big(s_{h+1}'(a_i) = s_i' \text{ and } \forall j < i, s_{h+1}'(a_j) \neq s_j' | s_h = s\big).$$

In general, calculating this distribution is intractable, and one must resort to approximating it by sampling (as done in Algorithm 4. Nonetheless, if next states are generated independently between actions, this distribution could be efficiently calculated as follows:

$$\mu(i | \ell, P_h(s)) = \Pr\big(s_{h+1}'(a_i) = s_i' \text{ and } \forall j < i, s_{h+1}'(a_j) \neq s_j' | s_h = s\big)$$

$$\stackrel{(1)}{=} \Pr\big\{s'(a_i) = s_i' \text{ and } \forall j < i \text{ s.t. } a_j \neq a_i, s'(a_j) \neq s_j' | s_h = s\big\}$$

$$\stackrel{(2)}{=} \Pr\{s'(a_i) = s_i' | s_h = s\} \prod_{a \neq a_i} \Pr\big\{\forall j < i \text{ s.t. } a_j = a, s'(a) \neq s_j' | s_h = s\big\}$$

$$\stackrel{(3)}{=} P_h(s_i' | s, a_i) \prod_{a \neq a_i}\left(1 - \sum_{j=1}^{i-1} \mathbb{1}\{a_j = a\} P_h(s_j' | s, a)\right).$$

Relation (1) holds since if $s'(a_i) = s_i'$, it cannot get any previous value of the same action in the list, so these events can be removed. Relation (2) is by the independence and (3) directly calculates the probabilities.

## C.5 The First Good Event – Concentration

Next, we define the events that ensure the concentration of all empirical measures. For rewards, an event handles the convergence of the empirical rewards to their mean. For the transitions, we want the Bellman operator, applied on the optimal value with the empirical model, to concentrate well, and we require the variance of values w.r.t. the empirical and real model to be close. Finally, the empirical measure $\hat{\mu}_h^k(i,j|s;\ell,\ell_h^*(s))$ must concentrate well around its mean for any list $\ell$ – this will allow the change-of-measure argument described in the proof sketch.

Formally, define the following good events:

$$E^r(k) = \left\{ \forall s,a,h : |r_h(s,a) - \hat{r}_h^{k-1}(s,a)| \le \sqrt{\frac{L_\delta^k}{n_h^{k-1}(s,a) \vee 1}} \right\}$$

$$E^\ell(k) = \left\{ \forall s,h, \forall \ell \in \mathcal{L}, \forall i,j \in [SA] : \left| \hat{\mu}_h^{k-1}(i,j|s;\ell,\ell_h^*(s)) - \mu(i,j|\ell,\ell_h^*(s);P_h(s)) \right| \right.$$
$$\left. \le \sqrt{\frac{4SAL_\delta^k \mu(i,j|s;\ell,\ell_h^*(s);P_h(s))}{n_h^{k-1}(s) \vee 1}} + \frac{2SAL_\delta^k}{n_h^{k-1}(s) \vee 1} \right\}$$

$$E^{pv1}(k) = \left\{ \forall s,h : \left| \mathbb{E}_{s' \sim P_h(s)}\left[V_h^*(s,s')\right] - \mathbb{E}_{s' \sim \hat{P}_h^{k-1}(s)}\left[V_h^*(s,s')\right] \right| \le \sqrt{\frac{2\mathrm{Var}_{s' \sim P_h(s)}(V_h^*(s,s'))L_\delta^k}{n_h^{k-1}(s) \vee 1}} + \frac{HL_\delta^k}{n_h^{k-1}(s) \vee 1} \right\}$$

$$E^{pv2}(k) = \left\{ \forall s,h : \left| \sqrt{\mathrm{Var}_{s' \sim P_h(s)}(V_h^*(s,s'))} - \sqrt{\mathrm{Var}_{s' \sim \hat{P}_h^{k-1}(s)}(V_h^*(s,s'))} \right| \le 4H\sqrt{\frac{L_\delta^k}{n_h^{k-1}(s) \vee 1}} \right\}$$

where we again use $L_\delta^k = \ln\frac{16S^3 A^2 H k^2(k+1)}{\delta}$. We define the first good event as

$$\mathbb{G}_1 = \bigcap_{k \ge 1} E^r(k) \bigcap_{k \ge 1} E^\ell(k) \bigcap_{k \ge 1} E^{pv1}(k) \bigcap_{k \ge 1} E^{pv2}(k),$$

for which the following holds:

**Lemma 12** (The First Good Event)**.** *It holds that* $\Pr(\mathbb{G}_1) \ge 1 - \delta/2$.

*Proof.* We prove that each of the events holds w.p. at least $1 - \delta/8$. The result then directly follows by the union bound. We also remark that due to the domain of the variables and their estimators (e.g., $[0,1]$ for the rewards), all bounds trivially hold when the counts equal zero, so w.l.o.g., we only prove the results for cases in which states/state-actions were already previously visited.

**Event** $\cap_{k \ge 1} E^r(k)$**.** Fix $k \ge 1, s, a, h$ and visits $n \ge 1$. Given all of these, the reward observations are i.i.d. random variables supported by $[0,1]$. Denoting the empirical mean based on these $n$ samples by $\hat{r}_h(s,a,n)$, by Hoeffding's inequality, it holds w.p. $1 - \frac{\delta}{8SAHk^2(k+1)}$ that

$$|r_h(s,a) - \hat{r}_h(s,a,n)| \le \sqrt{\frac{\ln\frac{16SAHk^2(k+1)}{\delta}}{2n}} \le \sqrt{\frac{L_\delta^k}{n}}.$$

Taking the union bound over all $n \in [k]$ at timestep $k$, we get that w.p. $1 - \frac{\delta}{8SAHk(k+1)}$

$$|r_h(s,a) - \hat{r}_h^{k-1}(s,a)| \le \sqrt{\frac{L_\delta^k}{n_h^{k-1}(s,a) \vee 1}},$$

and another union bound over all possible values of $s, a, h$ and $k \ge 1$ implies that $\cap_{k \ge 1} E^r(k)$ holds w.p. at least $1 - \delta/8$.

**The event** $\cap_{k \ge 1} E^\ell(k)$**.** For any fixed $k \ge 1, s, h$, a list $\ell \in \mathcal{L}$ and number of visits $n \in [k]$, we utilize Lemma 16 (event $E^p$) w.r.t. the distribution $\mu(i,j|\ell,\ell_h^*(s),P)$ (whose support is of size $M = (SA)^2$). When applying the lemma, notice that given the number of visits $n \ge 1$, the empirical distribution $\hat{\mu}_h^{k-1}(i,j|s;\ell,\ell_h^*(s))$ is the average of $n = n_h^{k-1}(s)$ i.i.d samples, so that for all $i,j \in [SA]$,

$$\left| \hat{\mu}_h^{k-1}(i,j|s;\ell,\ell_h^*(s)) - \mu(i,j|\ell,\ell_h^*(s);P_h(s)) \right| \le \sqrt{\frac{2\mu(i,j|\ell,\ell_h^*(s);P_h(s))\ln\frac{2(SA)^2}{\delta'}}{n}} + \frac{2\ln\frac{2(SA)^2}{\delta'}}{3n}$$

$$\le \sqrt{\frac{4\mu(i,j|\ell,\ell_h^*(s);P_h(s))\ln\frac{2SA}{\delta'}}{n}} + \frac{2\ln\frac{2SA}{\delta'}}{n}$$

w.p. $1 - \delta'$. Choosing $\delta' = \frac{\delta}{8|\mathcal{L}|SHk^2(k+1)}$ (such that $\ln \frac{2SA}{\delta'} \leq SA \ln \frac{16S^3A^2Hk^2(k+1)}{\delta}$ since $|\mathcal{L}| \leq (SA)^{SA}$), while taking the union bound on all $n \in [k]$, all $s, h$ and all lists $\ell \in \mathcal{L}$ implies that $\cap_{k \geq 1} E^\ell(k)$ holds w.p. at least $1 - \frac{\delta}{8}$.

**Events $\cap_{k \geq 1} E^{pv1}(k)$ and $\cap_{k \geq 1} E^{pv2}(k)$.** We repeat the arguments stated in Lemma 5. For any fixed $k \geq 1, s, h$ and number of visits $n \in [k]$, we utilize Lemma 16 w.r.t. the next-state distribution for all actions $P_h(s)$, the value $V_h^*(s, s') \in [0, H]$ and probability $\delta' = \frac{\delta}{8SHk^2(k+1)}$; we yet again remind that given the number of visits, samples are i.i.d.

As before, the events $\cap_{k \geq 1} E^{pv1}(k)$ and $\cap_{k \geq 1} E^{pv2}(k)$ hold w.p. at least $1 - \frac{\delta}{8}$ through the union bound first on $n \in [k]$ (to get the empirical quantities) and then on $s, h$ and $k \geq 1$. This proves that each of the events in $\mathbb{G}_1$ holds w.p. at least $1 - \frac{\delta}{8}$, so $\mathbb{G}_1$ holds w.p. at least $1 - \frac{\delta}{2}$. $\qquad\square$

## C.6 Optimism of the Upper Confidence Value Functions

We now prove that under the event $\mathbb{G}_1$, the values that MVP-TL outputs are optimistic.

**Lemma 13** (Optimism). *Under the first good event $\mathbb{G}_1$, for all $k \in [K]$, $h \in [H]$, $a \in \mathcal{A}$ and $s, s' \in \mathcal{S}$, it holds that $V_h^*(s, s', a) \leq \bar{V}_h^k(s, s', a)$. Moreover, for all $s' \in \mathcal{S}^A$, $V_h^*(s, s') \leq \bar{V}_h^k(s, s')$ and also $V_h^*(s) \leq \bar{V}_h^k(s)$.*

*Proof.* The proof of all claims follows by backward induction on $H$; the base case naturally holds for $h = H + 1$, where all values are defined to be zero.

Assume by induction that for some $k \in [K]$ and $h \in [H]$, the inequality $V_{h+1}^*(s) \leq \bar{V}_{h+1}^k(s)$ holds for all $s \in \mathcal{S}$; we will show that this implies that all stated inequalities also hold at timestep $h$. At this point, we also assume w.l.o.g. that $\bar{V}_h^k(s) < H$ (namely, not truncated), since otherwise, by the boundedness of the rewards, $V_h^*(s) \leq H = \bar{V}_h^k(s)$. In particular, under the good event $E^r(k)$, for all $s$ and $a$, it holds that $\hat{r}_h^{k-1}(s, a) + b_{k,h}^r(s, a) \geq r_h(s, a)$, so for all $s, a$ and $s'$, we have

$$\bar{V}_h^k(s, s', a) = \hat{r}_h^{k-1}(s, a) + b_{k,h}^r(s, a) + \bar{V}_{h+1}^k(s') \geq r_h(s, a) + V_{h+1}^*(s') = V_h^*(s, s', a).$$

where the inequality also uses the induction hypothesis. This proves the first part of the lemma. Moreover, it implies that

$$\bar{V}_h^k(s, s') = \max_{a \in \mathcal{A}}\{\bar{V}_h^k(s, s'(a), a)\} \geq \max_{a \in \mathcal{A}}\{V_h^*(s, s'(a), a)\} = V_h^*(s, s'), \tag{13}$$

and proves the second part of the statement.

To prove the last claim of the lemma, we use the monotonicity of the bonus, relying on Lemma 23. This lemma can be used when applied to the empirical distribution of all possible next-states $\hat{P}_h^{k-1}(s)$; indeed, the non-truncated optimistic value can be written as

$$\bar{V}_h^k(s) = \mathbb{E}_{s' \sim \hat{P}_h^{k-1}(s)}\left[\max_{a \in \mathcal{A}}\{\hat{r}_h^{k-1}(s, a) + b_{k,h}^r(s, a) + \bar{V}_{h+1}^k(s'(a))\}\right] + b_{k,h}^p(s)$$

$$\geq \mathbb{E}_{s' \sim \hat{P}_h^{k-1}(s)}\left[\bar{V}_h^k(s, s')\right] + \max\left\{\frac{20}{3}\sqrt{\frac{\mathrm{Var}_{s' \sim \hat{P}_h^{k-1}(s)}(\bar{V}_h^k(s, s'))L_\delta^k}{n_h^{k-1}(s) \vee 1}}, \frac{400}{9}\frac{3HL_\delta^k}{n_h^{k-1}(s) \vee 1}\right\},$$

which is exactly the required form in Lemma 23, w.r.t. the distribution $\hat{P}_h^{k-1}(s)$ and the values $\bar{V}_h^k(s, s')$ (while noticing that due to the truncation of the values and bonuses, $\bar{V}_h^k(s, s') \in [0, 3H]$). Thus, the lemma guarantees monotonicity in the value, so by Equation (13),

$$\bar{V}_h^k(s) \geq \mathbb{E}_{s' \sim \hat{P}_h^{k-1}(s)}\left[V_h^*(s, s')\right] + \max\left\{\frac{20}{3}\sqrt{\frac{\mathrm{Var}_{s' \sim \hat{P}_h^{k-1}(s)}(V_h^*(s, s'))L_\delta^k}{n_h^{k-1}(s) \vee 1}}, \frac{400}{9}\frac{3HL_\delta^k}{n_h^{k-1}(s) \vee 1}\right\}$$

$$\geq \mathbb{E}_{s' \sim \hat{P}_h^{k-1}(s)}\left[V_h^*(s, s')\right] + \frac{10}{3}\sqrt{\frac{\mathrm{Var}_{s' \sim \hat{P}_h^{k-1}(s)}(V_h^*(s, s'))L_\delta^k}{n_h^{k-1}(s) \vee 1}} + \frac{200}{3}\frac{HL_\delta^k}{n_h^{k-1}(s) \vee 1}$$

$$\geq \mathbb{E}_{s' \sim \hat{P}_h^{k-1}(s)}\left[V_h^*(s, s')\right] + \frac{10}{3}\sqrt{\frac{\mathrm{Var}_{s' \sim P_h(s)}(V_h^*(s, s'))L_\delta^k}{n_h^{k-1}(s) \vee 1}} + \frac{50HL_\delta^k}{n_h^{k-1}(s) \vee 1} \quad (\text{Under } E^{pv2}(k))$$

$$\geq \mathbb{E}_{s' \sim P_h(s)}\left[V_h^*(s, s')\right] \quad (\text{Under } E^{pv1}(k))$$

$$= V_h^*(s).$$

$\square$

## C.7 The Second Good Event – Martingale Concentration

In this subsection, we present three good events that allow replacing the expectation over the randomizations inside each episode by their realization. Let

$$
\begin{aligned}
Y_{1,h}^k &:= \bar{V}_{h+1}^k(s_{h+1}^k) - V_{h+1}^{\pi^k}(s_{h+1}^k) \\
Y_{2,h}^k &= \mathrm{Var}_{\boldsymbol{s}' \sim P_h(s_h^k)}(V_h^{\pi^k}(s_h^k, \boldsymbol{s}')) \\
Y_{3,h}^k &= b_{k,h}^r(s_h^k, a_h^k).
\end{aligned}
$$

The second good event is the intersection of the events $\mathbb{G}_2 = E^{\mathrm{diff}} \cap E^{\mathrm{Var}} \cap E^{br}$ defined as follows.

$$
E^{\mathrm{diff}} = \left\{ \forall h \in [H], K \geq 1 : \sum_{k=1}^K \mathbb{E}[Y_{1,h}^k | F_{k,h-1}] \leq \left(1 + \frac{1}{2H}\right) \sum_{k=1}^K Y_{1,h}^k + 18H^2 \ln \frac{6HK(K+1)}{\delta} \right\},
$$

$$
E^{\mathrm{Var}} = \left\{ K \geq 1 : \sum_{k=1}^K \sum_{h=1}^H Y_{2,h}^k \leq 2 \sum_{k=1}^K \sum_{h=1}^H \mathbb{E}[Y_{2,h}^k | F_{k-1}] + 4H^3 \ln \frac{6HK(K+1)}{\delta} \right\},
$$

$$
E^{br} = \left\{ \forall h \in [H], K \geq 1 : \sum_{k=1}^K \mathbb{E}[Y_{3,h}^k | F_{k,h-1}] \leq 2 \sum_{k=1}^K Y_{3,h}^k + 18 \ln \frac{6HK(K+1)}{\delta} \right\},
$$

We define the good event $\mathbb{G} = \mathbb{G}_1 \cap \mathbb{G}_2$.

**Lemma 14.** *The good event $\mathbb{G}$ holds with a probability of at least $1 - \delta$.*

*Proof.* The analysis of the first event follows $E^{\mathrm{diff}}$ exactly as the one of $E^{\mathrm{diff}1}$ in Lemma 7: define $W_k = \mathbb{1}\left\{ \bar{V}_h^k(s) - V_h^{\pi^k}(s) \in [0, H], \forall h \in [H], s \in \mathcal{S} \right\}$ (which happens a.s. under $\mathbb{G}_1$ due to the optimism in Lemma 13 and truncation) and $\tilde{Y}_{1,h}^k = W_k Y_{1,h}^k$, which is bounded in $[0, H]$ and $F_{k,h}$-measurable. The corresponding event w.r.t. this modified variables $\tilde{E}^{\mathrm{diff}}$ then holds w.p. $1 - \frac{\delta}{6}$ by Lemma 25, and as in Lemma 7, we can use the fact that $\mathbb{G}_1 \cap \tilde{E}^{\mathrm{diff}} = \mathbb{G}_1 \cap E^{\mathrm{diff}}$ to conclude this part of the proof.

Moving to the second event, since $V_h^{\pi^k}(s, \boldsymbol{s}') \in [0, H]$, then $\sum_{h=1}^H Y_{2,h}^k \in [0, H^3]$. Therefore, by Lemma 25 (w.r.t. the filtration $F_k$) with $C = H^3$ and any fixed $K$, we get w.p. $1 - \frac{\delta}{6HK(K+1)}$ that

$$
\sum_{k=1}^K \sum_{h=1}^H Y_{2,h}^k \leq 2 \sum_{k=1}^K \sum_{h=1}^H \mathbb{E}[Y_{2,h}^k | F_{k-1}] + 4H^3 \ln \frac{6HK(K+1)}{\delta}.
$$

Taking the union bound on all possible values of $K \geq 1$ proves that $E^{\mathrm{Var}}$ holds w.p. at least $1 - \frac{\delta}{6}$.

Finally, by definition, we have that $Y_{3,h}^k = b_{k,h}^r(s_h^k, a_h^k) \in [0, 1]$ and is $F_{k,h}$-measurable. Thus, for any fixed $k \geq 1$ and $h \in [H]$, using Lemma 25, we have w.p. $1 - \frac{\delta}{6HK(K+1)}$ that

$$
\sum_{k=1}^K \mathbb{E}[Y_{3,h}^k | F_{k,h-1}] \leq \left(1 + \frac{1}{2}\right) \sum_{k=1}^K Y_{3,h}^k + 18 \ln \frac{6HK(K+1)}{\delta} \leq 2 \sum_{k=1}^K Y_{3,h}^k + 18 \ln \frac{6HK(K+1)}{\delta},
$$

so that due to the union bound, $E^{br}$ holds w.p. $1 - \frac{\delta}{6}$.

To conclude, $\mathbb{G}_1$ holds w.p. $1 - \frac{\delta}{2}$ (Lemma 5) and the events $\tilde{E}^{\mathrm{diff}}, E^{\mathrm{Var}}, E^{br}$ each hold w.p. $1 - \frac{\delta}{6}$. As before, when accounting to the fact that $\tilde{E}^{\mathrm{diff}}$ and $E^{\mathrm{diff}}$ are identical under $\mathbb{G}_1$, the event $G = \mathbb{G}_1 \cap \mathbb{G}_2$ holds w.p. at least $1 - \delta$. $\square$

## C.8 Regret Analysis

**Theorem 2.** *When running MVP-TL, with probability at least $1 - \delta$ uniformly for all $K \geq 1$, it holds that* $\text{Reg}^T(K) \leq \mathcal{O}\left(\sqrt{H^2 SK}\left(\sqrt{H} + \sqrt{A}\right) \ln \frac{SAHK}{\delta} + H^3 S^4 A^3 \left(\ln \frac{SAHK}{\delta}\right)^2\right).$

*Proof.* Assume that the event $\mathbb{G}$ holds, which by Lemma 14, happens with probability at least $1 - \delta$. In particular, throughout the proof, we use optimism (Lemma 13), which implies that $0 \leq V_h^{\pi^k}(s, s') \leq V_h^*(s, s') \leq \bar{V}_h^k(s, s') \leq 3H$ (the upper bound is also by the truncation), as well as $0 \leq V_h^{\pi^k}(s) \leq V_h^*(s) \leq \bar{V}_h^k(s) \leq H$.

We first focus on lower-bounding the value of the policy $\pi^k$: by Remark 2, we have

$$V_h^{\pi^k}(s) = \mathbb{E}_{\boldsymbol{s'} \sim P_h(s)}\left[r_h(s, \pi_h^k(s, \boldsymbol{s'})) + V_{h+1}^{\pi^k}(s'(\pi_h^k(s, \boldsymbol{s'})))\right]$$

$$= \mathbb{E}_{\boldsymbol{s'} \sim P_h(s)}\left[\hat{r}_h^{k-1}(s, \pi_h^k(s, \boldsymbol{s})) + \bar{V}_{h+1}^k(s'(\pi_h^k(s, \boldsymbol{s'}))) + b_{k,h}^r(s, \pi_h^k(s, \boldsymbol{s'}))\right]$$

$$+ \mathbb{E}_{\boldsymbol{s'} \sim P_h(s)}\left[r_h(s, \pi_h^k(s, \boldsymbol{s'})) - \hat{r}_h^{k-1}(s, \pi_h^k(s, \boldsymbol{s'})) - b_{k,h}^r(s, \pi_h^k(s, \boldsymbol{s'}))\right]$$

$$+ \mathbb{E}_{\boldsymbol{s'} \sim P_h(s)}\left[V_{h+1}^{\pi^k}(s'(\pi_h^k(s, \boldsymbol{s'}))) - \bar{V}_{h+1}^k(s'(\pi_h^k(s, \boldsymbol{s'})))\right]$$

$$\overset{(1)}{=} \mathbb{E}_{\boldsymbol{s'} \sim P_h(s)}\left[\max_{a \in \mathcal{A}}\{\hat{r}_h^{k-1}(s, a) + \bar{V}_{h+1}^k(s'(a)) + b_{k,h}^r(s, a)\}\right]$$

$$+ \mathbb{E}_{\boldsymbol{s'} \sim P_h(s)}\left[r_h(s, \pi_h^k(s, \boldsymbol{s'})) - \hat{r}_h^{k-1}(s, \pi_h^k(s, \boldsymbol{s})) - b_{k,h}^r(s, \pi_h^k(s, \boldsymbol{s'}))\right]$$

$$+ \mathbb{E}_{\boldsymbol{s'} \sim P_h(s)}\left[V_{h+1}^{\pi^k}(s'(\pi_h^k(s, \boldsymbol{s'}))) - \bar{V}_{h+1}^k(s'(\pi_h^k(s, \boldsymbol{s'})))\right]$$

$$\overset{(2)}{\geq} \mathbb{E}_{\boldsymbol{s'} \sim P_h(s)}\left[\bar{V}_h^k(s, \boldsymbol{s'})\right] - 2\mathbb{E}_{\boldsymbol{s'} \sim P_h(s)}\left[b_{k,h}^r(s, \pi_h^k(s, \boldsymbol{s'}))\right]$$

$$- \mathbb{E}_{\boldsymbol{s'} \sim P_h(s)}\left[\bar{V}_{h+1}^k(s'(\pi_h^k(s, \boldsymbol{s'}))) - V_{h+1}^{\pi^k}(s'(\pi_h^k(s, \boldsymbol{s'})))\right]$$

where (1) is by the definition of $\pi^k$ and (2) uses the reward concentration event. Thus, we can write

$$\bar{V}_h^k(s) - V_h^{\pi^k}(s) \leq \mathbb{E}_{\boldsymbol{s'} \sim \hat{P}_h^{k-1}(s)}\left[\bar{V}_h^k(s, \boldsymbol{s'})\right] - \mathbb{E}_{\boldsymbol{s'} \sim P_h(s)}\left[\bar{V}_h^k(s, \boldsymbol{s'})\right] + 2\mathbb{E}_{\boldsymbol{s'} \sim P_h(s)}\left[b_{k,h}^r(s, \pi_h^k(s, \boldsymbol{s'}))\right]$$

$$+ \mathbb{E}_{\boldsymbol{s'} \sim P_h(s)}\left[\bar{V}_{h+1}^k(s'(\pi_h^k(s, \boldsymbol{s'}))) - V_{h+1}^{\pi^k}(s'(\pi_h^k(s, \boldsymbol{s'})))\right] + b_{k,h}^p(s)$$

$$= \underbrace{\mathbb{E}_{\boldsymbol{s'} \sim \hat{P}_h^{k-1}(s)}\left[\bar{V}_h^k(s, \boldsymbol{s'}) - V_h^*(s, \boldsymbol{s'})\right] - \mathbb{E}_{\boldsymbol{s'} \sim P_h(s)}\left[\bar{V}_h^k(s, \boldsymbol{s'}) - V_h^*(s, \boldsymbol{s'})\right] + b_{k,h}^p(s)}_{(i)}$$

$$+ \underbrace{\mathbb{E}_{\boldsymbol{s'} \sim P_h(s)}[V_h^*(s, \boldsymbol{s'})] - \mathbb{E}_{\boldsymbol{s'} \sim \hat{P}_h^{k-1}(s)}[V_h^*(s, \boldsymbol{s'})]}_{(ii)} + 2\mathbb{E}_{\boldsymbol{s'} \sim P_h(s)}\left[b_{k,h}^r(s, \pi_h^k(s, \boldsymbol{s'}))\right]$$

$$+ \mathbb{E}_{\boldsymbol{s'} \sim P_h(s)}\left[\bar{V}_{h+1}^k(s'(\pi_h^k(s, \boldsymbol{s'}))) - V_{h+1}^{\pi^k}(s'(\pi_h^k(s, \boldsymbol{s'})))\right] \tag{14}$$

**Bounding term** $(ii)$**:** using the concentration event $E^{pv1}(k)$, we have

$$(ii) \leq \sqrt{\frac{2\text{Var}_{\boldsymbol{s'} \sim P_h(s)}(V_h^*(s, \boldsymbol{s'}))L_\delta^k}{n_h^{k-1}(s) \vee 1}} + \frac{HL_\delta^k}{n_h^{k-1}(s) \vee 1}$$

$$\overset{(1)}{\leq} \sqrt{\frac{2\text{Var}_{\boldsymbol{s'} \sim P_h(s)}(V_h^{\pi^k}(s, \boldsymbol{s'}))L_\delta^k}{n_h^{k-1}(s) \vee 1}} + \frac{1}{8H}\mathbb{E}_{\boldsymbol{s'} \sim P_h(s)}\left[V_h^{\pi^k}(s, \boldsymbol{s'}) - V_h^{\pi_k}(s, \boldsymbol{s'})\right] + \frac{4H^2 L_\delta^k}{n_h^{k-1}(s) \vee 1} + \frac{HL_\delta^k}{n_h^{k-1}(s) \vee 1}$$

$$\overset{(2)}{\leq} \sqrt{\frac{2\text{Var}_{\boldsymbol{s'} \sim P_h(s)}(V_h^{\pi^k}(s, \boldsymbol{s'}))L_\delta^k}{n_h^{k-1}(s) \vee 1}} + \frac{1}{8H}\mathbb{E}_{\boldsymbol{s'} \sim P_h(s)}\left[\bar{V}_h^k(s, \boldsymbol{s'}) - V_h^{\pi_k}(s, \boldsymbol{s'})\right] + \frac{5H^2 L_\delta^k}{n_h^{k-1}(s) \vee 1}.$$

$$\tag{15}$$

Relation (1) uses Lemma 21 with the values $0 \leq V_h^{\pi^k}(s, \boldsymbol{s'}) \leq V_h^*(s, \boldsymbol{s'}) \leq H$ with $\alpha = 8H \cdot \sqrt{2L_\delta^k}$ and (2) is by optimism.

**Bounding term** $(i)$**:** We first focus on the transition bonus; to bound it, we apply Lemma 22 w.r.t. $\hat{P}_h^{k-1}(s'|s), P_h(s'|s)$, the values $0 \le V_h^{\pi^k}(s, s') \le V_h^*(s, s') \le \bar{V}_h^k(s, s') \le 3H$ (by optimism), under the event $E^{pv2}(k)$ and with $\alpha = 8H \cdot \frac{20}{3}\sqrt{L_\delta^k}$:

$$b_{k,h}^p(s) = \frac{20}{3}\sqrt{\frac{\text{Var}_{s'\sim\hat{P}_h^{k-1}(s)}(\bar{V}_h^k(s, s'))L_\delta^k}{n_h^{k-1}(s)\vee 1}} + \frac{400}{3}\frac{HL_\delta^k}{n_h^{k-1}(s)\vee 1}$$

$$\le \frac{1}{8H}\mathbb{E}_{s'\sim\hat{P}_h^{k-1}(s)}\big[\bar{V}_h^k(s, s') - V_h^*(s, s')\big] + \frac{1}{8H}\mathbb{E}_{s'\sim P_h(s)}\big[V_h^*(s, s') - V_h^{\pi^k}(s, s')\big]$$

$$+ \frac{20}{3}\sqrt{\frac{\text{Var}_{s'\sim P_h(s)}(V_h^{\pi^k}(s, s'))L_\delta^k}{n_h^{k-1}(s)\vee 1}} + \frac{1600H^2}{3n_h^{k-1}(s)\vee 1} + \frac{20}{3}\frac{4HL_\delta^k}{n_h^{k-1}(s)\vee 1} + \frac{400}{3}\frac{HL_\delta^k}{n_h^{k-1}(s)\vee 1}$$

$$\le \frac{1}{8H}\Big(\mathbb{E}_{s'\sim\hat{P}_h^{k-1}(s)}\big[\bar{V}_h^k(s, s') - V_h^*(s, s')\big] - E_{s'\sim P_h(s)}\big[\bar{V}_h^k(s, s') - V_h^*(s, s')\big]\Big)$$

$$+ \frac{1}{8H}\mathbb{E}_{s'\sim P_h(s)}\big[\bar{V}_h^k(s, s') - V_h^{\pi^k}(s, s')\big] + \frac{20}{3}\sqrt{\frac{\text{Var}_{s'\sim P_h(s)}(V_h^{\pi^k}(s, s'))L_\delta^k}{n_h^{k-1}(s)\vee 1}} + \frac{700H^2}{n_h^{k-1}(s)\vee 1}.$$

Substituting back to term $(i)$, we now have

$$(i) \le \left(1 + \frac{1}{8H}\right)\Big(\mathbb{E}_{s'\sim\hat{P}_h^{k-1}(s)}\big[\bar{V}_h^k(s, s') - V_h^*(s, s')\big] - E_{s'\sim P_h(s)}\big[\bar{V}_h^k(s, s') - V_h^*(s, s')\big]\Big)$$

$$+ \frac{1}{8H}\mathbb{E}_{s'\sim P_h(s)}\big[\bar{V}_h^k(s, s') - V_h^{\pi^k}(s, s')\big] + \frac{20}{3}\sqrt{\frac{\text{Var}_{s'\sim P_h(s)}(V_h^{\pi^k}(s, s'))L_\delta^k}{n_h^{k-1}(s)\vee 1}} + \frac{700H^2L_\delta^k}{n_h^{k-1}(s)\vee 1}.$$

The next step in the proof involves bounding the first term of $(i)$. At this point, we remind that both values can be written as $\bar{V}_h^k(s, s') = \max_a \bar{V}_h^k(s, s'(a), a)$ and $V_h^*(s, s') = \max_a V_h^*(s, s'(a), a)$, inducing the lists $\bar{\ell} = \bar{\ell}_h^k(s)$ and $\ell^* = \ell_h^*(s)$, respectively; thus the expectations can be written as (see Appendix C.4 for further details on the list representation, and in particular, Equation (11)):

$$\mathbb{E}_{s'\sim\hat{P}_h^{k-1}(s)}\big[\bar{V}_h^k(s, s') - V_h^*(s, s')\big] - \mathbb{E}_{s'\sim P_h(s)}\big[\bar{V}_h^k(s, s') - V_h^*(s, s')\big]$$

$$\overset{(1)}{=} \mathbb{E}_{i,j\sim\hat{\mu}_h^k(\cdot|s;\bar{\ell},\ell^*)}\Big[\bar{V}_h^k(s, s'_{\bar{\ell}(i)}, a_{\bar{\ell}(i)}) - V_h^*(s, s'_{\ell^*(j)}, a_{\ell^*(j)})\Big]$$

$$- \mathbb{E}_{i,j\sim\mu(\cdot|\bar{\ell},\ell^*,P_h(s))}\Big[\bar{V}_h^k(s, s'_{\bar{\ell}(i)}, a_{\bar{\ell}(i)}) - V_h^*(s, s'_{\ell^*(j)}, a_{\ell^*(j)})\Big]$$

$$\overset{(2)}{\le} \frac{1}{8H}\mathbb{E}_{i,j\sim\mu(\cdot|\bar{\ell},\ell^*,P_h(s))}\Big[\bar{V}_h^k(s, s'_{\bar{\ell}(i)}, a_{\bar{\ell}(i)}) - V_h^*(s, s'_{\ell^*(j)}, a_{\ell^*(j)})\Big] + \frac{3H(SA)^2L_\delta^k(2SA + 8H\cdot 4SA/4)}{n_h^{k-1}(s)\vee 1}$$

$$\overset{(1)}{\le} \frac{1}{8H}\mathbb{E}_{s'\sim P_h(s)}\big[\bar{V}_h^k(s, s') - V_h^*(s, s')\big] + \frac{30H^2(SA)^3L_\delta^k}{n_h^{k-1}(s)\vee 1}$$

$$\le \frac{1}{8H}\mathbb{E}_{s'\sim P_h(s)}\big[\bar{V}_h^k(s, s') - V_h^{\pi^k}(s, s')\big] + \frac{30H^2(SA)^3L_\delta^k}{n_h^{k-1}(s)\vee 1}$$

Relations (1) formulate the expectation using the list representations and backward, as done in Equation (11). For inequality (2) we rely on Lemma 24 with $\alpha = 8H$ under the event $E^\ell(k)$ and the optimism, which ensures that the value difference is bounded in $[0, 3H]$. We also remark that the support of the distributions is of size $(SA)^2$; were we to use the same result on the distributions $\hat{P}_h^{k-1}(s)$ and $P_h(s)$, the support would be of size $S^A$, which would lead to an exponential additive factor. And so, we finally have a bound of

$$(i) \le \frac{3}{8H}\mathbb{E}_{s'\sim P_h(s)}\big[\bar{V}_h^k(s, s') - V_h^{\pi^k}(s, s')\big] + \frac{20}{3}\sqrt{\frac{\text{Var}_{s'\sim P_h(s)}(V_h^{\pi^k}(s, s'))L_\delta^k}{n_h^{k-1}(s)\vee 1}} + \frac{735H^2(SA)^3L_\delta^k}{n_h^{k-1}(s)\vee 1}.$$

$$(16)$$

**Combining both terms.** Substituting this and Equation (15) into Equation (14), we have

$$\bar{V}_h^k(s) - V_h^{\pi^k}(s) \leq \frac{1}{2H}\mathbb{E}_{\boldsymbol{s}'\sim P_h(s)}\left[\bar{V}_h^k(s,\boldsymbol{s}') - V_h^{\pi_k}(s,\boldsymbol{s}')\right] + 9\sqrt{\frac{\mathrm{Var}_{\boldsymbol{s}'\sim P_h(s)}(V_h^{\pi^k}(s,\boldsymbol{s}'))L_\delta^k}{n_h^{k-1}(s)\vee 1}} + \frac{750H^2(SA)^3L_\delta^k}{n_h^{k-1}(s)\vee 1}$$

$$+ 2\mathbb{E}_{\boldsymbol{s}'\sim P_h(s)}\left[b_{k,h}^r(s,\pi_h^k(s,\boldsymbol{s}'))\right] + \mathbb{E}_{\boldsymbol{s}'\sim P_h(s)}\left[\bar{V}_{h+1}^k(s'(\pi_h^k(s,\boldsymbol{s}'))) - V_{h+1}^{\pi^k}(s'(\pi_h^k(s,\boldsymbol{s}')))\right].$$

and further bounding (using the concentration event $E^r(k)$

$$\bar{V}_h^k(s,\boldsymbol{s}')) - V_h^{\pi^k}(s,\boldsymbol{s}') = \hat{r}_h^{k-1}(s,\pi_h^k(s,\boldsymbol{s}')) + b_{k,h}^r(s,\pi_h^k(s,\boldsymbol{s}')) + \bar{V}_{h+1}^k(s'(\pi_h^k(s,\boldsymbol{s}')))$$

$$- r_h^{k-1}(s,\pi_h^k(s,\boldsymbol{s}')) - V_{h+1}^{\pi^k}(s'(\pi_h^k(s,\boldsymbol{s}')))$$

$$\leq \bar{V}_{h+1}^k(s'(\pi_h^k(s,\boldsymbol{s}'))) - V_{h+1}^{\pi^k}(s'(\pi_h^k(s,\boldsymbol{s}'))) + 2b_{k,h}^r(s,\pi_h^k(s,\boldsymbol{s}')),$$

we finally get the decomposition

$$\bar{V}_h^k(s) - V_h^{\pi^k}(s) \leq \left(1 + \frac{1}{2H}\right)\mathbb{E}_{\boldsymbol{s}'\sim P_h(s)}\left[\bar{V}_{h+1}^k(s'(\pi_h^k(s,\boldsymbol{s}'))) - V_{h+1}^{\pi^k}(s'(\pi_h^k(s,\boldsymbol{s}')))\right]$$

$$+ 9\sqrt{\frac{\mathrm{Var}_{\boldsymbol{s}'\sim P_h(s)}(V_h^{\pi^k}(s,\boldsymbol{s}'))L_\delta^k}{n_h^{k-1}(s)\vee 1}} + \frac{750H^2(SA)^3L_\delta^k}{n_h^{k-1}(s)\vee 1} + 3\mathbb{E}_{\boldsymbol{s}'\sim P_h(s)}\left[b_{k,h}^r(s,\pi_h^k(s,\boldsymbol{s}'))\right].$$

At this point, we choose to take $s = s_h^k$ and sum over all $k \in [K]$; specifically, for $\boldsymbol{s}' = \boldsymbol{s}_{h+1}'^k$, the action becomes $\pi_h^k(s,\boldsymbol{s}') = a_h^k$ and $s'(\pi_h^k(s,\boldsymbol{s}')) = s_{h+1}^k$. Formally, we can write the bound as

$$\sum_{k=1}^K \bar{V}_h^k(s_h^k) - V_h^{\pi^k}(s_h^k) \leq \left(1 + \frac{1}{2H}\right)\sum_{k=1}^K \mathbb{E}\left[\bar{V}_{h+1}^k(s_{h+1}^k) - V_{h+1}^{\pi^k}(s_{h+1}^k)|F_{k,h-1}\right]$$

$$+ 3\sum_{k=1}^K \mathbb{E}\left[b_{k,h}^r(s_h^k,a_h^k)|F_{k,h-1}\right] + 9\sum_{k=1}^K \sqrt{\frac{\mathrm{Var}_{\boldsymbol{s}'\sim P_h(s_h^k)}(V_h^{\pi^k}(s_h^k,\boldsymbol{s}'))L_\delta^k}{n_h^{k-1}(s_h^k)\vee 1}}$$

$$+ \sum_{k=1}^K \frac{750H^2(SA)^3L_\delta^k}{n_h^{k-1}(s_h^k)\vee 1}.$$

and, in particular, under the events $E^{\mathrm{diff}}$ and $E^{br}$, it holds that

$$\sum_{k=1}^K \bar{V}_h^k(s_h^k) - V_h^{\pi^k}(s_h^k) \leq \left(1 + \frac{1}{2H}\right)^2\sum_{k=1}^K \left(\bar{V}_{h+1}^k(s_{h+1}^k)) - V_{h+1}^{\pi^k}(s_{h+1}^k)\right) + 36H^2\ln\frac{6HK(K+1)}{\delta}$$

$$+ 3\sum_{k=1}^K b_{k,h}^r(s_h^k,a_h^k) + 54\ln\frac{6HK(K+1)}{\delta}$$

$$+ 9\sum_{k=1}^K \sqrt{\frac{\mathrm{Var}_{\boldsymbol{s}'\sim P_h(s_h^k)}(V_h^{\pi^k}(s_h^k,\boldsymbol{s}'))L_\delta^k}{n_h^{k-1}(s_h^k)\vee 1}} + \sum_{k=1}^K \frac{750H^2(SA)^3L_\delta^k}{n_h^{k-1}(s_h^k)\vee 1}.$$

To conclude the proof, we recursively apply this formula from $h = 1$ to $h = H + 1$ (where the values are zero) and use the optimism. This yields

$$
\begin{aligned}
\text{Reg}^T(K) &= \sum_{k=1}^{K} V_1^*(s_h^k) - V_1^{\pi^k}(s_h^k) \\
&\le \sum_{k=1}^{K} \bar{V}_1^k(s_h^k) - V_1^{\pi^k}(s_h^k) \hspace{5cm} \text{(Optimism)} \\
&\stackrel{(1)}{\le} 9\left(1 + \frac{1}{2H}\right)^{2H} \sum_{k=1}^{K}\sum_{h=1}^{H} \frac{\sqrt{\text{Var}_{\boldsymbol{s'} \sim P_h(s_h^k)}(V_h^{\pi^k}(s_h^k, \boldsymbol{s'})) L_\delta^k}}{\sqrt{n_h^{k-1}(s_h^k) \vee 1}} \\
&\quad + 3\left(1 + \frac{1}{2H}\right)^{2H} \sum_{k=1}^{K}\sum_{h=1}^{H} \sqrt{\frac{L_\delta^k}{n_h^{k-1}(s_h^k, a_h^k) \vee 1}} \\
&\quad + \left(1 + \frac{1}{2H}\right)^{2H} \sum_{k=1}^{K}\sum_{h=1}^{H} \frac{750 H^2 (SA)^3 L_\delta^k}{n_h^{k-1}(s_h^k) \vee 1} + 90 H^3 \left(1 + \frac{1}{2H}\right)^{2H} \ln \frac{6HK(K+1)}{\delta} \\
&\stackrel{(2)}{\le} 50\sqrt{H^3 SK} L_\delta^K + 50\sqrt{2S} H^2 \left(L_\delta^K\right)^{1.5} \\
&\quad + 9\sqrt{L_\delta^K}\left(SAH + 2\sqrt{SAH^2 K}\right) + 2050 H^3 S^4 A^3 L_\delta^K (2 + \ln(K)) + 250 H^3 L_\delta^K \\
&= \mathcal{O}\left(\sqrt{H^2 SK}\left(\sqrt{H} + \sqrt{A}\right) L_\delta^K + H^3 S^4 A^3 \left(L_\delta^K\right)^2\right).
\end{aligned}
$$

Relation $(1)$ is the recursive application of the difference alongside substitution of the reward bonuses, while relation $(2)$ is by Lemma 15 and Lemma 20. $\hspace{1cm} \square$

### C.8.1 Lemmas for Bounding Bonus Terms

**Lemma 15.** *Under the event $E^{\mathrm{Var}}$ it holds that*

$$\sum_{k=1}^{K}\sum_{h=1}^{H}\frac{\sqrt{\mathrm{Var}_{\boldsymbol{s}'\sim P_h(s_h^k)}(V_h^{\pi^k}(s_h^k,\boldsymbol{s}'))}}{\sqrt{n_h^{k-1}(s_h^k)\vee 1}}\leq 2\sqrt{H^3SKL_\delta^K}+\sqrt{8S}H^2L_\delta^K.$$

*Proof.* Similar to Lemma 9, we again rely on the lookahead version of the law of total variation to prove this bound. First, by Cauchy-Schwartz inequality, it holds that

$$\sum_{k=1}^{K}\sum_{h=1}^{H}\frac{\sqrt{\mathrm{Var}_{\boldsymbol{s}'\sim P_h(s_h^k)}(V_h^{\pi^k}(s_h^k,\boldsymbol{s}'))}}{\sqrt{n_h^{k-1}(s_h^k)\vee 1}}\leq\sqrt{\sum_{k=1}^{K}\sum_{h=1}^{H}\mathrm{Var}_{\boldsymbol{s}'\sim P_h(s_h^k)}(V_h^{\pi^k}(s_h^k,\boldsymbol{s}'))}\sqrt{\sum_{k=1}^{K}\sum_{h=1}^{H}\frac{1}{n_h^{k-1}(s_h^k)\vee 1}}.$$

We use Lemma 20 to bound the second term by

$$\sum_{k=1}^{K}\sum_{h=1}^{H}\frac{1}{n_h^{k-1}(s_h^k)\vee 1}\leq SH(2+\ln(K))$$

and focus on bounding the first term. Under $E^{\mathrm{Var}}$, we have

$$\sum_{k=1}^{K}\sum_{h=1}^{H}\mathrm{Var}_{\boldsymbol{s}'\sim P_h(s_h^k)}(V_h^{\pi^k}(s_h^k,\boldsymbol{s}'))$$

$$\leq 2\sum_{k=1}^{K}\mathbb{E}\left[\sum_{h=1}^{H}\mathrm{Var}_{\boldsymbol{s}'\sim P_h(s_h^k)}(V_h^{\pi^k}(s_h^k,\boldsymbol{s}'))\Big|F_{k-1}\right]+4H^3\ln\frac{6HK(K+1)}{\delta}\qquad\text{(Under }E^{\mathrm{Var}})$$

$$=2\sum_{k=1}^{K}\mathbb{E}\left[\left(\sum_{h=1}^{H}r_h(s_h^k,a_h^k)-V_1^{\pi^k}(s_1^k)\right)^2\Big|F_{k-1}\right]+4H^3\ln\frac{6HK(K+1)}{\delta}\qquad\text{(By Lemma 10)}$$

$$\leq 2H^2K+4H^3\ln\frac{6HK(K+1)}{\delta},$$

where the last inequality is since both the values and cumulative rewards are bounded in $[0,H]$. Combining both, we get

$$\sum_{k=1}^{K}\sum_{h=1}^{H}\frac{\sqrt{\mathrm{Var}_{\boldsymbol{s}'\sim P_h(s_h^k)}(V_h^{\pi^k}(s_h^k,\boldsymbol{s}'))}}{\sqrt{n_h^{k-1}(s_h^k)\vee 1}}\leq\sqrt{2H^2K+4H^3\ln\frac{6HK(K+1)}{\delta}}\sqrt{SH(2+\ln(K))}$$

$$\leq\sqrt{2H^2K+4H^3\ln\frac{6HK(K+1)}{\delta}}\sqrt{2SH\ln\frac{6HK(K+1)}{\delta}}$$

$$\leq 2\sqrt{H^3SKL_\delta^K}+\sqrt{8S}H^2L_\delta^K.$$

$\square$

## C.9 Example: Value Gain due to Transition Lookahead

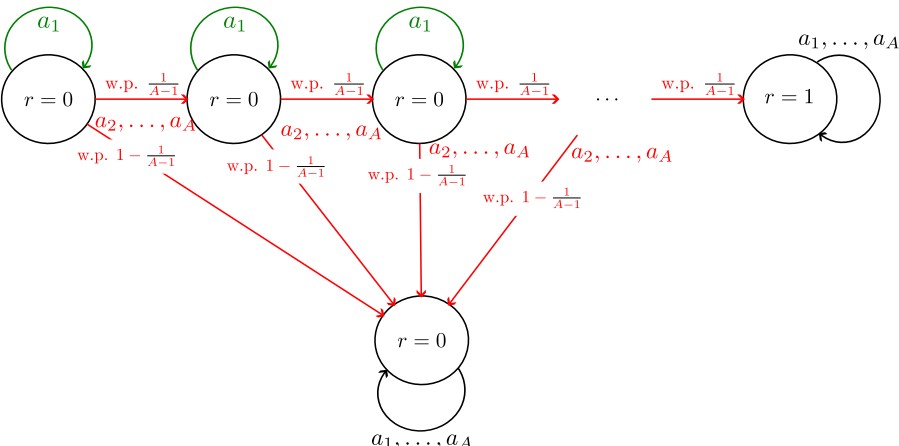

Figure 2: Random chain: agents start at the left side and must reach its right side to collect a reward.

We now present in further detail the example described at Section 3. This example is inspired by the one in Appendix C.3 in [Merlis et al., 2024], greatly simplifying it and achieving similar behavior for a much smaller environment.

Agents start at the left side of a chain of length $H/2$ (depicted in Figure 2) and have two options:

1. Play a safe action $a_1$ that leaves the agent in the same state (in green), or,
2. play one of the $A-1$ risky actions $a_2, \ldots, a_A$ (in red). Each of these actions moves the agent forward in the chain w.p. $\frac{1}{A-1}$, but leads to a terminal non-rewarding state w.p. $1 - \frac{1}{A-1}$.

At the end of the chain, the last state is an absorbing state with a unit reward.

Without lookahead, all agents can do is try to randomly reach the end of the chain, succeeding with probability $(A-1)^{-H/2}$. In particular, such agents cannot collect more than $V^{no} \leq H(A-1)^{-H/2}$. On the other hand, with transition lookahead, agents observe whether the risky actions allow moving forward in the chain or lead to the bad terminal state. If one action allows progressing in the chain (which happens w.p. $p = 1 - \left(1 - \frac{1}{A-1}\right)^{A-1} \geq 1 - 1/e$), a lookahead agent would take it, and otherwise, they will use $a_1$ to remain in the same state. In other words, optimal lookahead agents reach the reward after $H/2 - 1$ successful 'progression steps' with probability $p$ each. The probability of reaching the end of the chain using less than $5H/6$ steps is at least

$$\Pr\left(\mathrm{Bin}\left(\frac{5H}{6} - 1, 1 - \frac{1}{e}\right) > \frac{H}{2}\right) \geq c_0, \quad \text{for some absolute } c_0 > 0.$$

Under this event, the agent collects $\frac{H}{6}$ rewards, so the lookahead value is at least $V^{T,*} \geq \frac{c_0 H}{6} = \Omega(H)$.

To summarize, for this example, no lookahead optimal value is at most $\approx HA^{-H/2}$, while transition lookahead agents can collect a value of $\approx H$: transition lookahead increases the value by an exponential multiplicative factor. The difference between the two values is $G^T = \Omega(H)$, and following the discussion in Section 3, a sublinear transition lookahead regret would imply a negatively linear standard regret of $\mathrm{Reg}(K) \lesssim -HK$.

**Remark 3.** *The chain length was chosen to be $H/2$ for simplicity – similar conclusions can be achieved for a length of $\approx 1 - 1/e$. Then, the multiplicative increase in value due to transition lookahead would be $\approx (A-1)^{\left(1 - \frac{1}{e}\right)H}$, matching Proposition 2 in [Merlis et al., 2024]. In fact, setting the transition from the last state of the chain to the terminal state (rendering it possible to earn only one unit of reward), the analysis coincides with the one in [Merlis et al., 2024]. Following their exact derivation, the value with lookahead information is multiplicatively larger than its no-lookahead factor by an exponential factor of $\Theta\left((A-1)^{\min\left\{\left(1 - \frac{1}{e}\right)H - 1, S\right\} - 2}\right)$. This significantly improves the result in [Merlis et al., 2024], that only holds if $S \geq A^{\left(1 - \frac{1}{e}\right)H}$.*

# D   Auxiliary Lemmas

In this appendix, we prove various auxiliary lemma that will be used throughout our proofs.

## D.1   Concentration results

We first present and reprove a set of well-known concentration results.

**Lemma 16.** *Let $P$ be a distribution over a discrete set $\mathcal{X}$ of size $|\mathcal{X}| = M$ and let $X, X_1, \ldots, X_n$ be independent samples from this distribution. Also, let $U : \mathcal{X} \mapsto [0, C]$ for some $C > 0$ and define the empirical distribution $\hat{P}_n(x) = \frac{1}{n} \sum_{i=1}^{n} \mathbb{1}\{x_i = x\}$. Then, for any $\delta \in (0, 1)$, each of the following events hold w.p. at least $1 - \delta$:*

$$
E^p = \left\{ \forall x \in \mathcal{X}, |P(x) - \hat{P}_n(x)| \leq \sqrt{\frac{2P(x)\ln\frac{2M}{\delta}}{n}} + \frac{2\ln\frac{2M}{\delta}}{3n} \right\}
$$

$$
E^{pv1} = \left\{ \left| \sum_{x \in \mathcal{X}} \left( \hat{P}_n(x) - P(x) \right) U(x) \right| \leq \sqrt{\frac{2\mathrm{Var}_P(U(X))\ln\frac{2}{\delta}}{n}} + \frac{2C\ln\frac{2}{\delta}}{3n} \right\}
$$

$$
E^{pv2} = \left\{ \left| \sqrt{\mathrm{Var}_{\hat{P}_n}(U(X))} - \sqrt{\mathrm{Var}_P(U(X))} \right| \leq 4C\sqrt{\frac{\ln\frac{2}{\delta}}{n \vee 1}} \right\},
$$

*where $\mathrm{Var}_P(U(X)) = \sum_{x \in \mathcal{X}} P(x)U(x)^2 - \left( \sum_{x \in \mathcal{X}} P(x)U(x) \right)^2$.*

*Proof.* All the results require standard probability arguments and are stated for completeness.

For the first event $E^p$, notice that each of the components $\hat{P}_n(x)$ is the empirical mean of independent Bernoulli random variables $X_i(x)$ of mean $P(x)$. Therefore, by Bernstein's inequality, recalling that the variance of the variable $Ber(p)$ is $p(1-p)$, we get w.p. at least $1 - \frac{\delta}{M}$ that

$$
|P(x) - \hat{P}_n(x)| \leq \sqrt{\frac{2P(x)(1 - P(x))\ln\frac{2M}{\delta}}{n}} + \frac{2\ln\frac{2M}{\delta}}{3n} \leq \sqrt{\frac{2P(x)\ln\frac{2M}{\delta}}{n}} + \frac{2\ln\frac{2M}{\delta}}{3n}.
$$

Taking the union bound over all $x \in \mathcal{X}$ implies that $E^p$ holds w.p. at least $1 - \delta$.

For the second event $E^{pv1}$, we apply Bernstein's inequality on the variables $Y_i = U(X_i)$. The empirical mean is given by $\hat{Y}_n = \frac{1}{n} \sum_i U(X_i) = \sum_{x \in \mathcal{X}} \hat{P}_n(x)U(x)$ and its average is $\mathbb{E}[Y] = \sum_{x \in \mathcal{X}} P(x)U(x)$. Similarly, the variance of the random variables is $\mathrm{Var}(Y) = \mathrm{Var}_P(U(X))$. Thus, by Bernstein's inequality, w.p. at least $1 - \delta$,

$$
\left| \hat{Y}_n - \mathbb{E}[Y] \right| \leq \sqrt{\frac{2\mathrm{Var}(Y)\ln\frac{2}{\delta}}{n}} + \frac{2C\ln\frac{2}{\delta}}{3n}.
$$

Stating the bounds in terms of $X_i$ leads to the second event.

For the last event, we follow the analysis of [Efroni et al., 2021, Lemma 19], which in turn, relies on [Maurer and Pontil, 2009, Theorem 10]. Define $V_n = \frac{1}{2n(n-1)} \sum_{i,j=1}^{n} (U(X_i) - U(X_j))^2$. This is a well-known unbiased variance estimator, namely, $\mathbb{E}[V_n] = \mathrm{Var}_P(U(X))$, and by [Maurer and Pontil, 2009, Theorem 10], for any $\delta > 0$ it holds w.p. at least $1 - \delta$ that

$$
\left| \sqrt{V_n} - \sqrt{\mathrm{Var}_P(U(X))} \right| \leq C\sqrt{\frac{2\ln\frac{2}{\delta}}{n-1}},
$$

where we scaled the bound by $C$ to account for the values being in $[0, C]$.

Next, we relate $V_n$ to the empirical variance. By elementary algebra, we have

$$V_n = \frac{1}{2n(n-1)} \sum_{i,j=1}^{n} (U(X_i) - U(X_j))^2$$

$$= \frac{1}{n} \sum_{i=1}^{n} U(X_i)^2 - \frac{1}{n(n-1)} \sum_{i \neq j} U(X_i)U(X_j)$$

$$= \frac{1}{n} \sum_{i=1}^{n} U(X_i)^2 - \frac{n}{(n-1)} \left( \frac{1}{n} \sum_{i} U(X_i) \right)^2 + \frac{1}{n(n-1)} \sum_{i=1}^{n} U(X_i)^2$$

$$= \sum_{x \in \mathcal{X}} \hat{P}_n(x) U(x)^2 - \left( \sum_{x \in \mathcal{X}} \hat{P}_n(x) U(x) \right)^2 + \frac{1}{n(n-1)} \sum_{i=1}^{n} U(X_i)^2 - \frac{1}{n^2(n-1)} \left( \sum_{i=1}^{n} U(X_i) \right)^2.$$

The first two terms are exactly the variance w.r.t. the empirical distribution; therefore, using the inequality $\left| \sqrt{a} - \sqrt{b} \right| \leq \sqrt{|a - b|}$ for positive numbers, we have

$$\left| \sqrt{V_n} - \sqrt{\mathrm{Var}_{\hat{P}_n}(U(X))} \right| \leq \sqrt{\left| \frac{1}{n(n-1)} \sum_{i=1}^{n} U(X_i)^2 - \frac{1}{n^2(n-1)} \left( \sum_{i=1}^{n} U(X_i) \right)^2 \right|} \leq \sqrt{\frac{C^2}{n-1}}.$$

Combining both inequalities and recalling the trivial bound of $C$ on the difference, we get that w.p. at least $1 - \delta$,

$$\left| \sqrt{\mathrm{Var}_{\hat{P}_n}(U(X))} - \sqrt{\mathrm{Var}_P(U(X))} \right| \leq \min \left\{ C \sqrt{\frac{2 \ln \frac{2}{\delta}}{n-1}} + \sqrt{\frac{C^2}{n-1}}, C \right\} \leq 4C \sqrt{\frac{\ln \frac{2}{\delta}}{n \vee 1}}.$$

$\square$

Next, we present a short lemma that allows moving between different spaces of probabilities.

**Lemma 17.** *Let $\mathcal{X}$ be a finite set and let $X_1, \ldots, X_n \in \mathcal{X}$. Also, let $E_1, \ldots, E_m \subseteq \mathcal{X}$ be a partition of the set $\mathcal{X}$, namely, for all $i \neq j$, $E_i \cap E_j = \emptyset$ and $\cup_{i=1}^{m} E_i = \mathcal{X}$. Finally, let $f : \mathcal{X} \mapsto \mathbb{R}$ such that for all $i \in [m]$ and $x \in E_i$, it holds that $f(x) = f(i)$, and define*

$$\hat{P}_n(x) = \frac{1}{n} \sum_{\ell=1}^{n} \mathbb{1}\{X_\ell = x\}, \quad and, \quad \hat{Q}_n(i) = \frac{1}{n} \sum_{\ell=1}^{n} \mathbb{1}\{X_\ell \in E_i\}.$$

*Then, the following hold:*

1. *$\hat{Q}_n(i) = \hat{P}_n(E_i) \triangleq \sum_{x \in E_i} \hat{P}_n(x)$ and, in particular, $\mathbb{E}_{i \sim \hat{Q}_n}[f(i)] = \mathbb{E}_{x \sim \hat{P}_n}[f(x)]$.*

2. *If $P$ is a distribution over $\mathcal{X}$ and $X_1, \ldots, X_n \in \mathcal{X}$ are i.i.d. samples from $P$, then $\mathbb{E}[\hat{Q}_n(i)] = P(E_i) \triangleq Q(i)$. It also holds that $\mathbb{E}_{x \sim P}[f(x)] = \mathbb{E}_{i \sim Q}[f(i)]$.*

*Proof.* For the first part, we have by definition that

$$\hat{Q}_n(i) = \frac{1}{n} \sum_{\ell=1}^{n} \mathbb{1}\{X_\ell \in E_i\} = \sum_{x \in \mathcal{X}} \frac{1}{n} \sum_{\ell=1}^{n} \mathbb{1}\{X_\ell = x\} \mathbb{1}\{x \in E_i\} = \sum_{x \in \mathcal{X}} \hat{P}_n(x) \mathbb{1}\{x \in E_i\}$$

$$= \sum_{x \in E_i} \hat{P}_n(x) = \hat{P}_n(E_i).$$

In particular, it holds that

$$\mathbb{E}_{i \sim \hat{Q}_n}[f(i)] = \sum_{i=1}^{m} \hat{Q}_n(i) f(i) = \sum_{i=1}^{m} \sum_{x \in E_i} \hat{P}_n(x) f(i) \overset{(1)}{=} \sum_{i=1}^{m} \sum_{x \in E_i} \hat{P}_n(x) f(x) \overset{(2)}{=} \sum_{x \in \mathcal{X}} \hat{P}_n(x) f(x)$$

$$= \mathbb{E}_{x \sim \hat{P}_n}[f(x)],$$

where (1) is since $f$ is constant inside $E_i$ and (2) is since $\{E_i\}_{i=1}^m$ partition $\mathcal{X}$.

For the second part of the statement, notice that since the samples are i.i.d., it holds that $\mathbb{E}\left[\hat{P}_n(x)\right] = P(x)$, and therefore,

$$\mathbb{E}[\hat{Q}_n(i)] = \mathbb{E}\left[\sum_{x \in E_i} \hat{P}_n(x)\right] = \sum_{x \in E_i} P(x) = P(E_i) = Q(i).$$

Finally, as in the first part of the statement, it holds that

$$\mathbb{E}_{i \sim Q}[f(i)] = \sum_{i=1}^m Q(i)f(i) = \sum_{i=1}^m \sum_{x \in E_i} P(x)f(i) = \sum_{i=1}^m \sum_{x \in E_i} P(x)f(x) = \sum_{x \in \mathcal{X}} P(x)f(x)$$
$$= \mathbb{E}_{x \sim P}[f(x)].$$

$\qquad\square$

Finally, we present two specialized concentration results that are needed for reward and transition lookahead, respectively.

**Lemma 18.** *Let $X, X_1, \ldots X_n \in \mathbb{R}^d$ be i.i.d. random vectors over $[0,1]$ and let $C \geq 1$ be some constant. Then, for any $\delta \in (0,1)$, with probability at least $1 - \delta$,*

$$\forall u \in [0,C]^d, \quad \left| \mathbb{E}\left[\max_{i \in [d]}\{X(i) + u(i)\}\right] - \frac{1}{n} \sum_{\ell=1}^n \max_{i \in [d]}\{X_\ell(i) + u(i)\} \right| \leq 3\sqrt{\frac{d \ln \frac{9Cn}{\delta}}{2n}}.$$

*Proof.* Denote $m(u) = \mathbb{E}\left[\max_{i \in [d]}\{X(i) + u(i)\}\right]$ and $\hat{m}(u) = \frac{1}{n} \sum_{\ell=1}^n \max_{i \in [d]}\{X_\ell(i) + u(i)\}$ and fix any $u \in [0,C]^d$. Since the variables are bounded in $[0,1]$, their maximum is bounded almost surely in $[\max_i u(i), \max_i u(i) + 1]$, namely, an interval of unit length. Therefore, by Hoeffding's inequality, for any $\delta' \in (0,1)$, w.p. $1 - \delta'$

$$|m(u) - \hat{m}(u)| \leq \sqrt{\frac{\ln \frac{2}{\delta'}}{2n}}.$$

Now, for some $\epsilon \in (0,C]$, let $u_\epsilon$ be the closest vector to $u$ on a grid $\{0, \epsilon, 2\epsilon, \ldots, C\}^d$. Then, it clearly holds that

$$|m(u) - \hat{m}(u)| \leq |m(u_\epsilon) - \hat{m}(u_\epsilon)| + 2\epsilon.$$

Taking the union bound over all $\left(\left\lceil \frac{C}{\epsilon}\right\rceil + 1\right)^d$ possible choices for $u_\epsilon$ and fixing $\delta' = \frac{\delta}{\left(\left\lceil \frac{C}{\epsilon}\right\rceil+1\right)^d}$, we get w.p. $1 - \delta$ for all $u$ that

$$|m(u) - \hat{m}(u)| \leq \sqrt{\frac{\ln \frac{2\left(\left\lceil \frac{C}{\epsilon}\right\rceil+1\right)^d}{\delta}}{2n}} + 2\epsilon \leq \sqrt{\frac{d \ln \frac{6C}{\epsilon\delta}}{2n}} + 2\epsilon.$$

Now, fixing $\epsilon = \sqrt{\frac{d \ln \frac{6C}{\delta}}{2n}}$ and noting that $\frac{1}{\epsilon} \leq \sqrt{2n}$ for $C \geq 1$, we get

$$|m(u) - \hat{m}(u)| \leq \sqrt{\frac{d \ln \frac{6C\sqrt{2n}}{\delta}}{2n}} + 2\sqrt{\frac{d \ln \frac{6C}{\delta}}{2n}} \leq \sqrt{\frac{d \ln \frac{9Cn}{\delta}}{2n}} + 2\sqrt{\frac{d \ln \frac{6C}{\delta}}{2n}} \leq 3\sqrt{\frac{d \ln \frac{9Cn}{\delta}}{2n}}.$$

$\qquad\square$

**Lemma 19.** *Let $X, X_1, \ldots X_n \in \mathbb{R}^d$ be i.i.d. random vectors with components supported over the discrete set $[m]$ and let $C \geq 1$ be some constant. Then, uniformly over all $u \in [0, C]^{dm}$ w.p. $1 - \delta$:*

$$\left| \mathbb{E}\left[\max_i\{u(X(i), i)\}\right] - \frac{1}{n}\sum_{\ell=1}^n \max_i\{u(X_\ell(i), i)\} \right|$$

$$\leq \sqrt{\frac{2md\ln\frac{6n}{\delta}\operatorname{Var}(\max_i\{u(X(i), i)\})}{n}} + + \frac{8Cmd\left(\ln\frac{6n}{\delta}\right)^{1.5}}{n}.$$

*Proof.* We follow a similar path to Lemma 18 and use a covering argument. Denoting $w(u) = \mathbb{E}[\max_i\{u(X(i), i)\}]$ and $\hat{w}(u) = \frac{1}{n}\sum_{\ell=1}^n \max_i\{u(X_\ell(i), i)\}$, by Bernstein's inequality, for any $\delta' \in (0, 1)$ and fixed $u \in [0, C]^{dm}$, it holds w.p. $1 - \delta'$ that

$$|w(u) - \hat{w}(u)| \leq \sqrt{\frac{2\operatorname{Var}(\max_i\{u(X(i), i)\})\ln\frac{2}{\delta}}{n}} + \frac{2C\ln\frac{2}{\delta}}{3n}. \tag{17}$$

Now, for some $\epsilon \in (0, C]$, let $u_\epsilon$ be the closest matrix to $u$ on a grid $\{0, \epsilon, 2\epsilon, \ldots, C\}^{md}$ and denote $Z(u) = \max_i\{u(X(i), i)\}$ with samples $Z_i(u)$. By the smoothness of the max function, it holds that

$$|Z(u) - Z(u_\epsilon)| \leq \epsilon.$$

In particular, we also have that

$$\left|\mathbb{E}[Z(u)^2] - \mathbb{E}[Z(u_\epsilon)^2]\right| \leq \epsilon^2 + 2C\epsilon, \qquad \text{and} \qquad \left|\mathbb{E}[Z(u)]^2 - \mathbb{E}[Z(u_\epsilon)]^2\right| \leq \epsilon^2 + 2C\epsilon,$$

so we have

$$\left|\operatorname{Var}\left(\max_i\{u(X(i), i)\}\right) - \operatorname{Var}\left(\max_i\{u_\epsilon(X(i), i)\}\right)\right| = |\operatorname{Var}(Z(u)) - \operatorname{Var}(Z(u_\epsilon))| \leq 2\epsilon^2 + 4C\epsilon.$$

Similarly, it holds that

$$|w(u) - \hat{w}(u)| \leq |w(u_\epsilon) - \hat{w}(u_\epsilon)| + 2\epsilon.$$

Taking the union bound over all $\left(\lceil\frac{C}{\epsilon}\rceil + 1\right)^{md}$ possible choices for $u_\epsilon$ and fixing $\delta' = \frac{\delta}{\left(\lceil\frac{C}{\epsilon}\rceil+1\right)^{dm}}$, we get w.p. $1 - \delta$ for all $u$ that

$$|w(u) - \hat{w}(u)| \leq \sqrt{\frac{2\operatorname{Var}(\max_i\{u_\epsilon(X(i), i)\})\ln\frac{2\left(\lceil\frac{C}{\epsilon}\rceil+1\right)^{md}}{\delta}}{n}} + \frac{2C\ln\frac{2\left(\lceil\frac{C}{\epsilon}\rceil+1\right)^{md}}{\delta}}{3n} + 2\epsilon$$

$$\leq \sqrt{\frac{2md\operatorname{Var}(\max_i\{u_\epsilon(X(i), i)\})\ln\frac{6C}{\epsilon\delta}}{n}} + \frac{2Cmd\ln\frac{6C}{\epsilon\delta}}{3} + 2\epsilon$$

$$\leq \sqrt{\frac{2md\ln\frac{6C}{\epsilon\delta}(\operatorname{Var}(\max_i\{u(X(i), i)\}) + 2\epsilon^2 + 4C\epsilon)}{n}} + \frac{2Cmd\ln\frac{6C}{\epsilon\delta}}{3n} + 2\epsilon$$

$$\leq \sqrt{\frac{2md\ln\frac{6C}{\epsilon\delta}\operatorname{Var}(\max_i\{u(X(i), i)\})}{n}} + \sqrt{\frac{8mdC\epsilon\ln\frac{6C}{\epsilon\delta}}{n}} + \sqrt{\frac{4md\epsilon^2\ln\frac{6C}{\epsilon\delta}}{n}}$$

$$+ \frac{2Cmd\ln\frac{6C}{\epsilon\delta}}{3n} + 2\epsilon.$$

Now, fixing $\epsilon = \frac{C\ln\frac{6n}{\delta}}{n}$ and noticing that $\frac{6C}{\epsilon\delta} \leq \frac{6n}{\delta}$, we get

$$|w(u) - \hat{w}(u)| \leq \sqrt{\frac{2md\ln\frac{6n}{\delta}\operatorname{Var}(\max_i\{u(X(i), i)\})}{n}} + \frac{\sqrt{8md}C\ln\frac{6n}{\delta}}{n} + \frac{\sqrt{4md}C\left(\ln\frac{6n}{\delta}\right)^{1.5}}{n^{1.5}}$$

$$+ \frac{2Cmd\ln\frac{6n}{\delta}}{3n} + \frac{2C\ln\frac{6C}{\delta}}{n}$$

$$\leq \sqrt{\frac{2md\ln\frac{6n}{\delta}\operatorname{Var}(\max_i\{u(X(i), i)\})}{n}} + \frac{8Cmd\left(\ln\frac{6n}{\delta}\right)^{1.5}}{n}.$$

$\square$

## D.2 Count-Related Lemmas

**Lemma 20.** *The following bounds hold:*

$$\sum_{k=1}^{K}\sum_{h=1}^{H}\frac{1}{\sqrt{n_h^{k-1}(s_h^k,a_h^k)\vee 1}} \leq SAH + 2\sqrt{SAH^2K}, \quad \sum_{k=1}^{K}\sum_{h=1}^{H}\frac{1}{n_h^{k-1}(s_h^k,a_h^k)\vee 1} \leq SAH(2+\ln(K)),$$

$$\sum_{k=1}^{K}\sum_{h=1}^{H}\frac{1}{\sqrt{n_h^{k-1}(s_h^k)\vee 1}} \leq SH + 2\sqrt{SH^2K}, \quad \sum_{k=1}^{K}\sum_{h=1}^{H}\frac{1}{n_h^{k-1}(s_h^k)\vee 1} \leq SH(2+\ln(K)).$$

*Proof.* Recall that every time a state (or state-action) is visited, its visitation-count is increased by 1, up to $n_h^{K-1}(s,a)$ at the last episode. therefore, we can write

$$\sum_{k=1}^{K}\sum_{h=1}^{H}\frac{1}{\sqrt{n_h^{k-1}(s_h^k,a_h^k)\vee 1}} = \sum_{h=1}^{H}\sum_{s\in\mathcal{S}}\sum_{a\in\mathcal{A}}\sum_{k=1}^{K}\frac{\mathbb{1}\{s_h^k=s,a_h^k=a\}}{\sqrt{n_h^{k-1}(s,a)\vee 1}}$$

$$= \sum_{h=1}^{H}\sum_{s\in\mathcal{S}}\sum_{a\in\mathcal{A}}\sum_{i=0}^{n_h^{K-1}(s,a)}\frac{1}{\sqrt{i\vee 1}}$$

$$\leq \sum_{h=1}^{H}\sum_{s\in\mathcal{S}}\sum_{a\in\mathcal{A}}\left(1+2\sqrt{n_h^{K-1}(s,a)}\right)$$

$$\leq SAH + 2\sqrt{SAH\sum_{h=1}^{H}\sum_{s\in\mathcal{S}}\sum_{a\in\mathcal{A}}n_h^{K-1}(s,a)} \quad \text{(Jensen's inequality)}$$

$$\leq SAH + 2\sqrt{SAH^2K}.$$

where we bounded the total number of visits by the number of steps $HK$. Similarly, we also have

$$\sum_{k=1}^{K}\sum_{h=1}^{H}\frac{1}{n_h^{k-1}(s_h^k,a_h^k)\vee 1} = \sum_{h=1}^{H}\sum_{s\in\mathcal{S}}\sum_{a\in\mathcal{A}}\sum_{i=0}^{n_h^{K-1}(s,a)}\frac{1}{i\vee 1}$$

$$\leq \sum_{h=1}^{H}\sum_{s\in\mathcal{S}}\sum_{a\in\mathcal{A}}\left(2+\ln\left(n_h^{K-1}(s,a)\vee 1\right)\right) \leq SAH(2+\ln(K)).$$

We can likewise prove the inequalities for the state counts as follows:

$$\sum_{k=1}^{K}\sum_{h=1}^{H}\frac{1}{\sqrt{n_h^{k-1}(s_h^k)\vee 1}} = \sum_{h=1}^{H}\sum_{s\in\mathcal{S}}\sum_{k=1}^{K}\frac{\mathbb{1}\{s_h^k=s\}}{\sqrt{n_h^{k-1}(s)\vee 1}}$$

$$= \sum_{h=1}^{H}\sum_{s\in\mathcal{S}}\sum_{i=0}^{n_h^{K-1}(s)}\frac{1}{\sqrt{i\vee 1}}$$

$$\leq \sum_{h=1}^{H}\sum_{s\in\mathcal{S}}\left(1+2\sqrt{n_h^{K-1}(s)}\right)$$

$$\leq SH + 2\sqrt{SH\sum_{h=1}^{H}\sum_{s\in\mathcal{S}}n_h^{K-1}(s)} \quad \text{(Jensen's inequality)}$$

$$\leq SH + 2\sqrt{SH^2K},$$

and

$$\sum_{k=1}^{K}\sum_{h=1}^{H}\frac{1}{n_h^{k-1}(s_h^k)\vee 1} = \sum_{h=1}^{H}\sum_{s\in\mathcal{S}}\sum_{i=0}^{n_h^{K-1}(s)}\frac{1}{i\vee 1} \leq \sum_{h=1}^{H}\sum_{s\in\mathcal{S}}\left(2+\ln\left(n_h^{K-1}(s)\vee 1\right)\right) \leq SH(2+\ln(K)).$$

$\square$

### D.3 Analysis of Variance terms

**Lemma 21.** *Let $P$ be a distribution over a finite set $\mathcal{X}$ and let $X \sim P$. Also, let $V_1, V_2 : \mathcal{X} \mapsto [0, C]$ for some $C > 0$ such that $V_1(x) \leq V_2(x)$ for all $x \in \mathcal{X}$. Then, for any $\alpha, n > 0$, it holds that*

$$\frac{\sqrt{\mathrm{Var}_P(V_2(X))}}{\sqrt{n}} \leq \frac{\sqrt{\mathrm{Var}_P(V_1(X))}}{\sqrt{n}} + \frac{1}{\alpha}\mathbb{E}_P[V_2(X) - V_1(X)] + \frac{C\alpha}{4n}$$

*Proof.* By Lemma 26, we have

$$\sqrt{\mathrm{Var}_P(V_2(X))} - \sqrt{\mathrm{Var}_P(V_1(X))} \leq \sqrt{\mathrm{Var}_P(V_2(X) - V_1(X))}$$
$$\leq \sqrt{\mathbb{E}_P[(V_2(X) - V_1(X))^2]}$$
$$\leq \sqrt{C\mathbb{E}_P[V_2(X) - V_1(X)]}$$

where the last inequality is by the boundedness and since $V_1(x) \leq V_2(x)$. Thus, we can bound

$$\frac{\sqrt{\mathrm{Var}_P(V_2(X))} - \sqrt{\mathrm{Var}_P(V_1(X))}}{\sqrt{n}} \leq \frac{\sqrt{C\mathbb{E}_P[V_2(X) - V_1(X)]}}{\sqrt{n}}$$
$$= \sqrt{\mathbb{E}_P[V_2(X) - V_1(X)]} \cdot \sqrt{\frac{C}{n}}$$
$$\leq \frac{1}{\alpha}\mathbb{E}_P[V_2(X) - V_1(X)] + \frac{C\alpha}{4n},$$

where last inequality is due to Young's inequality ($ab \leq \frac{1}{\alpha}a^2 + \frac{\alpha}{4}b^2$ for all $\alpha > 0$). $\qquad\square$

**Lemma 22.** *Let $P, P'$ be distributions over a finite set $\mathcal{X}$ and let $X \sim P$. Also, let $V_1, V_2, V_3 : \mathcal{X} \mapsto [0, C]$ for some $C > 0$ such that $V_1(x) \leq V_2(x) \leq V_3(x)$ for all $x \in \mathcal{X}$. Finally, assume that*

$$\left| \sqrt{\mathrm{Var}_P(V_2(X))} - \sqrt{\mathrm{Var}_{P'}(V_2(X))} \right| \leq \beta$$

*for some $\beta > 0$. Then, for any $\alpha, n > 0$, it holds that*

$$\frac{\sqrt{\mathrm{Var}_{P'}(V_3(X))}}{\sqrt{n}} \leq \frac{\sqrt{\mathrm{Var}_P(V_1(X))}}{\sqrt{n}} + \frac{1}{\alpha}\mathbb{E}_{P'}[V_3(X) - V_2(X)] + \frac{1}{\alpha}\mathbb{E}_P[V_2(X) - V_1(X)] + \frac{C\alpha}{2n} + \frac{\beta}{\sqrt{n}}$$
$$\leq \frac{\sqrt{\mathrm{Var}_P(V_1(X))}}{\sqrt{n}} + \frac{1}{\alpha}\mathbb{E}_{P'}[V_3(X) - V_1(X)] + \frac{1}{\alpha}\mathbb{E}_P[V_3(X) - V_1(X)] + \frac{C\alpha}{2n} + \frac{\beta}{\sqrt{n}}.$$

*Proof.* We decompose the l.h.s. as follows

$$\frac{\sqrt{\mathrm{Var}_{P'}(V_3(X))}}{\sqrt{n}} = \frac{\sqrt{\mathrm{Var}_{P'}(V_3(X))} - \sqrt{\mathrm{Var}_{P'}(V_2(X))}}{\sqrt{n}} + \frac{\sqrt{\mathrm{Var}_{P'}(V_2(X))} - \sqrt{\mathrm{Var}_P(V_2(X))}}{\sqrt{n}}$$
$$+ \frac{\sqrt{\mathrm{Var}_P(V_2(X))} - \sqrt{\mathrm{Var}_P(V_1(X))}}{\sqrt{n}} + \frac{\sqrt{\mathrm{Var}_P(V_1(X))}}{\sqrt{n}}$$

We bound the first and third terms using Lemma 21 and bound the second term with the assumption and get

$$\frac{\sqrt{\mathrm{Var}_{P'}(V_3(X))}}{\sqrt{n}} \leq \frac{1}{\alpha}\mathbb{E}_{P'}[V_3(X) - V_2(X)] + \frac{C\alpha}{4n} + \frac{\beta}{\sqrt{n}}$$
$$+ \frac{1}{\alpha}\mathbb{E}_P[V_2(X) - V_1(X)] + \frac{C\alpha}{4n} + \frac{\sqrt{\mathrm{Var}_P(V_1(X))}}{\sqrt{n}}$$
$$= \frac{\sqrt{\mathrm{Var}_P(V_1(X))}}{\sqrt{n}} + \frac{1}{\alpha}\mathbb{E}_{P'}[V_3(X) - V_2(X)] + \frac{1}{\alpha}\mathbb{E}_P[V_2(X) - V_1(X)] + \frac{C\alpha}{2n} + \frac{\beta}{\sqrt{n}}$$
$$\leq \frac{\sqrt{\mathrm{Var}_P(V_1(X))}}{\sqrt{n}} + \frac{1}{\alpha}\mathbb{E}_{P'}[V_3(X) - V_1(X)] + \frac{1}{\alpha}\mathbb{E}_P[V_3(X) - V_1(X)] + \frac{C\alpha}{2n} + \frac{\beta}{\sqrt{n}},$$

where the last inequality uses the fact that $V_1(x) \leq V_2(x) \leq V_3(x)$ for all $x \in \mathcal{X}$. The last two bounds are the desired results. $\qquad\square$

# E  Existing Results

**Lemma 23** (Monotonic Bonuses,[Zhang et al., 2023], Appendix C.1). *For any $p \in \Delta^S$, $v \in \mathbb{R}_+^S$ s.t. $\|v\|_\infty \leq H$, $\delta' \in (0,1)$ and positive integer $n$, define the function*

$$f(p,v,n) = p^T v + \max\left\{\frac{20}{3}\sqrt{\frac{\mathrm{Var}_p(v)\ln\frac{1}{\delta'}}{n}}, \frac{400}{9}\frac{H\ln\frac{1}{\delta'}}{n}\right\}.$$

*Then, the function $f(p,v,n)$ is non-decreasing in each entry of $v$.*

**Lemma 24** (Efroni et al. 2021, Lemma 28). *Let $Y \in \mathbb{R}^S$ be a vector such that $0 \leq Y(s) \leq H$ for all $s \in \mathcal{S}$. Let $P_1$ and $P_2$ be two transition models and $n \in \mathbb{R}_+^{SA}$. If*

$$\left\{\forall (s,a,s') \in \mathcal{S} \times \mathcal{A} \times \mathcal{S}, h \in [H] : |P_{2,h}(s'|s,a) - P_{1,h}(s'|s,a)| \leq \sqrt{\frac{C_1 L_\delta^k P_{1,h}(s'|s,a)}{n(s,a) \vee 1}} + \frac{C_2 L_\delta^k}{n(s,a) \vee 1}\right\},$$

*for some $C_1, C_2 > 0$, then, for any $\alpha > 0$,*

$$|(P_{1,h} - P_{2,h})Y(s,a)| \leq \frac{1}{\alpha}\mathbb{E}_{s'\sim P_{1,h}(\cdot|s,a)}[Y(s')] + \frac{HSL_\delta^k(C_2 + \alpha C_1/4)}{n(s,a) \vee 1},$$

**Lemma 25** (Efroni et al. 2021, Lemma 27). *Let $\{Y_t\}_{t\geq 1}$ be a real-valued sequence of random variables adapted to a filtration $\{F_t\}_{t\geq 0}$. Assume that for all $t \geq 1$ it holds that $0 \leq Y_t \leq C$ a.s., and let $T \in \mathbb{N}$. Then each of the following inequalities holds with probability greater than $1 - \delta$.*

$$\sum_{t=1}^T \mathbb{E}[Y_t|F_{t-1}] \leq \left(1 + \frac{1}{2C}\right)\sum_{t=1}^T Y_t + 2(2C+1)^2\ln\frac{1}{\delta},$$

$$\sum_{t=1}^T Y_t \leq 2\sum_{t=1}^T \mathbb{E}[Y_t|F_{t-1}] + 4C\ln\frac{1}{\delta}.$$

**Lemma 26** (Standard Deviation Differences, e.g., Zanette and Brunskill 2019, lines 48-51). *Let $P \in \Delta_d$ be some distribution over $[d]$ and let $V_1, V_2 \in \mathbb{R}^d$. Then, it holds that*

$$\sqrt{\mathrm{Var}_P(V_1)} - \sqrt{\mathrm{Var}_P(V_2)} \leq \sqrt{\mathrm{Var}_P(V_1 - V_2)}.$$

**Lemma 27** (Law of Total Variance, e.g., Zanette and Brunskill 2019, Lemma 15). *For any no-lookahead policy $\pi$, it holds that*

$$\mathbb{E}\left[\sum_{h=1}^H \mathrm{Var}(V_{h+1}^\pi(s_{h+1})|s_h)|\pi, s_1\right] = \mathbb{E}\left[\left(\sum_{h=1}^H r_h(s_h, a_h) - V_1^\pi(s_1)\right)^2 |\pi, s_1\right],$$

*where $\mathrm{Var}(V_{h+1}^\pi(s_{h+1})|s_h)$ is the variance of the value at step $s_{h+1}$ given state $s_h$ and under the policy $\pi$, due to the policy randomization and next-state transition probabilities.*

