# OpenReview forum: "Reinforcement Learning with Lookahead Information"
_NeurIPS.cc/2024/Conference — NeurIPS 2024 poster_

### Official Review · Reviewer_Dj8h · 2024-06-20

**Soundness:** 3
**Presentation:** 4
**Contribution:** 3
**Rating:** 6
**Confidence:** 1

**Summary:**

This paper introduces reinforcement learning (RL) problems where agents observe one-step lookahead information (either rewards or transitions) before choosing actions in episodic tabular MDPs. Two relevant lines of work exist: the control literature, which studies a similar lookahead concept in the continuous state-space scenario, and the RL planning community, which commonly obtains lookahead information from learned transition models. However, this paper assumes the reward/transition information to be available before selecting an action. The core contributions are:

1) Formalising the look-ahead setting for the reward and transition in an episodic MDP setting.
2) Derivation of the Bellman equations in the original space by setting up an equivalence with an equivalent new MDP.
3) Development of two algorithms for reward (MVP-RL) and transition lookahead ( MVP-TL).
4) First sub-linear regret bound win the lookahead setting.

**Strengths:**

This paper is the first to provide regret bound on the lookahead learning setting. This encompass a somewhat broad spectrum of problems that were independently studied such as the Canadian traveler problem and the prophet inequalities.

They paper is well written and easy to follow for non-expert in learning theory. It presents the core ideas in an understandable way in the main paper and use the appendix for technical proofs.

**Weaknesses:**

The paper could be strengthened by adding experimental results studying the difference in behaviour and performance between standard RL algorithm, MVP and the proposed solution MVP-RL. More specifically, I would be interested in understanding the difference in behaviour when changing the tails of the reward/transition distributions.

**Questions:**

How applicable is the theoretical argument you used to a model-free version of MVP-RL/TL?

**Limitations:**

The limitations outlined in the paper provide a fair representation of how the theoretical results could be extended in various directions, such as multi-step and stochastic action sets.

---

> ### Author Rebuttal · Authors · 2024-08-04
>
> We thank the reviewer for the positive feedback and refer to the general comment regarding experiments. In particular, MVP converges to the no-lookahead value, so there would be a linear difference between its performance and our algorithms, so the algorithms are not really comparable.
>
> While the tail of the distribution will somewhat affect the sublinear regret rates, its dominant effect would be on the lookahead value. This is since any effect on the value is equivalent to linear regret terms - much larger than the $\sqrt{K}$ terms. In simple environments, the effect of the tail on the value can be analyzed in closed form, so we do not need to resort to numerical evaluation. One interesting property is that the behavior of the lookahead value is not a simple function of the tail, and similar tails might have many different behaviors.
>
> -----
>
>  Consider the following examples:
>
> **Reward lookahead in a prophet-like problem (Figure 1).** The value only depends on the reward distribution when moving from $s_i$ to $s_f$.
>
> 1) *Bernoulli distributions.* Assume that when transitioning from $s_i$ to $s_f$, we obtain rewards $Ber(p)$. In particular, the optimal no-lookahead value is always $V^{no-lookahead}=p$. On the other hand, the lookahead value will be the probability of observing at least one unit reward $V^{R,*} = 1 - (1-p)^{AH}$, and in particular,
>     - If $p<1/(AH)$, we get that $V^{R,*}\approx HAp$ - higher by a factor of $AH$ than the no-lookahead value.
>     - If $p\approx 1/2$, then the lookahead value will approach $V^{R,*}\approx 1$ - constant-factor improvement.
>     - Finally, for $p\approx 1$, both values are roughly $1$ - no improvement.
>
>     Interestingly, when thinking on standard ‘tail measures’ (e.g., Chernoff bound for averages), the *least concentrated* situation (with the heaviest tails) is when $p\approx 1/2$, even when only considering the upper tail of the distribution. On the other hand, the two lighter-tailed situations behave very differently: in one, we get a huge performance boost while in the other, we gain nothing.
>
> 2) *Gaussian distributions* $\mathcal{N}(\mu,\sigma)$. Analytically calculating the value is quite hard, but using the properties of the maximum of Normal R.V.s, it is quite easy to (approximately) bound the reward-lookahead value in the interval $V^{R,*}\in [\mu+\sigma\log(A), \mu+\sigma\log(AH)]$. Without lookahead, the value would be $V^{no-lookahead}=\mu$, so the difference between the values is proportional to the variance.
>
> 3) More generally, if we limit ourselves to bounded distributions with expectation $\mu$ and  effective support $[a,b]$, for large horizons, the lookahead value approaches $b$, while the no lookahead value will stay $\mu$. The specific tail might affect the rate at which it happens, but will not change the limiting value.
>
> -----
>
> **Tail effects with transition lookahead.** We admit that we are not familiar with a notion of a tail for a transition kernel over a finite state space, but we will try to illustrate how stochasticity in the transitions affects lookahead values. Let us still consider transition lookahead on a chain (variant of the example in Section 4). To simplify things, assume that we have a chain of length $n<H$ with two actions. One action moves forward w.p. $p$ (and w.p. $1-p$, we ‘fail’ and cannot collect any rewards). A second action does not change the state. We aim to reach the end of the chain to collect a unit reward. For no-lookahead agents, the probability of successfully traversing the chain is $p^{n}$. For one-step lookahead agents, it is the probability of having at least $n$ successes for the binomial $Bin(H,p)$. In particular,
> - When $p>> 1- 1/n$, both agents will succeed to collect the reward - have unit values.
> - For $p<< n/H$, both are likely to fail, but the lookahead agent still has an exponentially larger probability to succeed traversing the chain.
> - In the intermediate regime, the lookahead agent will succeed and the no-lookahead agent will fail.
>
> Thus, in some senses, the lookahead is more valuable in the ‘heavy tail’ regime (intermediate probability), but we see two different behaviors in the ‘light tail’ regime - similar values for large $p$ and exponentially different values for small $p$.
>
> -----
>
> **To summarize:** while it seems that ‘heavier tail’ generally helps lookahead agents, there is no clear way to translate tail properties to lookahead values.
>
> -----
>
> **Model-Free Algorithms:** generally speaking, even in the classical RL setting, the analysis techniques of model-based on model-free algorithms are very different, and we are not aware of any study that unifies regret analysis for both settings. We still believe that some elements could be taken from our work to the model-free setting - for example, using the structure and properties of the optimal value and the optimal policy (and, in particular, the list structure for transition lookahead), or using uniform concentration and avoiding pessimism. Nonetheless, the regret analysis will be more involved than the classic analysis of Q-learning [19] and will probably require new notions of Q-values that allow incorporating the observations into the policy. For example, with reward lookahead, the Q-values should not include immediate rewards, so that the reward observations could be later added when interacting with the environment.

---

### Official Review · Reviewer_V5fX · 2024-07-07

**Soundness:** 2
**Presentation:** 3
**Contribution:** 3
**Rating:** 6
**Confidence:** 3

**Summary:**

The authors proposed new forms of Bellman equations for environments where the agent knows the reward or transition outcomes one step ahead (without knowing the full model).

**Strengths:**

While previous papers (e.g., Boutilier et al. 2018) discussed utilizing lookahead information (and proved convergence), the authors claim they are the first to present regret results.

**Weaknesses:**

While the theoretical contribution is clear, the authors must also provide practical validation.

**Questions:**

- Perform experimental validation to illustrate the practical performance.
  For instance, it would be necessary to check the learning curves and the resulting performance.
  The authors should also discuss the practical implementation.

- H is the important parameter to be determined. Provide a practical guide line for choosing H.

- can this method be extended to off-policy learning ?
  Is the method data-efficient ?

**Limitations:**

Some aspects of the approach might be seen as incremental advancements  rather than groundbreaking theoretical analysis .

---

> ### Author Rebuttal · Authors · 2024-08-04
>
> We thank the reviewer for the feedback and refer to the general response on simulations.
>
> * As we explained in the general response, we agree that evaluation is important and believe that the best way to do so is to adapt our work to a deep RL setting, but this is outside the scope of this paper. Nonetheless, we would like to emphasize that our theoretical bounds *provably guarantee* well-behaved learning curves in tabular settings - in fact, since the bounds are minimax optimal, there exist some environments in which our algorithms converge at the optimal rate (up to absolute constants). We present a detailed algorithm description that includes the exact implementation details of both algorithms in Appendices B.3 and C.3, including the precise statement of the bonus and the update rules; any implementation of the theoretical algorithm would follow these schemes to the letter.
>
> * Horizon H: the horizon is indeed a very important parameter in episodic RL - after H steps, the interaction is over and a new episode starts. In most applications, the choice of the horizon is induced by the environment - the interaction *cannot* last more than a fixed number of steps (‘external termination of the agent/timeout’). In this sense, it is very different than a discount factor that induces a soft effective horizon - often tuned by the algorithm designer. Finite-horizon algorithms can also work with any upper bound on the number of steps - though this degrades the performance, the effect of $H$ on the regret is usually assumed to be much less significant than any dependence on the state space. In general, and to our knowledge, all previous works on finite-horizon regret minimization assume the horizon is given to the agent as an input [19, 9, 33, 12, 14, 30, 35, 36]. Nonetheless, a few papers study regret minimization in alternative interaction models, including average reward [18] or stochastic shortest paths [*]. We believe that extending our results to these settings is an interesting future work.
>
> * Our approach achieves optimal data efficiency, in the sense that our regret bounds are minimax optimal. In other words, for any algorithm, there exists an environment such that the difference between the collected reward and the optimal one must be larger than ours (up to absolute/additive constant). More broadly, our approach is model-based, and such approaches tend to use data more efficiently than model-free methods.
>
> * We admit that we do not fully understand the question about off-policy learning. In our algorithm, we use all the data from historical interactions to estimate an environment model and use this model to perform optimistic planning - calculate the optimal agent at an optimistic environment. This includes data from previous deployed policies which are different from the current one. In addition, while this planning module outputs a value that determines our policy, we never perform a policy evaluation of previously used policies and do not rely on policy iteration/computations that are usually considered as policy improvement. As such, we believe that our algorithm can be described as an off-policy algorithm. If the question was about using offline data - we believe that such data could be utilized to improve our model estimation. If it was about model-free algorithms - as we mention in the conclusions section, extending our paper to model-free algorithms (e.g., Q-learning) would be very natural - we refer to the response to reviewer Dj8h for more details. In case we did not answer the intended question, we would appreciate it if the reviewer could clarify the question and we will gladly answer in the reviewer-author discussion period.
>
> ------
>
> [*] Rosenberg, Aviv, et al. "Near-optimal regret bounds for stochastic shortest path." International Conference on Machine Learning. PMLR, 2020.

---

> > ### Comment · Reviewer_V5fX · 2024-08-13
> >
> > Thanks for your reply. I will maintain my score after reading your reply.

---

### Official Review · Reviewer_CUjN · 2024-07-08

**Soundness:** 3
**Presentation:** 3
**Contribution:** 3
**Rating:** 5
**Confidence:** 3

**Summary:**

This manuscript proposes the RL method with lookahead information. The authors discuss two scenarios: reward lookahead and transition lookahead. Under such scenarios, the proposed method estimates the reward distribution and transition distribution, respectively. Then the monotonic value propagation skill is applied to calculate the value function. The authors show that the proposed method has strong theoretical properties and the reward regret is strongly bounded under two circumstances.

**Strengths:**

The manuscript is well organized, and the structure is clear. The authors shows very promising bound for both reward lookahead and transition lookahead scenarios.

**Weaknesses:**

This is a theoretical paper, however, the authors miss to deliver some numerical or empirical studies. It is suggested to add some empirical experiments, at least with simulated data.

Algorithm 1&2 shows the procedure for training, I am confused about the inference process. How to select the action give certain state in inference? The authors are suggested to give some explanations in the Algorithm 1&2.

**Questions:**

Line 150， the sentence should be "in this way"?

The estimated reward/transition distribution $\hat{R}^k_h$ and $\hat{P}^k_h$ are key components for the proposed method, it is suggested to give more details on the distribution estimation part.

For both cases, the authors proposes the bonuses, I am confused why do we need the bonus? only for calculating the variance value? However, even without the bonus value, we can also update the value function $\bar{V}^k_h(s)$, right?

---

> ### Author Rebuttal · Authors · 2024-08-04
>
> We thank the reviewer for the response and refer to the general comment on empirical simulations.
>
> We apologize for any clarity issue and will make an effort to clarify the setting and the algorithm. Our paper studies an online setting where we repeatedly interact with an unknown environment in episodes, with the goal of collecting as much reward as possible throughout the interaction. Thus, the training and inference are interleaved - on each episode, we aim to use all the historical data to maximize the reward, on the one hand, while continuing to explore to improve future performance, on the other hand. The regret is a measure of how well we perform this task - can we collect rewards almost as much as the optimal policy that knows the environment in advance?
>
> Every episode starts with a planning phase, where we calculate an optimistic value (to ensure exploration), and then continue with interaction for a full episode ($H$ steps) with the environment. When each episode ends, the environment resets to a new initial state and the process repeats itself for $K$ episodes. For example, in Algorithm 1, lines 4-6 represent the planning stage, while line 9 represents the deployed policy for this episode. In Appendix B.3, we provide a more detailed version of the algorithm, where the planning stage is on lines 4-10 and the deployed policy is given on line 11.
>
> With reward lookahead, every time we reach a state $s$ at step $h$, we first observe the rewards $R(a)$ for all actions and then use our approximate planning to pick up an action that maximizes the expected long-term gain from an action $a_h \in \arg\max_a [ R(a)+\sum_{s’}\hat{P}_h(s’ | s, a){\bar{V}} _{h+1}(s’)+ bonus]$. The first term accounts for the immediate *reward observation* while the second handles future values. The bonus ensures optimism/exploration. Importantly, we start without knowing anything about the environment, and without the bonuses, we would not be incentivized to visit new parts of the state space. As a result, we would not have data on these unvisited areas and would converge to a suboptimal policy. This is similar to the way that UCB1 does exploration in bandits, just adapted to RL. With transition lookahead, we apply similar principles, but the agent observes the next state for any action before acting instead of the rewards.
>
> If we understand correctly, the reviewer considers an altenative setting, in which either the data is an offline data set and we need to find a good policy to deploy (‘offline RL’), or where we interact with an environment to gather data that will allow calculated a good policy (‘best-policy identification’) - while lookahead could definitely be studied there, we focused on online regret minimization.
>
> # Questions:
>
>  * Line 150 - noted, thanks!
>
> * We will further elaborate on how the empirical distributions are calculated in the final version of the paper - thanks for the comment. The distributions are essentially uniformly sampling from a buffer: we keep all past reward/transition observations, and every time we want to sample an observation, we just take what we saw at a uniformly random past time step. More specifically, since we calculate an expectation over these distributions, this is just an empirical average using past observations - the explicit formula is provided in line 9 of Algorithms 3 and 4 (which are in the appendix). In practical implementations, it could be adapted using a finite buffer that tries to keep diverse information on different regions of the state space, or more elaborate distribution estimation mechanisms.
>
>     For transition kernels, we slightly abuse notations - we denote by $\hat{P}_h^k(s’ | s,a)$ - the empirical probability to transition from state $s$ to $s’$ when playing $a$ at step $h$ - the fraction of times in the past that we took action $a$ at state $s$ and transitioned to $s’$. The distribution $\hat{P}_h^k(s)$ is the empirical joint distribution of the next states at $s$ across all actions (that is, an $A$-dimension vector of next-states). It is calculated using the sampling from a buffer, as discussed in the previous paragraph.
>
> * The bonuses are needed for exploration, we hope that the previous part of the response clarified it, but if not, we will gladly provide additional clarifications.

---

### Official Review · Reviewer_49N6 · 2024-07-11

**Soundness:** 3
**Presentation:** 3
**Contribution:** 3
**Rating:** 7
**Confidence:** 3

**Summary:**

The paper considers the setting where the agent can see the possible next rewards and next states without assuming a prior knowledge of the environment dynamics. The predicted next rewards and next states are estimated by empirical distribution. The paper considers extending Monotonic Value Propagation to such a setting and proves that the proposed algorithms can achieve tight regret bounds.

**Strengths:**

- A tight regret bound is proved for the proposed algorithm, establishing theoretical justification for lookahead information and advantages of planning in RL in general.
- The paper does not assume known environment dynamics as in most previous works, which makes the algorithm applicable to standard RL settings. The lack of known environment dynamics may bring various challenges to planning, such as agents not relying on the lookahead information when the estimated environment dynamics are still far from the true one in the early stages. The paper shows that the lookahead information can still be very beneficial despite such challenges.
- The paper is well-written, and the proof is easy to follow.

**Weaknesses:**

Even though a tight regret bound has been proved, empirical experiments with examples showing how the agent uses the lookahead information will strengthen the paper.

**Questions:**

There has been prior work in deep RL that makes use of lookahead information even when the environment dynamic is unknown. One example is the Thinker algorithm [1], which allows agents to select an imaginary action trajectory to collect n-step lookahead information before selecting an action in each step (the environment dynamics are also assumed to be unknown). The related work section should be updated to reflect this (e.g. line 73-79). However, as these works are mostly empirical without proving the regret bound, I still recommend that the paper be accepted, given its theoretical significance.

[1] Chung, Stephen, Ivan Anokhin, and David Krueger. "Thinker: learning to plan and act." Advances in Neural Information Processing Systems 36 (2024).

**Limitations:**

The authors adequately addressed the limitations.

---

> ### Author Rebuttal · Authors · 2024-08-04
>
> We thank the reviewer for the positive comments and refer to the general response for a detailed discussion on simulations in our paper. We also provide an additional example of the advantage of lookahead information in the response to reviewer MArZ, and discuss how the distribution affects the lookahead value in the response to reviewer Dj8h - we are willing to further discuss this in the final version of the paper.
>
> We acknowledge that in our literature survey, we focused on previous theoretical work; we apologize for that and will try to extend our survey to also cover more practical works, including [1]. Yet, we would like to point out that most previous works consider lookahead exclusively as a planning mechanism - that is, use it to calculate a better policy in the standard RL feedback model. In our framework, we know that we will encounter the exact same trajectory both in planning and when interacting with the environment, and this information could sometimes be leveraged to obtain a much higher value (see also the example in the response to reviewer MArZ that illustrates it). The two lookahead notions intersect when the environment is deterministic. On the other hand, in stochastic environments, the agent can actively take advantage of the stochasticity: learn the observation distribution in future states and decide in which future state observing rewards/transitions before acting would be more beneficial. Another potential intersection could appear when analyzing multi-step lookahead information - then, lookahead planners might be applied to effectively utilize lookahead information even in stochastic environments. We will further discuss this in our work.

---

> > ### Comment · Reviewer_49N6 · 2024-08-11
> >
> > Thanks for the response. I maintain my score after reading the responses and reviews.
> >
> > One thing to note - works like [1] can usually be used to perform what you described. For example, in [1], one can replace the learned world model (the state-reward network in the dual network that is proposed) with the known environment dynamics so the agent can learn to take advantage of the stochasticity as in your example. The main difference with your proposed algorithm is that the agent has to decide which action to try instead of seeing all possible action's consequences. This, however, is essential for multi-step lookahead, as one cannot enumerate all action sequences with a decent depth.

---

> > > ### Author Response · Authors · 2024-08-11
> > >
> > > We thank the reviewer again.
> > >
> > > Combining multi-step lookahead with the decision of which future information to query is indeed a great question, also from a theoretical point of view! We will read the work in further detail and discuss the relations in the final version of the paper.

---

### Official Review · Reviewer_MArZ · 2024-07-14

**Soundness:** 3
**Presentation:** 3
**Contribution:** 2
**Rating:** 5
**Confidence:** 4

**Summary:**

The paper studies an RL problem with a special setting, called one-step lookahead, where the agent can observe the reward or the state at the next step before the current action is taken. The paper focuses on the problem with an unknown environment (transition function). The authors proposed an efficient algorithms leveraging the empirical distribution of the lookahead information and claimed that the algorithms achieve tight regret against a strong baseline.

**Strengths:**

1. The paper studies an interesting RL problem where one-step lookahead information is available to the agent while the environment is unknown.

2. The paper clearly presents the problem, the solution, and a comparison between the proposed algorithm and the baseline in terms of regret bound.

3. The paper offers explanation of the terms in the regret bounds and justified its explanation.

**Weaknesses:**

1. One concern is the application of such a lookahead setting. The agents during training and running needs to know what will be realized in order to make actions at the current state. Not sure what real-world scenarios this setting can be applicable to.


2. RL with lookahead information has been investigated before from a theoretical point of view. See [R1, p64]. [R2] [R3]. [R1] discusses the lookahead in the approximation of the bellman function. [R2-R3] considers controlled lookahead where the agents decide the step of lookahead as a strategy. It is not straightforward to see in this paper how the lookahead studied in this paper different from those references.

[R1] Bertsekas, Dimitri. Reinforcement learning and optimal control. Vol. 1. Athena Scientific, 2019.
[R2] Biedenkapp, André, et al. "TempoRL: Learning when to act." International Conference on Machine Learning. PMLR, 2021.
[R3] Huang, Yunhan, Veeraruna Kavitha, and Quanyan Zhu. "Continuous-time markov decision processes with controlled observations." 2019 57th Annual Allerton Conference on Communication, Control, and Computing (Allerton). IEEE, 2019.


3.  It is not clear the source of the baseline mentioning in the paper. For example, "compared to a stronger baseline that also has access to lookahead information". The paper should includes the reference whenever the baseline is compared with the proposed solution.

**Questions:**

1. When the agents have access to both reward lookahead and transition lookahead, how would the regret bound be different?

2. Why doesn't the paper present a case study that illustrates how would the agent behave differently between a normal RL setting and a lookahead setting?

**Limitations:**

Discussed in the Weakness section.

---

> ### Author Rebuttal · Authors · 2024-08-04
>
> We thank the reviewer for the feedback.
> # Weaknesses
> 1.There are numerous applications where exact or approximate lookahead information is present:
> * Transactions/Market interaction - whenever the agent performs transactions, the traded items and their prices are mostly observed before the trade takes place - reward lookahead. This is also relevant for problems such as inventory management.
> * Communication networks - some networks continuously monitor the communication quality in different channels and use this information to choose which channel to use/avoid. This is similarly relevant in routing or situations where a protocol dictates when/where data could be sent.
> * Packing - When trying to pack items optimally, the learner often observes the next item/few items and uses this observation to decide how to pack the current item. One famous example is Tetris, where the next block is observed. This can also be extended to systems with an observable queue.
> * Navigation - depending on the problem, traffic information could be translated to either reward or transition lookahead. A related example is ride-sharing - travelers’ future location can be seen as future reward information.
> * Weather-dependent applications - weather predictions are extremely accurate in the near future, and can determine either the transition or reward (depending on the application)
> * Electricity grids - Electricity consumption can be accurately predicted in the near future, as well as electricity supply.
>
> Some of these applications fit well to our model, while in others, the agent gets noisy predictions/multiple steps in the future - we believe that the numerous potential extensions/generalizations will further motivate the community to work on similar models.
>
> 3.When we measure regret, we calculate the difference between the value of the *optimal lookahead agent* and the value of the learning agent. The value of the optimal lookahead agent is much higher than the value of agents without lookahead (see the examples at the beginning of Sections 3 and 4, which explain why this value can be larger by a factor of roughly $AH$ and $A^H$, respectively, and an additional example below). When we talk about a stronger baseline, we talk about this optimal lookahead value - we want our agent to achieve sublinear regret compared to it and not compared to the standard no-lookahead value. Standard off-the-shelf RL agents cannot converge to the optimal lookahead value and will suffer linear regret vs. this baseline.
>
> 2.We thank the reviewer for the references and apologize that our discussion in the related work is not clear enough. There are two concepts of lookahead in the literature. In the learning community (and as far as we could see, also in [R1]), it is mainly used as a planning technique - that is, the model is not accurate enough/too complex for long-term planning, and lookahead is used as a computational tool to estimate a policy with high ‘standard’ (no-lookahead)  value. For example, in [R1], it is motivated as an approach that mitigates the influence of errors in future value estimations. In particular, the aim is still to calculate the optimal Markovian policy, and lookahead is a means to calculate it.  We, on the other hand, aim to achieve the higher lookahead value (as done in some existing formulations of MPC). This is not a semantic difference - the algorithm actively utilizes the fact that additional information will be revealed in the future to achieve higher values. To our understanding, this difference also holds in [R2-R3] - they do not rely on new future information but rather plan how to act multiple steps to the future. We will extend our discussion to cover this and include the suggested references.
> ## Example
> To further illustrate the difference in values, consider the following 3-state environment.
> * The agent starts at $s_1$ and has two options: move to state $s_2$ and gain 0 reward or transition to state $s_3$ and deterministically gain a reward of $R=0.6$.
> * In $s_2$, two actions lead to state $s_3$, each giving a reward of $Ber(0.5)$.
> * $s_3$ is a no-rewarding terminal state.
>
> Without lookahead information, an optimal agent would move directly from state $s_1$ to $s_3$, earning a value $V=0.6$. This is also true for agents calculated via rollouts/lookahead planning: the possible trajectories are $s_1->s_3$ and $s_1->s_2->s_3$, and if we use rollouts of length 3, we get that the trajectory $s_1->s_3$ is better in expectation (value $0.6$ compared to $0.5$ when going through $s_2$).
>
> In contrast, in our setting, we assume to have lookahead information, that is, once we reach a state, we know that rewards will be revealed before choosing an action. Specifically, when reaching $s_2$, we can pick the action with the maximal *realized* reward - collect a unit reward w.p. $0.75$. In state $s_1$ we can still only earn $R=0.6$, so the optimal policy changed: now it is optimal to take the path $s_1->s_2->s_3$ (with value $V=0.75$). In other words, even though the expected rewards in $s_2$ are lower, the agent decides to go there because the rewards are more stochastic, and it relies on the fact that when observing the rewards, this stochasticity can be used to collect a higher value.
> # Questions
> 1. When having both reward and transition lookahead, we believe that our algorithms could be naturally generalized by estimating both distributions simultaneously. Then, using similar techniques, it should be possible to prove a regret bound of $O(\sqrt{H^4SAK})$ or better. We also suspect that the lower bound for this setting is $\Omega(\sqrt{H^3SK})$, so the hypothesized upper bound would be tight up to a factor of $\sqrt{SA}$.
> 2. We intended that the examples at the beginning of Sections 3 and 4 would be the illustrative examples that show the difference between situations with and without lookahead. We will try to further clarify them - we would appreciate any further feedback on what is missing in the examples.

---

### Author Rebuttal · Authors · 2024-08-04

## Experiments

Some of the reviewers expressed concern due to the lack of experiments in the paper.

While conducting experiments is always interesting, our paper theoretically studies a new setting for which there are no existing algorithms with theoretical regret guarantees. Thus, when comparing our approach to any existing algorithms that converge to the ‘standard’ no-lookahead value (which is much lower than the lookahead value), the difference between the values will dominate the performance. As such, it is very challenging to devise experiments that provide meaningful insights beyond the fact that the algorithm converges (which the proof already ensures). On the other hand, it is possible to discuss both theoretically and numerically the difference between the standard values and lookahead values, to demonstrate how much we can gain from utilizing lookahead information. We refer to the response to reviewer Dj8h for an additional discussion on lookahead vs. no-lookahead values in various scenarios. If the reviewers think it would be beneficial, we could extend this discussion and provide plots that illustrate the difference between the two values in the paper.

We agree with the reviewers that further pursuing the empirical study of this setting is of great importance due to its vast potential applications (as we detail in the response to reviewer MArZ). However, for this to be relevant for large-scale applications, this evaluation should be done while adapting our approach to deep RL, which is outside the scope of our paper. Our scope is more theoretical - we formalize a new setting with numerous potential applications and fully analyze planning and learning. In particular, the regret bounds of our algorithms are minimax-optimal (‘converge at the fastest possible rate for some problems’). Moreover, our algorithms and proofs have many novel elements: instead of estimating the expected reward/transition kernel, our algorithms work with the joint reward/next-state distribution and integrate distribution estimation into planning and learning; we show how to modify the classic regret analysis techniques to bypass the strong dependence that the setting creates between actions and rewards/next states;  we demonstrate how to use the list representation of the policy to obtain tighter regret bounds; and more.

---

### Comment · Area_Chair_b1cx · 2024-08-11
**Questions**

Dear authors,

Thanks for the rebuttal. As I go over the reviews and your rebuttal, I have some quick questions which I hope to get some clarifications:

1. The paper refers to lower bounds in standard (i.e., no lookahead information) RL settings in several places (e.g., Line 190). Since the current setting provides additional information to the agent, one would expect that the upper bound you obtain to penetrate through the lower bounds in standard settings (for agents without such additional information), but I couldn't find a crisp statement on this in the paper.

More concretely, according to Table 1 in Jin et al. (2018), best upper bound is $\sqrt{H^2 SA T}$ (their $T$ is your $K$, and I also verified the reward boundedness conventions in two papers which are identical). Both of your results (Line 48) seem no better than this bound (the reward lookahead one is worse by $\sqrt{T}$, and the transition look-ahead one has an extra additive term)? I would appreciate a clarification here. If your result penetrates through the standard RL lower bound, a crisp, unambiguous complexity comparison will be very helpful.

* I also noticed the $\sqrt{H^3 SA T}$ lower bound on Line 191, which contradicts the upper bound I quoted above. Likely this is due to whether transition is time homogeneous...?

2. I agree with Line 149 that state augmentation is the most natural idea. The paper discusses why this approach does not work, but I am not entirely convinced by the argument. Reward continuity is discussed as a major difficulty (Line 40), but (1) one can usually discretize reward to get around it, and (2) reward learning is relatively easier than transition learning (as discussed on Line 289) and usually doesn't dominate the final rate. So one can simply ignore the reward lookahead information and solely focus on the transition lookahead information, in which there is no concern of continuity. So what is the main difficulty here? I imagine the problem is that since there is additional information for every action, the state space will blow-up in a way that may be exponential in A. Regardless, I think a formal description of the augmented MDP (define state, action, reward, transition, etc., after the augmentation) will be very useful, possibly in the appendix.

AC

---

> ### Author Response · Authors · 2024-08-11
>
> Dear area chair,
>
> Thank you for your questions.
>
> 1. The key reason that our paper breaks through the lower bound is the different definition of regret, and in particular, the definition of the optimal value. Specifically, denote $V^*$ the standard no-lookahead value used in RL, and $V^{R, *}$ the reward lookahead value, with the corresponding regret bounds $Reg$ and $Reg^R$.
>
>     Notice that for any agent (lookahead or not), one can always write $Reg = (V^* - V^{R,*})K + Reg ^R$.
>
>    By definition, the lookahead agent is more powerful, so mostly $V^* < V^{R,*}$. Since the first term in the decomposition above is negative linear while the second term is just sublinear, any difference between the values imposes a *negative linear regret* - completely breaking the standard lower bounds in RL.
>
>     For example, in Section 3, we described situations where $V^*\approx 1/AH$ while $V^{R,*}\approx 1$. Then, in terms of standard regret, for large enough $K,A,H$ we would get $Reg \approx -K$. We can add this explanation to the paper.
>
> * In Jin et al. (2018), they have $T=HK$ (see caption under table 1), where $K$ is the number of episodes (as in our paper). Therefore, the lower bounds are the same - both allow non-stationary transition.
>
> 2. While reward learning doesn't usually dominate the rates, it is due to horizon factors: reward learning usually leads to a term of $\sqrt{H^2SAK}$ while transition learning to $\sqrt{H^3SAK}$. Even if we can just discretize the state for the reward component to a state $S^d$ (while not changing transition learning complexity), we would still have unavoidable dependence in the regret on $S^d$. Importantly, $S^d$ needs to encompass the reward observations for *all actions simultaneously*, so even if we use $N$ cells to discretize each reward component, we would get a term of $N^A$. Therefore, as soon as we do it, our bounds would explode (even if reward learning would just have affected the additive constant!!).
>
>     Moreover, moving the reward to the state means that reward observations are now state transitions, so using standard approaches would lead to a regret bound of $\sqrt{H^3S^dAK}$.
>
>     You are completely right about the transition lookahead - since we need to add the next state observation for all actions simultaneously, the state blows by a factor $S^A$.
>
>     We actually already use the augmented MDP for both proofs (mostly for the planning results) and describe in detail one way to do this extension in Appendices B.2 and C.2. If the AC thinks it would help clarity, we could further discuss it in the main paper. We could also discuss more about the implications of augmentation and discretization on learning, extending the discussion in this response.
>
> ----
>
> We hope that we clarified all the issues; we will gladly provide more clarifications if we did not convey the message clearly enough and answer any other questions.

---

> > ### Comment · Area_Chair_b1cx · 2024-08-11
> >
> > Thanks for the clarification and I think I understand now. Basically with additional information (lookahead) the optimal value of the MDP increases compared to vanilla agent, so standard algorithms that don't leverage the information can't compete. I was focusing on how the information helps with learning efficiency and omitted its role in planning.
> >
> > Re blow-up: the explanation makes sense. As a minor change, perhaps it will be good to point to the construction in the appendix from the main text, and whenever the paper uses the word "exponential", say exponential-in-what (action, dimension, etc.).

---

> > > ### Author Response · Authors · 2024-08-11
> > >
> > > Yes, exactly!
> > >
> > > Thanks for the improvement suggestions - we will correct all these issues in the final version of the paper. We will also reflect on our writing and presentation to make sure that all the questions that were raised in the discussion here will be clearer.

---

### Decision · Program_Chairs · 2024-09-25

**Decision:**

Accept (poster)

**Comment:**

The reviewers unanimously agree that the paper studies an interesting problem of RL with additional side information ("lookahead"). The authors are advised to revise the paper according to the comments given by the reviewers and the AC, particularly regarding comparison to vanilla RL without side information.